# FedDAG: Clustered Federated Learning via Global Data and Gradient Integration for Heterogeneous Environments

**Anik Pramanik**[1]**, Murat Kantarcioglu**[2]**, Vincent Oria**[1]**, Shantanu Sharma**[1]
[1]New Jersey Institute of Technology, USA     [2]Virginia Tech, USA

## Abstract

Federated Learning (FL) enables a group of clients to collaboratively train a model without sharing individual data, but its performance drops when client data are heterogeneous. Clustered FL tackles this by grouping similar clients. However, existing clustered FL approaches rely solely on either data similarity or gradient similarity; however, this results in an incomplete assessment of client similarities. Prior clustered FL approaches also restrict knowledge and representation sharing to clients within the same cluster. This prevents cluster models from benefiting from the diverse client population across clusters. To address these limitations, FedDAG introduces a clustered FL framework, FedDAG, that employs a weighted, class-wise similarity metric that integrates both data and gradient information, providing a more holistic measure of similarity during clustering. In addition, FedDAG adopts a dual-encoder architecture for cluster models, comprising a primary encoder trained on its own clients' data and a secondary encoder refined using gradients from complementary clusters. This enables cross-cluster feature transfer while preserving cluster-specific specialization. Experiments on diverse benchmarks and data heterogeneity settings show that FedDAG consistently outperforms state-of-the-art clustered FL baselines in accuracy.

## 1 Introduction

Federated Learning (FL) enables users/clients to collaboratively train a model on their data without sharing it with other clients or a central entity (McMahan et al., 2017). However, diversity in user behavior results in heterogeneous data distributions, known as *non-identically independently distributed* (non-IID) data, across clients. This heterogeneity can lead to slower convergence and suboptimal accuracy of the global model (Kairouz et al., 2021). More specifically, non-IID data can arise due to various factors, including class/label skew, feature skew, quantity shift, concept shift, and concept drift — common types of data heterogeneity. *Class/label skew* refers to the non-identical distribution of labels/classes at different clients, e.g., the absence of a label at one client while the same label is present at other clients (Zhang et al., 2022a). *Feature skew* occurs when distributions vary due to different personalization nuances, e.g., an alphabet letter can be written in different ways (Li et al., 2021b). *Quantity shift* happens when different clients have different amounts of data (Wang et al., 2021), e.g., an online retailer with millions of transaction records is compared to a local store with only a few hundred records. *Concept shift* happens when different clients assign the same label to fundamentally different data samples due to variations in local data distributions or labeling criteria (Kang et al., 2024).

Clustered FL handles non-IID data effectively, especially when distinct groups of clients display substantial variations in their data distributions (Ghosh et al., 2020; Guo et al., 2024; Vahidian et al., 2023). In clustered FL, clients are grouped into clusters based on their similarities in their data distributions, and each cluster trains its own model tailored to its specific data. However, despite their advantages, existing clustered FL approaches suffer from the following limitations:

**1. Improper Similarity Method.** Cluster FL approaches use either data or gradient alone to compute similarity for clustering. Cluster FL approaches (Sattler et al., 2020; Long et al., 2023; Ghosh et al., 2020) that use gradients or loss values to cluster clients can group clients incorrectly due to the

high dimensionality of data or the presence of various skews in client data (Vahidian et al., 2023). Other drawbacks of these approaches include: requiring each client to evaluate multiple global models every round (Ghosh et al., 2020; Licciardi et al., 2025), delaying cluster formation until many training iterations, and requiring clients to upload full model updates (Sattler et al., 2020).

On the other hand, the data-based approach, such as PACFL (Vahidian et al., 2023), only considers label skew and does not account for skew issues like concept shift. Moreover, PACFL defines inter-client similarity as the minimum cosine angle between the clients' feature subspaces. However, by relying on the smallest angle across the subspaces, PACFL may yield high similarity even when only a small portion of the clients' data is similar, while the remaining subspaces are vastly dissimilar.

**2. Global Representation Sharing.** Existing Clustered FL approaches restrict knowledge sharing to clients within the same cluster. This prohibits clients across clusters to benefit from low-level latent representations. One way FedSoft (Ruan & Joe-Wong, 2022) and FedRC (Guo et al., 2024) address this issue by incorporating multiple cluster models through soft clustering with learnable cluster importance weights. However, in these approaches, a client's model becomes a noisy blend of several cluster models. While this blending may occasionally benefit data that aligns with several clusters, the added noise from unrelated clusters may degrade the performance on the client's primary dataset, since the model is no longer explicitly optimized for its own data.

**3. Limited Consideration of Distribution Skews.** Clustered FL techniques (Sattler et al., 2020; Ghosh et al., 2020; Vahidian et al., 2023; Licciardi et al., 2025) primarily address label skew. However, these approaches do not account for concept shift or quantity shift.

**4. Predefined Cluster Numbers.** Existing clustered FL approaches lack adaptive mechanisms for automatically adjusting the number of clusters. For example, IFCA (Ghosh et al., 2020) requires the optimal number of clusters to be specified in advance. Sattler et al. (2020) adopts a recursive strategy to split clusters when gradients converge to a stationary point but cannot merge clusters when needed, such as upon the arrival of new clients. Zeng et al. (2023) supports merging clusters but not splitting them. Li et al. (2024a) evaluates candidate clustering using traditional clustering metrics that do not account for the unique characteristics of FL setting. These limitations raise the following crucial question:

*How can we overcome the above challenges posed by various skews in heterogeneous data distributions by utilizing both data and gradient information to dynamically cluster clients and enabling representation sharing among clusters in FL?*

**Our contribution.** This work proposes a novel algorithm, entitled clustered Federated Learning via global DatA and Gradient integration (FEDDAG). FEDDAG introduces a novel method to compute similarities among clients and an innovative approach that combines data and gradient information for improved client grouping. To combine data- and gradient-based similarity to achieve a more accurate similarity matrix, FEDDAG assigns each client a weight that indicates how much emphasis to place on data versus gradient information. FEDDAG optimizes these weights using an entropy-based loss that sharpens the final adjacency matrix. To further improve client similarity estimation, FEDDAG extends the data-based approach PACFL (Vahidian et al., 2023) by performing class-wise comparisons rather than comparing entire data subspaces—restricting comparisons to subspaces corresponding to the same class across clients. This approach yields a more accurate similarity metric and naturally accounts for concept shift. In addition, FEDDAG assigns weights to the class-wise similarity values to address quantity shift. FEDDAG also improves upon the existing gradient-based similarity so that client computes gradients for at most one model per round and transmits only a compressed gradient.

These above mechanisms improve similarity computation and lead to better client clustering. We further enhance FEDDAG by employing a dual-encoder architecture to enable effective representation sharing across clusters. During the training phase, each cluster model consists of: (i) a primary encoder, optimized using the cluster's own client data, and (ii) a secondary encoder, designed to learn complementary features from other clusters. The outputs of the two encoders are concatenated along the feature dimension, and a classifier is trained on the combined representation. This design facilitates cross-cluster knowledge transfer while preserving cluster-specific specialization.

Compared to prior works, to our knowledge, FEDDAG is the only work that addresses all four types of data heterogeneity: label skew, feature skew, concept shift, and quantity shift. FEDDAG accounts for concept shift by performing class-wise comparisons when computing similarity between clients'

data. Additionally, FEDDAG introduces an adaptive clustering mechanism that automatically determines the optimal number of clusters through a novel evaluation metric. Specifically, it generates a range of candidate clusterings using hierarchical clustering (HC) (Day & Edelsbrunner, 1984) and evaluates them with a novel federated-aware metric that rewards compact cluster formation while penalizing over-splitting.[1] In summary, the contributions of this paper are as follows:

1. A new clustered FL algorithm, FEDDAG, that combines both data and gradient similarity for better client clustering and improves data similarity estimation with a class-wise weighted method.
2. FEDDAG introduces a novel method for knowledge and representation sharing across clusters by employing a dual-encoder architecture.
3. This work introduces a novel federated-aware metric to evaluate candidate clusterings and automatically determine the optimal number of clusters.
4. We evaluate FEDDAG under non-IID data, having class skew, feature skew, concept shift, and quantity shift, and across different degrees of heterogeneity (e.g., high vs. low). Table 1 reports the accuracy of FEDDAG in comparison to existing clustered FL methods. Detailed experimental results are provided in §5.

> **The full version of the paper and code is available at** `https://tinyurl.com/2rbkb3zu`.

## 2 LITERATURE REVIEW

There exists an extensive body of work on improving the performance of FL in data-heterogeneous environments via clustered FL, knowledge distillation, meta-learning, data augmentation, and related techniques. Below, we summarize the approaches most relevant to our work; additional related directions are discussed in Appendix A.1.

Table 1: Accuracy (%) of FEDDAG vs. clustering baselines under non-IID label skew (20%) and quantity shift (Dirichlet $\alpha'=1$).

| Algorithm | Technique | CIFAR-10 | FMNIST |
|---|---|---|---|
| PACFL | Data (D) | 90.45±0.30 | 94.41±0.31 |
| CFL | Gradient (G) | 72.80±0.66 | 86.97±0.23 |
| IFCA | Gradient (G) | 89.68±0.17 | 94.03±0.09 |
| **FEDDAG (Ours)** | **D + G + Global Feature Sharing** | **94.53±0.12** | **96.82±0.18** |

**Clustered FL** techniques address distribution shift by grouping clients based on their data distributions. PACFL (Vahidian et al., 2023) clusters clients by analyzing principal angles between client data subspaces, but it ignores label information, making it prone to incorrect clustering under concept shift. Ding & Wang (2022) constructs $K$ shared models based on each client's dataset contribution. Another line of work (Ghosh et al., 2020; Licciardi et al., 2025) uses loss values on gradients to iteratively cluster clients each training round. Other methods group clients via gradient or parameter similarity (Sattler et al., 2020; Zhang et al., 2024), while soft clustering enables clients to join multiple clusters (Ruan & Joe-Wong, 2022; Guo et al., 2024). Additional methods, such as Long et al. (2023); Marfoq et al. (2021); Wu et al. (2023), rely on maximizing log-likelihood functions or modeling joint distributions. Compared to these methods, FEDDAG combines data and gradient information for better clustering and enables knowledge sharing across clusters.

**Knowledge distillation (KD)** approaches such as Lin et al. (2020); Li & Wang (2019) use a global dataset to transfer knowledge from local teacher models to a global student model. FedFTG (Zhang et al., 2022b) trains a generator to approximate the input space of local models and uses it to generate pseudo-data. Another line of work, data-free KD, generates pseudo-data directly from a pretrained teacher model to perform knowledge distillation (Guo et al., 2023; Chen et al., 2019). DeepImpression (Nayak et al., 2019) recovers approximate real data by modeling the output space of the teacher model, while DeepInversion (Yin et al., 2020) further refines pseudo-data by regularizing the distribution of intermediate feature maps. Instead of relying on a public/pseudo dataset our proposed FEDDAG's global parameters are updated directly using data from complementary source clusters, enabling cross-cluster knowledge sharing.

## 3 FEDDAG ALGORITHM

FEDDAG, a framework for clustered FL, can be formulated as an empirical risk minimization (ERM) problem over $N$ clients, each holding a local dataset $D_i=(X_i, Y_i)$, where $X_i$ and $Y_i$ de-

---

[1] Over-splitting is a common issue in HC for FL that can violate key principles of FL by producing degenerate clusters with very few clients (Licciardi et al., 2025).

note the input samples and labels, respectively. The data can be non-iid and may exhibit various skews (as discussed in §1). The server partitions the clients into $Z$ clusters $\mathbb{C}_1, \ldots, \mathbb{C}_Z$. The objective is to minimize the local loss $\mathcal{L}(Y_i, F_{z(i)}(X_i))$ for each client $i \in N$, where $z(i)$ is the cluster assignment determined by FEDDAG. Simplified FEDDAG cluster-level model is defined as:

$$F_z(\cdot) = \psi\big(\phi(\cdot; \Theta_z^f);\ \Theta_z^c\big) \tag{1}$$

Here, $\phi$ is the feature encoder and $\psi$ is the classifier head. FEDDAG also supports a more expressive dual-encoder architecture, where the outputs of two encoders are jointly processed by the classifier head, as represented below:

$$F_z(\cdot) = \psi\big(\phi^{(1)}(\cdot; \Theta_z^{1f}), \phi^{(2)}(\cdot; \Theta_z^{2f})\ ;\ \Theta_z^c\big) \tag{2}$$

We describe FEDDAG (see Algorithm 1) in two parts. First, we introduce the weighted class-wise approach (Algorithm 2 in Appendix A.3) for computing data similarity among clients and combine both data and gradient to improve clustering (Algorithm 3 in Appendix A.3). The improved clustering can be directly used for traditional clustered FL, resulting in higher accuracy (see §5). We then further enhance FEDDAG with a dual-encoder mechanism (described in §4) that enables inter-cluster representation sharing during FL training, which further increases FEDDAG's performance. An illustration of FEDDAG is shown in Figure 1 and its components are described below.

## 3.1 GRADIENT-BASED SIMILARITY

**High-level idea.** FEDDAG introduces a lightweight method for computing gradient similarity. Prior approaches such as Sattler et al. (2020) and Kim et al. (2024b) periodically send gradient updates to the server to measure client similarity. In contrast, our approach has each client first train locally on its own data (without federation) for a few rounds to partially converge the gradients. We observed that two such rounds (10 local steps each) are sufficient to achieve partial convergence, making inter-client similarity more distinguishable (see experiments on Local Steps ($t_g$) in Appendix §B.2). To further reduce communication, FEDDAG transmits a $k$-sparse version of the gradients (retaining only $k$ coordinates) to the server for similarity computation (Wangni et al., 2018).

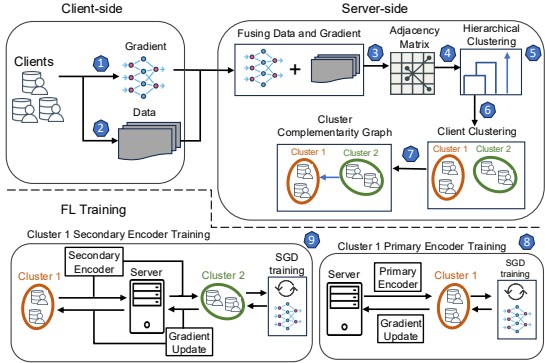

Figure 1: Overview of FEDDAG. Clients compute principal vectors and gradients to build an adjacency matrix and a graph indicating which clusters can supply features for cross-cluster sharing. Training proceeds in two phases: (1) the primary encoder and classifier are trained on each cluster's local data; (2) the secondary encoder of a requesting cluster is trained on source cluster's data

**Details of the method.** Each client $i \in N$ is initialized with random parameters $\theta_i^0$ and performs local training (without federation) on $D_i$ for $t_g = 2$ rounds (see Appendix §B.2) to obtain a gradient update $\Delta^i$. The update is $k$-sparsified—retaining only a small random subset of entries (typically 1–2%) (Wangni et al., 2018). The sparsified update $\tilde{\Delta}^i$ is then sent to the server, which constructs a pairwise similarity matrix. The similarity $\mathcal{G}_{i,j}$ between clients $i$ and $j$ is computed as:

$$\mathcal{G}_{i,j} = \cos^{-1}\left(\frac{\langle \tilde{\Delta}^i, \tilde{\Delta}^j \rangle}{\|\tilde{\Delta}^i\| \, \|\tilde{\Delta}^j\|}\right) \times \frac{180}{\pi}, \quad \forall i, j \in N. \tag{3}$$

## 3.2 WEIGHTED CLASS-WISE DATA-BASED SIMILARITY

**High-level idea.** Our goal is to construct a data-based similarity matrix that will be fused with the gradient matrix for clustering. Unlike the existing data-based approach, PACFL (Vahidian et al., 2023), which compares the entire data subspaces of two clients, we measure similarity in a class-wise manner and assign weights to the class-level similarities to compute the final client similarity.

**Details of the method.** Let $C$ be the total number of classes, and $D_{i,c}$ the data of client $i \in N$ for class $c \in C$. Each client applies truncated SVD (Klema & Laub, 1980) on the transpose of $D_{i,c}$ to compute $p$ principal vectors per class, denoted $U_c^i = [u_1, \ldots, u_p]$. These vectors are then sent to

---

**Algorithm 1:** FEDDAG Algorithm

---

**Input:** Number of clients $N$, sampling rate $R \in (0, 1]$, $C$ classes.     **Output:** Updated global model parameters

1  Initialize client $i \in N$ with random $\theta_i^0$
2  **for** *each round* $t = 0, 1, \ldots$ **do**
3      $m \leftarrow \max(R \cdot N, 1)$ // Sampling rate
4      $S_m \leftarrow \{i_1, \ldots, i_m\}$ // Set of $m$ sampled clients
5      **for** *each client* $i \in N$ **in parallel do**
6         **if** $t \le t_g$ **then**
7            Local training of $\theta_i^0$ with client $i$ local data (no federation)
8            **if** $t = t_g$ **then**
9               Client $i$ sends sparsified local model update $\tilde{\Delta}^i$ to server
10              Client $i$ performs SVD and extracts principal vectors $U_c^i$, $\forall c \in C$ and sends to server
11              Server forms $\mathcal{A} \leftarrow$ `ProximityMatrix`$(U^*, \tilde{\Delta}^*)$ (Algorithm 2)// Adjacency matrix
12              Server computes optimal Clustering $\{\mathbb{C}_1, \ldots, \mathbb{C}_Z\} \leftarrow$ `OptimalClustering`$(\mathcal{A}, S_\alpha)$
                (Algorithm 3)// Find best clustering
13              Server computes the *CC-Graph H* as per Eq. 12
14              Server initiates $\Theta_z^{1f}$ as in Eq. 22, and $\Theta_z^{2f}$ and $\Theta_z^c$ randomly // cluster encoder initialization
15        **else**
16           Server sends $\{\Theta_{z(i)}^{1f}, \Theta_{z(i)}^{2f}, \Theta_{z(i)}^c\}$ and $\Theta_{z(i)}^{2f'} = \sum_{j:H(j,z(i))=1} \Theta_j^{2f}$ to client $i$
17           Client $i$ sets $(\theta_i^{1f}, \theta_i^c) \leftarrow (\Theta_{z(i)}^{1f}, \Theta_{z(i)}^c)$ and trains them via SGD as in Eq. 17 // primary training phase
18           Client $i$ sets $\theta_i^{2f'} \leftarrow \Theta_{z(i)}^{2f'}$ and updates via SGD as in Eq. 20 // Secondary training phase
19           Client $i$ broadcasts $(\theta_i^{1f}, \theta_i^c)$ and $\theta_i^{2f'}$ to server
20     **if** $t \ge t_g$ **then**
21        **for** *each cluster* $z = 1$ *to* $Z$ **do**
22           Update $\Theta_z^{1f}$ and $\Theta_z^c$, as in Eq. 18
23           Update learner cluster $\Theta_{j:H(j,z)=1}^{2f}$, as in Eq. 21

---

the server to compute the data similarity matrix.[2] For each class $c$, the server computes the principal angle (Jain et al., 2013) between $U_c^i$ and $U_c^j$, indicating the similarity between clients $i$ and $j$ as:

$$\mathcal{V}'_{i,j,c} = \min_{\mathbf{v} \in U_c^i, \mathbf{x} \in U_c^j} \cos^{-1}\left(\frac{|\mathbf{v}^\top \mathbf{x}|}{\|\mathbf{v}\| \cdot \|\mathbf{x}\|}\right), \quad \forall i, j \in N. \tag{4}$$

If class $c$ is present in only one of the clients, $\mathcal{V}'_{i,j,c} = 90°$; if in neither, $\mathcal{V}'_{i,j,c} = 0°$. Next, the server assigns weights $\mathcal{W}_{i,j,c}$ to each class-wise similarity $\mathcal{V}'_{i,j,c}$ to reflect class frequency differences (i.e., quantity skew) between clients $i$ and $j$. This weighting scheme ensures that larger differences in class frequency lead to higher dissimilarity values. The weights are computed as:

$$\mathcal{W}_{i,j,c} = \frac{\max(\ln(|D_{i,c}| + \epsilon), \ln(|D_{j,c}| + \epsilon))}{\min(\ln(|D_{i,c}| + \epsilon), \ln(|D_{j,c}| + \epsilon))} \tag{5}$$

then min–max normalized to a bounded range $[1-\delta, 1+\delta]$, where $\delta > 0$ controls the server's tolerance to frequency imbalance. The final similarity between clients $i$ and $j$ is:

$$\mathcal{V}_{i,j} = \frac{1}{|C|} \sum_{c=1}^C \mathcal{V}'_{i,j,c} \mathcal{W}'_{i,j,c}, \qquad \mathcal{W}'_{i,j,c} \leftarrow \text{normalized } \mathcal{W}_{i,j,c}. \tag{6}$$

### 3.3 COMBINING DATA & GRADIENT — ALGORITHM 2

**High-level idea.** After constructing the data and gradient similarity matrices, FEDDAG applies min-max normalization and then combines them into a single proximity/adjacency matrix.

**Details of the method.** Given the normalized $\hat{\mathcal{V}}_{i,j}$ and $\hat{\mathcal{G}}_{i,j}$, FEDDAG learns a weight vector $\mathbf{w} = (w_1, \ldots, w_N)^\top \in [0, 1]^N$, where each $w_i$ is assigned to client $i$ to control the relative importance of gradient versus data similarity. FEDDAG then fuses the normalized matrices to construct the proximity matrix as follows:

$$\mathcal{A}_{i,j} = w_i \hat{\mathcal{G}}_{i,j} + (1 - w_i)\hat{\mathcal{V}}_{i,j}, \quad 1 \le i < j \le N, \quad \mathcal{A}_{j,i} = \mathcal{A}_{i,j}. \tag{7}$$

FEDDAG optimizes $\mathbf{w}$ by minimizing the entropy loss:

$$\mathcal{L}_{\text{en}} = -\frac{1}{N} \sum_{i=1}^N \sum_{j=1}^N \tilde{\mathcal{A}}_{i,j} \log \tilde{\mathcal{A}}_{i,j}, \quad \tilde{\mathcal{A}}_{i,j} = \frac{e^{\mathcal{A}_{i,j}}}{\sum_{k=1}^N e^{\mathcal{A}_{i,k}}} \tag{8}$$

---

[2]In FEDDAG, clients share a small set of principal vectors and class frequency information with the server to compute similarity. These principal vectors are not actual client data, but a linear combination of them. Moreover, the number of principal vectors shared with the server is less than 1% of the size of the dataset for each class per client. This approach aligns with prior works, such as PACFL(Vahidian et al., 2023).

where $\tilde{\mathcal{A}}_{i,j}$ is the row-wise softmax normalization of $\mathcal{A}_{i,j}$. In Eq. 8, the loss $\mathcal{L}_{\text{en}}$ sharpens each row of the fused matrix $\mathcal{A}_{i,j}$, encouraging each client to retain only its strongest neighbors (Ghasedi Dizaji et al., 2017). This, in turn, guides $\mathbf{w}$ to favor the view (i.e., data or gradient) that leads to a more clusterable affinity structure. FEDDAG learns the weight vector $\mathbf{w}$ using a lightweight multi-layer perceptron (MLP) (Almeida, 2020) trained via gradient descent to minimize the entropy loss $\mathcal{L}_{\text{en}}$. Finally, FEDDAG constructs the proximity matrix using the learned $\mathbf{w}$ as shown in Eq. 7.

### 3.4 OPTIMAL CLUSTERING — ALGORITHM 3

**High-level idea.** FEDDAG introduces an adaptive clustering mechanism that automatically identifies the optimal number of clusters. This mechanism incorporates a novel federated-aware metric to evaluate clustering quality.

**Details of the method.** Given the proximity matrix $\mathcal{A}_{i,j}$, the server applies agglomerative hierarchical clustering (HC). In HC, the *clustering threshold* $\alpha \in (0, 1]$ controls merges: clusters with pairwise distances below $\alpha$ are merged. Smaller $\alpha$ yields more clusters; larger $\alpha$ merges more broadly. The server iterates over different $\alpha$ values to generate candidate clusterings $\{\mathbb{C}_1, \ldots, \mathbb{C}_Z\}$, each with a distinct number of clusters $Z$. Each clustering is evaluated using two metrics. Compactness loss $\mathcal{L}_1$ promotes tight clusters, while degeneracy penalty $\mathcal{L}_2$ discourages small clusters:

$$\mathcal{L}_1 = \sum_{z=1}^{Z} \frac{1}{|\mathbb{C}_z|^2} \sum_{i,j \in \mathbb{C}_z} \mathcal{A}_{i,j}, \qquad \mathcal{L}_2 = \frac{1}{Z} \sum_{z=1}^{Z} \exp\left( \frac{\max\{0, \bar{\mathbb{C}} - \gamma \sigma_{\mathbb{C}} - |\mathbb{C}_z|\}}{\tau} \right) \tag{9}$$

where $\bar{\mathbb{C}} = N/Z$ and $\sigma_{\mathbb{C}}$ denote the mean and standard deviation of cluster sizes. A cluster $\mathbb{C}_z$ is penalized if size $|\mathbb{C}_z| < \bar{\mathbb{C}} - \gamma \sigma_{\mathbb{C}}$, with $\tau > 0$ controlling sharpness. The total loss is

$$\mathcal{L}_{\{\mathbb{C}_1, \ldots, \mathbb{C}_Z\}} = \mathcal{L}_1 + \lambda \mathcal{L}_2, \tag{10}$$

where $\lambda > 0$ balances the two terms. Lower $\mathcal{L}_1$ (tighter clusters) and $\mathcal{L}_2$ (less over-splitting) indicate better partitions. FEDDAG selects the clustering with the lowest loss and relatively few clusters.

## 4 GLOBAL REPRESENTATION SHARING (GRS)

**High-level idea.** In the previous section, we have combined data and gradient information to improve clustering. This section introduces global representation sharing across clusters during the training phase via a dual-encoder mechanism to further enhance FEDDAG's ability to learn complementary representations. The process for determining which clusters should complement each other and how training is carried out is described below:

**Building Cluster Complementarity Graph (CC-Graph).** We first construct a directed graph that identifies, for each cluster, which other clusters can supply the class representations it lacks. Intuitively, a cluster has a *demand* for a class if that class is underrepresented among its clients, and a *supply* if the class is well represented.

For each client $i$ and each class $c \in C$, let $m_i$ denote the number of distinct classes present on client $i$ and let $r_{i,c} \in \{0, \ldots, m_i - 1\}$ be the *rarity rank* of class $c$ on that client, where $r_{i,c} = 0$ means that $c$ is the rarest class on client $i$. For a single client $i$ and each class $c$, we define the client-level demand score as $(m_i - r_{i,c})$ and the supply score as $(r_{i,c} + 1)$, so that rarer classes induce higher demand while more frequent classes induce higher supply. To obtain cluster-level scores, we aggregate the client-level values. For a requesting cluster $\mathbb{C}_p$ and a source cluster $\mathbb{C}_q$, the demand and supply for class $c$ can be computed as shown below:

$$d_{p,c} = \sum_{i \in \mathbb{C}_p} (m_i - r_{i,c}), \quad s_{q,c} = \frac{1}{|\mathbb{C}_q|} \sum_{i \in \mathbb{C}_q} (r_{i,c} + 1), \tag{11}$$

where $d_{p,c}$ captures how strongly $\mathbb{C}_p$ lacks class $c$, and $s_{q,c}$ measures how abundantly $\mathbb{C}_q$ represents class $c$ on average. Combining demand and supply yields the *complementarity score* between a requesting cluster $p$ and a source cluster $q$:

$$H'_{p,q} = \sum_{c \in C} d_{p,c}\, s_{q,c}, \quad H'_{p,p} = -\infty. \tag{12}$$

A large value of $H'_{p,q}$ indicates that $\mathbb{C}_p$ has high demand for exactly those classes for which $\mathbb{C}_q$ has high supply. However, $H'_{p,q}$ only accounts for the relative quantity of each class and does not capture

the *quality* or alignment of the data between the two clusters. To make the CC-Graph sensitive to alignment, we incorporate the per-class principal-angle information $\mathcal{V}'_{i,j,c}$ between client subspaces (see Section §3.2) into the complementarity score. For each client pair $(i,j)$ and class $c$, we first clip the class-wise angle $\mathcal{V}'_{i,j,c}$ to the range $[0°, 90°]$ and then map it to $[0,1]$ as follows:

$$\Gamma_{i,j,c} = 1 - \frac{\mathcal{V}'_{i,j,c}}{90°}, \quad \bar{\Gamma}_{p,q,c} = \frac{1}{|\mathbb{C}_p|\,|\mathbb{C}_q|} \sum_{i \in \mathbb{C}_p} \sum_{j \in \mathbb{C}_q} \Gamma_{i,j,c}. \tag{13}$$

Here, the mapped value $\Gamma_{i,j,c}$ is close to 1 when the class-$c$ feature subspaces of clients $i$ and $j$ are well aligned and close to 0 when they are poorly aligned. To obtain a cluster-level alignment score $\bar{\Gamma}_{p,q,c}$, we average over all client pairs across clusters $p$ and $q$. Finally, we incorporate the alignment score into the demand–supply term $H'_{p,q}$ to compute a refined complementarity score as:

$$H_{p,q} = \sum_{c \in C} d_{p,c}\, s_{q,c}\, \bar{\Gamma}_{p,q,c}, \quad H_{p,p} = -\infty. \tag{14}$$

Here, a high value of $H_{p,q}$ indicates that $\mathbb{C}_q$ is a strong complementary source for $\mathbb{C}_p$: the terms $d_{p,c}$ and $s_{q,c}$ capture relative quantity, while $\bar{\Gamma}_{p,q,c}$ ensures that complementarity also reflects how well the corresponding class-$c$ representations are aligned between the two clusters.

Finally, we sparsify this score matrix into a directed adjacency matrix. For each row $p$, we keep only the top-$k$ largest values $H_{p,q}$ to build the CC-Graph. An edge $p \to q$ in this CC-Graph indicates that cluster $\mathbb{C}_p$ will receive class representations from cluster $\mathbb{C}_q$.

**Training using dual encoders.** For each client $i \in \mathbb{C}_z$, the prediction model can be described as:

$$F_z(X_i) = \psi\Big(\phi^{(1)}(X_i; \Theta_z^{1f}), \phi^{(2)}(X_i; \Theta_z^{2f})\,;\, \Theta_z^c\Big) \tag{15}$$

FEDDAG optimizes the parameters $\{\Theta_z^{1f}, \Theta_z^{2f}, \Theta_z^c\}_{z=1}^Z$ to minimize the weighted empirical loss across $N$ clients. This is achieved through parallel training phases of the primary and secondary encoders. During the primary phase for each cluster, the primary encoder $\Theta_z^{1f}$ and the classifier $\Theta_z^c$ are optimized using data from clients $i \in \mathbb{C}_z$, enabling the model to learn its own cluster-specific features. During the secondary phase, cluster $\mathbb{C}_z$ enriches the secondary encoders of clusters that seek to learn from it, as directed by the CC-Graph $H$. The procedures for both phases and their unified training strategy are detailed below.

**(i) Primary encoder training.** For each cluster, we optimize the primary encoder $\Theta_z^{1f}$ and the classifier $\Theta_z^c$ via gradient descent, while keeping the secondary encoder $\Theta_z^{2f}$ fixed. To approximate this, each client $i \in \mathbb{C}_z$ initializes its local parameters as $(\theta_i^{1f}, \theta_i^c) \leftarrow (\Theta_z^{1f}, \Theta_z^c)$ and keeps the secondary encoder $\Theta_z^{2f}$ frozen. The local loss is then defined as:

$$\ell_i(\theta_i^{1f}, \theta_i^c) = \mathcal{L}(Y_i, \psi(\phi^{(1)}(X_i; \theta_i^{1f}), \phi^{(2)}(X_i; \Theta_{z(i)}^{2f}); \theta_i^c)) \tag{16}$$

Using the client loss defined in Eq. 16, each client performs SGD training to update $(\theta_i^{1f}, \theta_i^c)$ as:

$$(\theta_i^{1f}, \theta_i^c) \leftarrow (\theta_i^{1f}, \theta_i^c) - \eta\, \nabla_{(\theta_i^{1f}, \theta_i^c)}\, \ell_i(\theta_i^{1f}, \theta_i^c),\ \forall i \in \mathbb{C}_z \tag{17}$$

FEDDAG weighted aggregates the local primary encoder and classifier updates $(\theta_i^{1f} - \Theta_z^{1f})$ and $(\theta_i^c - \Theta_z^c)$ from each client $i \in \mathbb{C}_z$ to update $(\Theta_z^{1f}, \Theta_z^c)$ as:

$$\Theta_z^{1f} \leftarrow \Theta_z^{1f} + \sum_{i \in \mathbb{C}_z} \frac{|D_i|}{\sum_{k \in \mathbb{C}_z} |D_k|}(\theta_i^{1f} - \Theta_z^{1f}), \quad \Theta_z^c \leftarrow \Theta_z^c + \sum_{i \in \mathbb{C}_z} \frac{|D_i|}{\sum_{k \in \mathbb{C}_z} |D_k|}(\theta_i^c - \Theta_z^c) \tag{18}$$

**(ii) Secondary encoder training.** Given the CC-Graph $H$, an edge $j \to z$ means that learner cluster $\mathbb{C}_j$ asks source cluster $\mathbb{C}_z$ to refine its secondary encoder $\Theta_j^{2f}$ using $\mathbb{C}_z$'s data. To achieve this, all learner clusters $\{\mathbb{C}_j : H(j,z) = 1\}$ of $\mathbb{C}_z$ first aggregate their current secondary encoders into a single combined encoder and send this combined encoder to $\mathbb{C}_z$. Clients in $\mathbb{C}_z$ then jointly train this received secondary encoder on their local data (with the primary encoder and classifier frozen), and the resulting gradients are aggregated and sent back to the learner clusters so that each of them can update its own secondary encoder. The process is described in detail below.

For each source cluster $\mathbb{C}_z$, we optimize the secondary encoders $\{\Theta_j^{2f}\}$ of the clusters that seek to learn from $\mathbb{C}_z$. First, given the *CC-Graph $H$*, FEDDAG first aggregate the secondary encoders of all learner clusters into a single combined encoder: $\Theta_z^{2f'} = \sum_{j : H(j,z)=1} \Theta_j^{2f}$ and sends it to $\mathbb{C}_z$. Then, each client $i \in \mathbb{C}_z$ initializes its local instance of the secondary encoder with the received encoder

as $\theta_i^{2f'} \leftarrow \Theta_z^{2f'}$, while keeping the primary $\Theta_z^{1f}$ and the classifier $\Theta_z^c$ fixed; and then minimizes the following loss:

$$\ell_i'(\theta_i^{2f'}) = \mathcal{L}\big(Y_i, \psi(\phi^{(1)}(X_i; \Theta_z^{1f}), \phi^{(2)}(X_i; \theta_i^{2f'}); \Theta_z^c)\big) \tag{19}$$

Using this loss, each client performs SGD to update its local secondary-encoder parameters $\theta_i^{2f'}$ as:

$$\theta_i^{2f'} \leftarrow \theta_i^{2f'} - \eta \, \nabla_{\theta_i^{2f'}} \, \ell_i'(\theta_i^{2f'}) \tag{20}$$

FEDDAG then weighted aggregates the local gradients $(\theta_i^{2f'} - \Theta_z^{2f'})$ for secondary encoder from each client $i \in \mathbb{C}_z$ and broadcasts the aggregated gradient back to the learner clusters. FEDDAG then updates the secondary encoder of each learner cluster $\mathbb{C}_j$ (where $H(j, z) = 1$) using the aggregated gradient as follows:

$$\Theta_j^{2f} \leftarrow \Theta_j^{2f} + \sum_{i \in \mathbb{C}_z} \frac{|D_i|}{\sum_{k \in \mathbb{C}_z} |D_k|} (\theta_i^{2f'} - \Theta_z^{2f'}) \tag{21}$$

**Unifying Primary and Secondary Training.** Since the primary and secondary encoder updates are independent (Eq. 17, 20), they can be trained in parallel. However, because the primary $\Theta_z^{1f}$ and secondary $\Theta_z^{2f}$ encoders are intended to capture complementary information, initializing them both randomly may lead to redundant features. To avoid this, we ensure the primary encoder is partially converged before joint training starts. Specifically, during gradient-based similarity computation in §3.1, each client $i$ trains a local model to partial convergence. We reuse the resulting feature extractors $\theta_i^{0f}$ to initialize the global primary encoder $\Theta_z^{1f}$, thereby avoiding extra training rounds:

$$\Theta_z^{1f} = \sum_{i \in \mathbb{C}_z} \frac{|D_i|}{\sum_{k \in \mathbb{C}_z} |D_k|} \theta_i^{0f}, \quad \forall z \in Z, \tag{22}$$

**FEDDAG structure summary.** During the initial rounds, FEDDAG determines the optimal clustering configuration (see Algorithm 1, Lines 1–14). Once the clustering is established, FEDDAG parallelly executes two phases: a primary training phase and a secondary global feature-sharing phase (Algorithm 1, Lines 15–23). Additional mechanisms for incorporating new clients and adapting to distribution shifts without interrupting training are provided in Appendix A.

## 5 EXPERIMENTS

This section experimentally evaluates FEDDAG, compares it against existing works, and investigates: (i) FEDDAG accuracy, (ii) Finding optimal clustering, (iii) Ablation studies, (iv) During evaluation, we report two variants of our method: FEDDAG*, which is restricted to the approach in §3—combining data and gradient information to form clusters and then training a standard clustered FL model (single encoder and classifier) without global representation sharing—and FEDDAG, which is the full algorithm that additionally incorporates dual-encoder inter-cluster sharing described in §4.

Table 2: Exp 5: Performance comparison for concept shift across datasets.

| Algorithm | CIFAR-10 | FMNIST | SVHN |
|---|---|---|---|
| FedAvg | 42.87±0.36 | 42.68±0.49 | 37.93±0.39 |
| FedBR | 62.41±0.28 | 82.81±0.17 | 80.12±0.22 |
| FedSoft | 64.34±0.38 | 75.89±0.15 | 76.35±0.40 |
| PACFL | 59.82±0.22 | 78.42±0.35 | 78.82±0.12 |
| CFL | 61.48±0.15 | 82.73±0.23 | 79.15±0.36 |
| CFL-GP | 66.74±0.28 | 84.71±0.13 | 82.38±0.13 |
| FedGWC | 65.91±0.19 | 83.85±0.21 | 81.63±0.28 |
| FedRC | 65.48±0.33 | 79.87±0.14 | 77.86±0.29 |
| IFCA | 64.58±0.39 | 84.67±0.21 | 81.56±0.14 |
| FEDDAG* | 67.79±0.27 | 86.03±0.21 | 83.73±0.19 |
| **FEDDAG** | **69.90±0.20** | **88.93±0.13** | **85.34±0.21** |

**Baselines.** We compare FEDDAG against SOTA methods: (i) single-model FL: FedAvg (McMahan et al., 2017), FedProx (Li et al., 2020), (ii) personalized FL methods: PerFedAvg (Fallah et al., 2020), (iii) non-clustered non-IID FL: FedMix (Yoon et al., 2021), FedBR (Guo et al., 2023), (iv) clustered FL — data-based: PACFL (Vahidian et al., 2023), (v) clustered FL — gradient-based: IFCA (Ghosh et al., 2020), CFL (Sattler et al., 2020), FedSoft (Ruan & Joe-Wong, 2022), FedRC (Guo et al., 2024), FedGWC (Licciardi et al., 2025), CFL-GP (Kim et al., 2024a).

Table 3: Exp 3: Ablation study of cross-cluster representation sharing under 20% label skew (Dirichlet $\alpha' = 0.25$), comparing FEDDAG, FEDDAG$^\dagger$ (dual encoder w/o GRS), and FEDDAG* (single encoder).

| Algorithm | CIFAR-10 | FMNIST | SVHN | CIFAR-100 |
|---|---|---|---|---|
| FEDDAG$^\dagger$ | 88.79±0.20 | 92.61±0.31 | 91.95±0.25 | 70.28±0.38 |
| FEDDAG* | 88.67±0.18 | 92.75±0.22 | 91.87±0.26 | 70.37±0.33 |
| **FEDDAG** | **90.76±0.12** | **93.82±0.20** | **93.91±0.23** | **72.84±0.30** |

**Experimental Setup.** We consider 100 clients, with 20% randomly selected per round. Unless stated otherwise, all experiments run for 200 rounds with each selected client performing 10 local

Table 4: Exp 1: Performance comparison for Data Distribution I with a high degree of quantity shift (Dirichlet $\alpha' = 0.25$)

| | 20% Label Skew | | | | 30% Label Skew | | | |
|---|---|---|---|---|---|---|---|---|
| **Algorithm** | **CIFAR-10** | **FMNIST** | **SVHN** | **CIFAR-100** | **CIFAR-10** | **FMNIST** | **SVHN** | **CIFAR-100** |
| FedAvg | $42.02 \pm 1.17$ | $53.11 \pm 0.31$ | $69.79 \pm 0.51$ | $47.16 \pm 0.91$ | $54.24 \pm 0.08$ | $72.86 \pm 0.40$ | $64.15 \pm 0.64$ | $50.99 \pm 1.35$ |
| FedProx | $43.98 \pm 0.17$ | $53.61 \pm 0.20$ | $74.75 \pm 0.27$ | $50.56 \pm 0.70$ | $54.99 \pm 0.20$ | $68.22 \pm 0.16$ | $64.80 \pm 0.25$ | $48.66 \pm 0.80$ |
| PerFedAvg | $81.09 \pm 0.35$ | $86.51 \pm 0.19$ | $89.20 \pm 0.05$ | $65.59 \pm 0.02$ | $77.45 \pm 0.24$ | $89.77 \pm 0.15$ | $88.23 \pm 0.31$ | $57.38 \pm 0.10$ |
| FedMix | $77.94 \pm 0.26$ | $83.55 \pm 0.31$ | $83.12 \pm 0.29$ | $60.33 \pm 0.24$ | $76.90 \pm 0.33$ | $81.96 \pm 0.27$ | $82.21 \pm 0.34$ | $53.55 \pm 0.30$ |
| FedBR | $81.62 \pm 0.28$ | $85.32 \pm 0.23$ | $84.05 \pm 0.30$ | $61.61 \pm 0.37$ | $81.48 \pm 0.37$ | $84.12 \pm 0.25$ | $84.78 \pm 0.38$ | $56.32 \pm 0.33$ |
| FedSoft | $76.44 \pm 0.18$ | $84.58 \pm 0.14$ | $83.75 \pm 0.33$ | $62.54 \pm 0.41$ | $72.48 \pm 0.17$ | $85.15 \pm 0.17$ | $82.43 \pm 0.40$ | $55.24 \pm 0.43$ |
| PACFL | $86.93 \pm 0.40$ | $91.90 \pm 0.47$ | $89.88 \pm 0.25$ | $66.11 \pm 0.29$ | $84.66 \pm 0.29$ | $91.96 \pm 0.25$ | $90.48 \pm 0.23$ | $58.30 \pm 0.56$ |
| CFL | $68.67 \pm 0.76$ | $81.90 \pm 0.10$ | $79.83 \pm 0.38$ | $57.38 \pm 0.95$ | $67.57 \pm 0.69$ | $80.64 \pm 0.21$ | $75.21 \pm 0.09$ | $49.63 \pm 1.29$ |
| CFL-GP | $85.25 \pm 0.17$ | $89.13 \pm 0.35$ | $87.83 \pm 0.22$ | $67.89 \pm 0.20$ | $83.98 \pm 0.28$ | $91.14 \pm 0.14$ | $90.01 \pm 0.11$ | $59.71 \pm 0.76$ |
| FedGWC | $85.97 \pm 0.13$ | $91.02 \pm 0.17$ | $89.35 \pm 0.10$ | $69.19 \pm 0.48$ | $83.58 \pm 0.21$ | $91.45 \pm 0.12$ | $88.94 \pm 0.15$ | $56.52 \pm 0.40$ |
| FedRC | $75.12 \pm 0.28$ | $88.32 \pm 0.23$ | $88.05 \pm 0.30$ | $63.25 \pm 0.37$ | $76.48 \pm 0.37$ | $88.12 \pm 0.25$ | $85.78 \pm 0.38$ | $54.32 \pm 0.33$ |
| IFCA | $86.64 \pm 0.13$ | $90.93 \pm 0.17$ | $89.51 \pm 0.10$ | $69.08 \pm 0.48$ | $83.45 \pm 0.37$ | $91.50 \pm 0.11$ | $88.81 \pm 0.09$ | $56.33 \pm 0.40$ |
| FEDDAG* | $88.67 \pm 0.18$ | $92.75 \pm 0.22$ | $91.87 \pm 0.26$ | $70.37 \pm 0.33$ | $86.95 \pm 0.21$ | $92.18 \pm 0.15$ | $90.97 \pm 0.13$ | $60.84 \pm 0.65$ |
| **FEDDAG** | $\mathbf{90.76 \pm 0.12}$ | $\mathbf{93.82 \pm 0.20}$ | $\mathbf{93.91 \pm 0.23}$ | $\mathbf{72.84 \pm 0.30}$ | $\mathbf{89.87 \pm 0.19}$ | $\mathbf{92.72 \pm 0.13}$ | $\mathbf{92.65 \pm 0.11}$ | $\mathbf{63.21 \pm 0.60}$ |

epochs (batch size 10, SGD). The principal vector $U_c^i$ transmitted per class is roughly 1% the size of $|D_{i,c}|$. For gradient similarity $\mathcal{G}_{i,j}$, each client trains locally for $t_g=2$ rounds. To construct the *CC-Graph*, we select the top-$k=2$ source clusters.

**Datasets.** We use four popular datasets for the image classification task in FL setting, i.e., CIFAR-10 (Krizhevsky et al., 2009), FMNIST (Xiao et al., 2017), SVHN (Netzer et al., 2011), and CIFAR-100 (Krizhevsky et al., 2009).

**Non-IID Data.** We use multiple data distributions to simulate traditional and complex data skews:

• **Data Distribution I:** *This distribution evaluates* FEDDAG *under combined label skew and quantity shift.* To simulate label skew, we randomly select $\rho\%$ of labels and assign them to random client groups, repeating the process until all clients are assigned—similar to PACFL. For quantity shift, we allocate samples of the assigned labels using the Dirichlet factor (Ng et al., 2011). A real-world example is predictive text input, where users may discuss similar topics, but word distributions vary due to individual preferences and typing habits.

• **Data Distribution II:** *This distribution evaluates* FEDDAG *under concept shift.* Following prior work (Jothimurugesan et al., 2023; Guo et al., 2024), we simulate concept shift by modifying the labels of a subset of clients. For example, label $y$ is changed to $(C-y)$ or $(y+1)\%C$, where $C$ is the total number of classes. We perform three such transformations to simulate three distinct concepts. Similar modifications are applied to the test set.

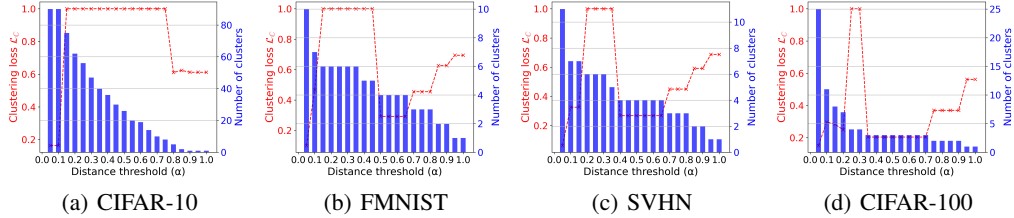

(a) CIFAR-10      (b) FMNIST      (c) SVHN      (d) CIFAR-100

Figure 2: Exp 2: Clustering score vs cluster $\alpha$ and number of clusters for finding optimal clustering.

• **Data Distribution III:** *This distribution evaluates* FEDDAG *under a different form of label skew.* We adopt the Latent Dirichlet Allocation (LDA) method from Hsu et al. (2019), using Dirichlet concentration factors $\alpha' = 0.25$ and $\alpha' = 1.0$.

> *Additional experiments (e.g., performance evaluation, communication rounds) on the above and new (feature skew) distributions, hyperparameter tuning, implementation details, ablation studies are provided in Appendix B. Algorithm theoretical issues, such as convergence, complexity, and privacy analysis; distribution and client shifts are discussed in Appendix A.*

**Experiments on Data Distribution I**

**Exp 1: Performance evaluation.** We consider class skew $\rho = 20\%$ and 30%, with the Dirichlet concentration parameter $\alpha'$ set to *1* for low and *0.25* for high quantity shift. Table 4 shows the results for $\alpha' = 0.25$, while the results for $\alpha' = 1$ are included in Appendix B.3. We observe that

Table 5: Exp 5: Performance comparison under LDA skew ($\alpha' = 0.25$ and $\alpha' = 1.0$).

| Algorithm | $\alpha' = 0.25$ | | | $\alpha' = 1.0$ | | |
|---|---|---|---|---|---|---|
| | CIFAR-10 | FMNIST | SVHN | CIFAR-10 | FMNIST | SVHN |
| FedAvg | $66.48 \pm 0.21$ | $47.26 \pm 0.28$ | $46.13 \pm 0.48$ | $41.78 \pm 0.73$ | $85.48 \pm 0.36$ | $81.89 \pm 0.31$ |
| FedSoft | $71.08 \pm 0.26$ | $83.75 \pm 0.26$ | $85.67 \pm 0.19$ | $73.83 \pm 0.42$ | $87.85 \pm 0.31$ | $85.92 \pm 0.13$ |
| PACFL | $73.91 \pm 0.43$ | $85.93 \pm 0.12$ | $87.23 \pm 0.20$ | $80.52 \pm 0.15$ | $93.31 \pm 0.28$ | $92.17 \pm 0.23$ |
| CFL | $67.46 \pm 0.12$ | $85.18 \pm 0.17$ | $85.19 \pm 0.25$ | $78.94 \pm 0.18$ | $83.16 \pm 0.26$ | $82.75 \pm 0.28$ |
| CFL-GP | $73.84 \pm 0.28$ | $86.43 \pm 0.14$ | $88.04 \pm 0.19$ | $83.57 \pm 0.15$ | $92.21 \pm 0.23$ | $91.67 \pm 0.19$ |
| FedRC | $70.19 \pm 0.42$ | $85.24 \pm 0.22$ | $87.91 \pm 0.26$ | $81.76 \pm 0.16$ | $88.27 \pm 0.22$ | $86.29 \pm 0.42$ |
| IFCA | $74.43 \pm 0.32$ | $87.53 \pm 0.21$ | $88.81 \pm 0.13$ | $82.27 \pm 0.19$ | $92.79 \pm 0.33$ | $92.12 \pm 0.15$ |
| FEDDAG* | $75.52 \pm 0.27$ | $89.65 \pm 0.16$ | $91.27 \pm 0.22$ | $85.03 \pm 0.21$ | $93.95 \pm 0.20$ | $93.08 \pm 0.18$ |
| **FEDDAG** | $\mathbf{77.84 \pm 0.23}$ | $\mathbf{91.88 \pm 0.10}$ | $\mathbf{93.17 \pm 0.18}$ | $\mathbf{87.62 \pm 0.14}$ | $\mathbf{94.68 \pm 0.13}$ | $\mathbf{94.15 \pm 0.11}$ |

single global FL baselines (e.g., FedAvg, FedProx) perform poorly under heterogeneity due to model drift (Zhao et al., 2018), while clustered FL methods yield stronger performance. Both variants of FEDDAG outperform state-of-the-art baselines—including data-based methods (e.g., PACFL) and gradient-based methods (e.g., IFCA, FedGWC). The lighter variant, FEDDAG*, achieves strong performance by combining data and gradient information to yield improved clustering. The full FEDDAG further enhances accuracy by enabling complementary representation sharing across clusters, allowing them to learn richer feature spaces.

**Exp 2: Finding Optimal Cluster Formation.** The server iterates over the clustering threshold $\alpha$ in Agglomerative HC at regular intervals (e.g., 0.05) to generate candidate clusterings. For each, the clustering loss $\mathcal{L}_{\{\mathbb{C}_1,...,\mathbb{C}_Z\}}$ (see §3.4) is computed. In Figure 2, the x-axis shows $\alpha$; the red curve indicates loss, and blue bars denote the number of clusters. Unlike traditional metrics (e.g., inertia) where loss decreases with more clusters, we observe abrupt increases in loss even as the number of clusters decreases for certain $\alpha$ values. This is due to FEDDAG's federated-aware clustering loss penalizing over-splitting into small clusters. The optimal $\alpha$ is selected as the point with low clustering loss and a relatively small number of clusters (e.g., for Figure 2(b) $\alpha^* = 0.65$).

**Exp 3: Ablation Studies.** We examine whether accuracy gains from inter-cluster global representation sharing (GRS) via the dual-encoder architecture (see §4) arise from genuine feature enrichment or simply from increased model parameters. To isolate this effect, we implement a dual-encoder variant with GRS disabled: during secondary-encoder training, instead of receiving representations from other clusters, each client trains its secondary encoder only on its own data and aggregates within its cluster. We denote this variant FEDDAG[†]; it is distinct from FEDDAG, which uses a single encoder. As shown in Table 3, full FEDDAG (with GRS) achieves the highest accuracy, while FEDDAG[†] performs comparably to FEDDAG, confirming that the gains of FEDDAG stem from cross-cluster representation sharing rather than model size alone.

### Experiment on Data Distribution II

**Exp 4: Performance under concept shift.** Table 2 compares the performance of SOTA algorithms and FEDDAG on different datasets under concept shift and shows that FEDDAG achieves higher accuracy than the baselines. This improvement stems from FEDDAG's class-wise comparison mechanism, which provides more accurate similarity estimation under concept shift than existing methods.

### Experiment on Data Distribution III

**Exp 5: Performance under varying LDA skew.** Table 5 shows accuracy under LDA-based skew with $\alpha' = 0.25$ and $\alpha' = 1.0$. FEDDAG consistently outperforms SOTA methods by leveraging cross-cluster feature sharing and integrating data and gradient information for clustering, leading to robust performance under LDA-based partition.

## 6  CONCLUSION

We develop a novel algorithm, FEDDAG, that addresses the limitations of existing clustered FL techniques and effectively tackles data heterogeneity challenges in FL by developing a novel method that combines both data and gradient information to cluster clients more effectively. Furthermore, FEDDAG utilizes representation sharing across clusters and incorporates an efficient mechanism to automatically determine the optimal number of clusters. Experiments on various heterogeneous data distributions demonstrate that FEDDAG outperforms existing approaches in terms of accuracy.

ACKNOWLEDGMENTS

Author M. K. was supported in part by NSF awards DMS-2204795, OAC-2115094, CNS-2331424, ITE-2452833, ARL/Army Research Office awards W911NF-24-1-0202 and W911NF-24-2-0114, and Virginia Commonwealth Cyber Initiative grants. Author V. O. was supported in part by the National Science Foundation (NSF) under Grant DGE 2043104. Author S. S. was supported by NSF Grant 2245374. We thank NJIT HPC facility for providing GPU support for our experiments.

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

# A  TECHNICAL DISCUSSION AND ANALYSIS

This section presents a detailed discussion and breakdown of FEDDAG, covering key design elements, communication and privacy considerations, adaptability to new clients and shifting distributions, and practical implementation details.

## A.1  ADDITIONAL RELATED WORK

**Data Augmentation** based FL techniques propose sharing a small global dataset among clients and combining it with their local datasets to mitigate heterogeneity (Tuor et al., 2021; Xin et al., 2020). Approaches such as Xin et al. (2020); Yoon et al. (2021); Guo et al. (2023) employ generative adversarial networks (GANs) or averaged local data (Random Sample Mean) to create privacy-preserving pseudo-data that is used to reduce bias in client models. Astraea (Duan et al., 2019) constructs a globally balanced data distribution by performing local augmentation across participating clients. VHL (Tang et al., 2022) creates virtual samples from noise shared across clients to regularize training by aligning local feature representations with these virtual data.

**Meta-learning** approaches include personalized FL (Arivazhagan et al., 2019; Liang et al., 2020; Li et al., 2024b) and model regularization methods (Li et al., 2021a; T Dinh et al., 2020; Karimireddy et al., 2020). Per-FedAvg (Fallah et al., 2020) is a personalized variant of FedAvg based on the Model-Agnostic Meta-Learning (MAML) framework. FedPer (Arivazhagan et al., 2019) and FedRep (Collins et al., 2021) split the backbone into a feature extractor and a head to share feature information, while FedRoD (Chen & Chao, 2021) maintains a shared feature extractor and two heads. FedSimSup (Liu et al., 2025) uses a local supervisor and data-similarity–weighted inter-learning model to better align global knowledge with heterogeneous local data. Our approach, FEDDAG, maintains global–personal decoupling via a dual-encoder architecture, where the global parameters are updated only by a selected subset of clients instead of being influenced by all clients.

**Client Weighting and Selection** Another line of work tackles non-IID FL by selecting participants or assigning aggregation weights based on client properties such as data quality, label distribution, model alignment, and historical contribution. Representative strategies for assigning weights or selecting clients include linear programming (Pramanik et al., 2023), submodular optimization, reinforcement learning, bandit-based optimization, and adaptive aggregation-weight optimization (Balakrishnan et al., 2022; Bhope et al., 2023; Xu et al., 2024a; He et al., 2025; Shi et al., 2025).

## A.2  PRELIMINARIES

**Principal Angles Between Two Subspaces.** Consider two subspaces, $\mathcal{V} = \text{span}\{\mathbf{v}_1, \ldots, \mathbf{v}_p\}$ and $\mathcal{X} = \text{span}\{\mathbf{x}_1, \ldots, \mathbf{x}_q\}$, where $\mathcal{V}$ and $\mathcal{X}$ are $p$-dimensional and $q$-dimensional subspaces of $\mathbb{R}^n$, respectively. The sets $\{\mathbf{v}_1, \ldots, \mathbf{v}_p\}$ and $\{\mathbf{x}_1, \ldots, \mathbf{x}_q\}$ are orthonormal, with $1 \leq p \leq q$. A sequence of $p$ principal angles, $0 \leq \Phi_1 \leq \Phi_2 \leq \cdots \leq \Phi_p \leq \frac{\pi}{2}$, is defined to measure the similarity between the subspaces. These angles are calculated as:

$$\Phi(\mathcal{V}, \mathcal{X}) = \min_{\mathbf{v} \in \mathcal{V}, \mathbf{x} \in \mathcal{X}} \cos^{-1}\left(\frac{|\mathbf{v}^T \mathbf{x}|}{\|\mathbf{v}\|\|\mathbf{x}\|}\right) \tag{23}$$

where $\|\cdot\|$ is the norm. The smallest of these angles is $\Phi_1(\mathbf{v}_1, \mathbf{x}_1)$, with the vectors $\mathbf{v}_1$ and $\mathbf{x}_1$ as the corresponding principal vectors. The principal angle distance serves as a metric to quantify the separation between subspaces Jain et al. (2013).

**Agglomerative hierarchical clustering (HC).** (Day & Edelsbrunner, 1984) is a popular method in machine learning for grouping similar objects based on an adjacency (proximity) matrix. We found HC to be the best fit for FEDDAG. We also experimented with other clustering algorithms, e.g., K-means (Pramanik et al., 2020) and graph clustering (Schaeffer, 2007), but we observed that the clustering algorithm does not make much difference in cluster formation. HC begins by treating each data point as its own cluster. During each iteration, HC identifies two clusters that are most similar and merges them. The criterion for selecting which clusters to merge depends on a linkage method; e.g., in *single linkage*, the $L_2$ (Euclidean) distance between two clusters is defined as the smallest distance between any pair of points from the two clusters. As a merging criterion, FEDDAG defines a ***clustering threshold*** $\alpha \in (0, 1]$, such that any two clusters with a distance less than $\alpha$ are merged.; e.g., $\alpha=1$ results in all clients being grouped into a single cluster.

## A.3 FEDDAG TECHNICAL COMPONENTS

An illustration of the FEDDAG algorithm is shown in Figure 1. The algorithm for class-wise weighted data-based similarity computation is shown in Algorithm 2. And, the algorithm for combining both data and gradient information to improve clustering is shown in Algorithm 3.

## A.4 CONVERGENCE ANALYSIS

Following Pillutla et al. (2022) that works on partial model personalization, we consider the shared–personalized objective:

$$\min_{u, V} F(u, V) := \frac{1}{n} \sum_{i=1}^{n} F_i(u, v_i), \tag{24}$$

where $u$ denotes shared parameters and $V = \{v_i\}_{i=1}^{n}$ personalized parameters. In our dual-encoder model (Eq. equation 15), for each cluster $z$ we map the *secondary encoder* as the *shared* block and the *primary encoder* (optionally together with the classifier) as the *personalized* block:

$$u_z \longmapsto \Theta_z^{2f} \quad \text{(shared: secondary encoder)},$$

$$V_z \longmapsto (\Theta_z^{1f}, \Theta_z^{c}) \quad \text{(personalized: primary encoder + classifier)}.$$

Given a fixed clustering $\{\mathbb{C}_z\}_{z=1}^{Z}$ (one-shot data and gradient combined similarity; see §3), the cluster-level empirical risk can be written in the shared–personalized form of Pillutla et al. (2022):

$$\min_{\{u_z, V_z\}_{z=1}^{Z}} F(\{u_z, V_z\}) = \sum_{z=1}^{Z} \sum_{i \in \mathbb{C}_z} \frac{|D_i|}{\sum_{k \in \mathbb{C}_z} |D_k|} F_i(u_z, V_z),$$

$$F_i(u_z, V_z) = \mathcal{L}\Big(Y_i, \psi\big(\phi^{(1)}(X_i; \Theta_z^{1f}), \phi^{(2)}(X_i; u_z); \Theta_z^{c}\big)\Big).$$

Thus, for each cluster $z$, Pillutla et al. (2022)'s analysis applies to the pair $(u_z, V_z)$, and the full objective is a weighted sum over clusters. So, based on this, we will define notations, assumptions, and the convergence analysis below:

**Block notation and participation model.** For each cluster $z \in \{1, \ldots, Z\}$ in the fixed partition $\{\mathbb{C}_z\}_{z=1}^{Z}$, we decompose the parameters as

$$u_z := \Theta_z^{2f} \quad \text{(cluster–global / secondary encoder)}, \tag{25}$$

$$V_z := (\Theta_z^{1f}, \Theta_z^{c}) \quad \text{(cluster–personal: primary encoder + classifier)}. \tag{26}$$

Let $m$ be the total number of clients and $m_z := |\mathbb{C}_z|$ the number of clients in cluster $z$; define the cluster weights

$$\pi_z := \frac{m_z}{m}, \qquad \sum_{z=1}^{Z} \pi_z = 1. \tag{27}$$

In each communication round, cluster $z$ samples $s_z$ clients (without replacement) and runs $E$ local steps. The average per-round participation fraction is

$$\bar{q} := \sum_{z=1}^{Z} \pi_z \frac{s_z}{m_z} \in (0, 1]. \tag{28}$$

**Loss and per-client objective.** For client $i \in \mathbb{C}_z$ with data $D_i = (X_i, Y_i)$, define

$$F_i(u_z, V_z) := \mathcal{L}\Big(Y_i, \psi\big(\phi^{(1)}(X_i; \Theta_z^{1f}), \phi^{(2)}(X_i; u_z); \Theta_z^{c}\big)\Big), \tag{29}$$

and the cluster-weighted empirical risk

$$F\big(\{u_z, V_z\}_{z=1}^{Z}\big) := \sum_{z=1}^{Z} \sum_{i \in \mathbb{C}_z} \frac{|D_i|}{\sum_{k \in \mathbb{C}_z} |D_k|} F_i(u_z, V_z). \tag{30}$$

**Scope of the analysis.** The full FEDDAG algorithm includes three dynamic components: (i) an initial clustering phase that combines data- and gradient-based similarity (Algorithm 1, Lines 1–14), (ii) a mechanism to attach newcomers to existing clusters (Appendix, Algorithm 4), and (iii) an optional re-clustering step under significant distribution shift. In this subsection, we analyze the *stationary training regime between such events*: we condition on a fixed client set and a fixed partition $\{\mathbb{C}_z\}_{z=1}^Z$ and study the convergence behavior of the shared–personalized objective Eq. 30 under this partition. Newcomers or mild distribution changes that do not trigger re-clustering can be viewed as small perturbations of the variance and heterogeneity constants in our bound, whereas a re-clustering event corresponds to switching to a new objective with a new partition and new constants. The theorem below should therefore be interpreted as a *per-phase* guarantee for any interval between two re-clustering events.

**Assumptions used in the theorem.** We state the standard conditions in our block notation; expectations are w.r.t. the algorithm's sampling and stochasticity.

**Assumption A.1** (Smoothness). *Each client loss in equation 29 is $L$-smooth in $(u_z, V_z)$. For all $(u_z, V_z)$ and $(u'_z, V'_z)$,*

$$\left\| \nabla_{(u_z, V_z)} F_i(u_z, V_z) - \nabla_{(u_z, V_z)} F_i(u'_z, V'_z) \right\| \leq L \left\| (u_z, V_z) - (u'_z, V'_z) \right\|. \tag{31}$$

*Equivalently, $F_i$ is $L$-smooth in each sub-block $\Theta_z^{1f}$, $\Theta_z^{2f}$, and $\Theta_z^c$.*

**Assumption A.2** (Unbiased stochastic gradients with bounded variance). *For any sampled client $i \in \mathbb{C}_z$,*

$$\mathbb{E}\left[ \tilde{\nabla}_{u_z} F \right] = \nabla_{u_z} F, \qquad \mathbb{E}\left[ \| \tilde{\nabla}_{u_z} F - \nabla_{u_z} F \|^2 \right] \leq \sigma_{u,z}^2, \tag{32}$$

$$\mathbb{E}\left[ \tilde{\nabla}_{V_z} F \right] = \nabla_{V_z} F, \qquad \mathbb{E}\left[ \| \tilde{\nabla}_{V_z} F - \nabla_{V_z} F \|^2 \right] \leq \sigma_{V,z}^2, \tag{33}$$

*where $\nabla_{V_z} F := (\nabla_{\Theta_z^{1f}} F, \nabla_{\Theta_z^c} F)$. Define the cluster-weighted variances*

$$\bar{\sigma}_u^2 := \sum_{z=1}^Z \pi_z \sigma_{u,z}^2, \qquad \bar{\sigma}_V^2 := \sum_{z=1}^Z \pi_z \sigma_{V,z}^2. \tag{34}$$

**Assumption A.3** (Gradient diversity / heterogeneity). *Let $F_z(u_z, V_z) := \frac{1}{\sum_{k \in \mathbb{C}_z} |D_k|} \sum_{i \in \mathbb{C}_z} |D_i| F_i(u_z, V_z)$ be the average loss in cluster $z$. There exist finite constants $\delta_{\text{in}}^2 \geq 0$ and $\delta_{\text{out}}^2 \geq 0$ such that*

$$\sum_{z=1}^Z \pi_z \left\| \nabla_{u_z} F_z - \nabla_{u_z} F \right\|^2 \leq \delta_{\text{in}}^2, \qquad \sum_{z=1}^Z \pi_z \left\| \nabla_{V_z} F_z - \nabla_{V_z} F \right\|^2 \leq \delta_{\text{in}}^2, \tag{35}$$

*and the cross-cluster mismatch (relevant to the sharing step) is bounded by $\delta_{\text{out}}^2$.*

**Assumption A.4** (Phase-wise stable clustering). *For the training phase under consideration, the partition $\{\mathbb{C}_z\}_{z=1}^Z$ obtained at initialization ($t=0$ of this phase) remains fixed for the analysis horizon $t = 1, \ldots, T$: no clients are reassigned, and clusters do not split or merge within this phase. Potential re-clustering events (triggered by significant distribution shift) are modeled as the start of a new phase with its own objective and constants and are therefore outside the scope of the current per-phase guarantee.*

**Assumption A.5** (Cross-cluster sharing noise). *The cross-cluster representation sharing (via the CC-Graph) is either deterministic (no additional noise), or it introduces an additive variance bounded by $\sigma_{\text{share}}^2$ in the updates of the $u$-blocks.*

**Initial suboptimality.** We denote the initial gap by

$$\Delta_\ell := F\left( \{u_z^0, V_z^0\}_{z=1}^Z \right) - F^\star, \tag{36}$$

where $F^\star$ is the optimal value of equation 30.

**Theorem A.1** (Convergence of FEDDAG (per-cluster globals, dual encoders)). *Let the assumptions above hold. Choose learning rates $\eta = \tau/(LE)$ and $\eta_{\text{share}} = \Theta(1/L)$, for a constant $\tau$ depending on $L$, the variance terms, heterogeneity, and participation. Then, ignoring absolute constants and provided clustering is stable,*

$$\frac{1}{T} \sum_{t=1}^T \left[ \frac{1}{L} \sum_{z=1}^Z \mathbb{E} \| \nabla_{u_z} F \|^2 + \frac{1}{mL} \sum_{i=1}^m \mathbb{E} \| \nabla_{V_{c(i)}} F \|^2 \right] \leq \frac{(\Delta_\ell \, \sigma_{\text{sim},1}^2)^{1/2}}{T^{1/2}} + \frac{(\Delta_\ell^2 \, \sigma_{\text{sim},2}^2)^{1/3}}{T^{2/3}} + \mathcal{O}\left( \frac{1}{T} \right),$$
$$\tag{37}$$

*where the effective variance terms are*

$$\sigma_{\text{sim},1}^2 \;=\; \frac{2}{L}\left(\delta_{\text{in}}^2 \sum_{z=1}^{Z} \pi_z\left(1 - \frac{s_z}{m_z}\right) \;+\; \frac{\bar{\sigma}_u^2}{L} \;+\; \sum_{z=1}^{Z} \pi_z \frac{s_z}{m_z}\sigma_{V,z}^2 \;+\; \sigma_{\text{share}}^2\right), \qquad (38)$$

$$\sigma_{\text{sim},2}^2 \;=\; \frac{2}{L}\left(\delta_{\text{in}}^2 + \delta_{\text{out}}^2 \;+\; \bar{\sigma}_u^2 + \bar{\sigma}_V^2 \;+\; \sigma_{\text{share}}^2\right)\left(1 - \frac{1}{E}\right). \qquad (39)$$

*Remark 1 (Clustering stability).* The bound relies on a fixed partition; oscillations due to re-clustering invalidate the descent decomposition. In practice, stability is supported empirically by (i) one-shot blended (data+gradient) clustering at $t=0$ and (ii) the fact that training is conducted within the fixed clusters thereafter.

*Remark 2 (What differs vs. single-global frameworks).* In FEDDAG, we aggregate *both* the per-cluster global blocks $u_{1:Z}$ (secondary encoders, coupled via the CC-Graph) and the cluster-personal blocks $V_{1:Z}$ (primary encoder + classifier). Consequently, $\sigma_{\text{sim},1}^2$ and $\sigma_{\text{sim},2}^2$ expose: (i) per-cluster sampling $s_z/m_z$ (larger $s_z$ improves the first term), (ii) local steps $E$ (fewer local steps reduce the drift factor $1 - 1/E$), and (iii) cross-cluster sharing noise $\sigma_{\text{share}}^2$ (zero for deterministic Laplacian smoothing; small but positive for stochastic distillation). The asymptotic $T^{-1/2}$ rate is observed once all devices are seen on average at least once; a convenient sufficient condition (up to constants) is

$$T \;\geq\; \frac{\Delta_\ell}{\sigma_{\text{sim},1}^2} \; \max\left\{\frac{(1-\bar{q})\,E}{\bar{q}}, \; 2\right\}, \qquad \bar{q} = \sum_{z=1}^{Z} \pi_z \frac{s_z}{m_z}. \qquad (40)$$

## A.5 COMMUNICATION AND COMPUTATION COMPLEXITY

FEDDAG minimizes communication and computation overhead, aligning with the scalability requirements of federated learning systems. Before dual-encoder joint training begins, each client locally trains for $t_g$ rounds without federation (see Algorithm 1). At the end of this phase, each client uploads: (i) a $k$-sparse gradient $\tilde{\Delta}_i$ of dimension $k \ll |D_i|$, and (ii) class-wise $p$ principal vectors $U_c^i \in \mathbb{R}^{d \times r}$ for $c = 1, \ldots, C$. The number of principal vectors $p$ is kept small (typically 1–2% of the class size). Hence, the combined communication cost of $\tilde{\Delta}_i$ and $U_c^i$ is negligible relative to the size of the model parameter space $|\Theta|$. The computation of principal vectors via SVD incurs a cost of $\mathcal{O}(FN^2)$ per client, assuming a local dataset of $N$ samples and $F$ features with $N > F$.

Once the proximity matrix and clustering are finalized, FEDDAG maintains the same per-round communication cost as FedAvg in terms of transmitting model parameters. However, due to its dual-encoder architecture, it additionally transmits a secondary encoder ($\Theta^{2f}$) alongside the primary encoder ($\Theta^{1f}$), both of equal size. In each training round, selected clients perform two local SGD phases:

- *Primary phase* — standard local update on $(\theta^{1f}, \theta^c)$.

- *Secondary phase* — additional local update on $\theta^{2f'}$, which has the same size as $\theta^{1f}$.

If both phases are executed in the same round, the local computation cost is approximately $2\times$ that of FedAvg. However, the two phases can be alternated when computation is constrained—for example, updating the primary encoder for several rounds (e.g., 5 rounds) followed by a single round updating the secondary encoder. We also perform an experiment evaluating the trade-offs of these alternating schedules, showing how they affect convergence speed; see §B.9.

Since updates to the primary and secondary encoders are independent, the correctness and convergence of the final model are preserved under this alternating schedule.

## A.6 CLUSTERING OVERHEAD OF FEDDAG

While §A.5 analyzes the overall complexity of FEDDAG, here we focus specifically on the clustering component and compare it to other clustered FL approaches. Conceptually, clustering in

FEDDAG can be separated into two parts: (i) a *one-time* warm-up and initial clustering phase and (ii) a re-clustering phase to handle distribution shift.

**(i) Initial clustering phase.** In this part, we discuss the computation and communication overhead of the warm-up and initial clustering phase:

**Computation overhead.** In the initial phase, each client runs $t_g$ rounds of local warm-up epochs. As detailed in Appendix B.2 (Hyperparameter Tuning), a small number of rounds, such as $t_g = 2$, is enough to obtain the client gradient signatures for similarity computation. Another important aspect is that these warm-up updates are not discarded. While the clients simply do not participate in FL aggregation during this short period, once clustering is assigned, training continues directly from the warmed-up model. The same warmed-up parameters are also used to initialize the secondary encoder training during our dual-encoder-based clustered FL, so the warm-up is not an additional or wasted cost.

Another aspect of FEDDAG's initial clustering is computing per-class principal vectors via truncated SVD for a client with $N$ samples and feature dimension $F$, which costs $\mathcal{O}(FN^2)$. This one-time SVD, together with $t_g \leq 2$ warm-up rounds, constitutes the main clustering-side computation overhead in FEDDAG. The SVD cost is incurred only once during initialization and is amortized over all subsequent communication rounds. This computation overhead of FEDDAG is compared with other clustered FL approaches as follows:

- PACFL also performs an SVD/PCA-type step per client to extract principal components, so the clustering-stage complexity is of the same order. In FEDDAG, one addition that can create a slight increase is the $t_g$ rounds of warm-up, but FEDDAG mainly adds the reuse of the $t_g$ warm-up for the model during the clustering phase, rather than treating clustering as a separate, discarded pre-processing step.

- In contrast, iterative clustering methods (IFCA, CFL, CFL-GP, FedSoft, FedRC) re-evaluate clustering at every communication round: at round $t$, the server sends $K$ cluster models to each participating client, and each client must evaluate all $K$ models locally to decide its cluster assignment. Over $T$ rounds, this leads to roughly $KT$ model evaluations per client, whereas FEDDAG pays the SVD + warm-up cost only once and then trains on a fixed cluster assignment (unless a rare re-clustering event is triggered; see below).

**Communication overhead.** After warm-up, each client uploads a $k$-sparse gradient $\tilde{\Delta}_i$ (only 1–2% of coordinates) and a small number of per-class principal vectors $U_c^i$ (about 1–2% of the class data size). The resulting communication is negligible compared to sending the full model parameters $|\Theta|$. Both the sparsified gradient and the principal vectors are communicated only once during the clustering phase, so their cost is also amortized over the entire training run. The communication overhead is compared to other clustered FL approaches as follows:

- PACFL has a similar one-shot communication pattern for principal components during clustering, as it also sends principal vectors. FEDDAG incurs a slightly higher cost because it additionally sends the sparsified gradient, but this is still a single extra message of size 1–2% of the model.

- Iterative clustered FL methods such as IFCA, CFL, CFL-GP, FedRC, and FedSoft must send all $K$ cluster models (each of size $M$) to every participating client in every round, so each client receives on the order of $KM$ parameters per round and returns updates for its chosen cluster. In contrast, FEDDAG only sends 1–2% of the model parameters $M$ and principal vectors whose total size is about 1–2% of the client data.

**(ii) Re-clustering overhead.** We discussed that FEDDAG performs re-clustering under severe distribution shift (Appendix A.8) as an infrequent, deployment-time procedure: it is triggered only after a major shift in the client distribution and is not used in our main experiments (e.g., when the Wasserstein distance changes by more than 20%). When activated, it simply re-runs the same warm-up + principal vector computation and clustering procedure, again without resetting the models, so the cost is similar to the initial clustering phase as above. During the FL training period (apart from the re-clustering mechanism), FEDDAG does not require further clustering, as it trains on the assigned clusters.

To complement the qualitative discussion of computation and communication overhead above, we next provide a simple numerical complexity analysis that quantifies how the clustering cost of one-

time SVD and partial gradient convergence in FEDDAG compares to the per-round clustering cost of iterative clustered FL methods. Secondly, we also provide an alternative approach to compute principal vectors that reduces the overall clustering overhead.

### A.6.1 NUMERICAL ANALYSIS OF CLUSTERING OVERHEAD

The discussion above compares FEDDAG qualitatively to iterative clustered FL methods. We now provide a simple numerical complexity analysis to quantify the relative overhead.

For each client with local sample size $N$ and feature dimension $F$, let $X \in \mathbb{R}^{N \times F}$ denote its feature matrix. A full SVD of $X$ would cost on the order of:

$$\mathcal{O}\big(\min\{NF^2, FN^2\}\big),$$

but FEDDAG only requires the top $p \ll \min(N, F)$ singular vectors per class. Using efficient truncated or randomized SVD, as shown here Halko et al. (2011), the per-client cost reduces to:

$$\mathcal{O}(pNF),$$

which is linear in both $N$ and $F$, and this cost is incurred only once during the warm-up / initial clustering phase.

By contrast, iterative clustered FL methods (e.g., IFCA, CFL, CFL-GP, FedGWC, FedRC, FedSoft) maintain $K$ cluster models and, in each communication round, send all $K$ models to every participating client. To update cluster assignments, each client evaluates all $K$ models on its $N$ local samples. Let $M_{\mathrm{fwd}}$ denote the cost of a single forward pass of the full model on one sample. Then the per-round and total clustering costs of these methods are

$$\text{iterative per-round cost: } \mathcal{O}(KNM_{\mathrm{fwd}}).$$

$$\text{iterative total cost over } T \text{ rounds: } \mathcal{O}(KTNM_{\mathrm{fwd}}).$$

Comparing the clustering cost of FEDDAG and iterative clustered FL methods for a single client, we first account for the components of FEDDAG's clustering overhead. This consists of the one-time truncated SVD used for initial clustering and the $t_g = 2$ warm-up rounds needed to obtain stable gradient signatures. Even though these warm-up updates are reused for subsequent training (and hence are not wasted), we conservatively count them toward the clustering overhead. Thus, the per-client clustering cost of FEDDAG is

$$\text{FedDAG clustering cost: } \mathcal{O}\big(pNF\big) + \mathcal{O}\big(2NM_{\mathrm{fwd}}\big),$$

where $p$ is the number of principal vectors, $N$ is the local sample size, $F$ is the feature dimension, and $M_{\mathrm{fwd}}$ denotes the cost of a single forward pass of the full model on one sample.

By contrast, iterative clustered FL methods require evaluating all $K$ cluster models on the client's $N$ local samples in every communication round and repeating this over $T$ rounds. The resulting per-client clustering cost is:

$$\text{iterative clustering cost: } \mathcal{O}\big(KTNM_{\mathrm{fwd}}\big).$$

The ratio between the total iterative clustering cost and the (conservatively defined) clustering cost of FEDDAG is therefore

$$R = \frac{\text{iterative clustering cost}}{\text{FEDDAG clustering cost}} = \frac{KTNM_{\mathrm{fwd}}}{pNF + 2NM_{\mathrm{fwd}}} = \frac{KTM_{\mathrm{fwd}}}{pF + 2M_{\mathrm{fwd}}}.$$

To obtain a concrete sense of scale, consider a CIFAR-10 setup with 100 clients. The training set has 50,000 examples, so each client holds about $N \approx 500$ samples. Assume a standard convolutional network with three convolutional layers followed by two fully connected layers, whose encoder outputs $F = 256$-dimensional features and whose total parameter count is approximately $5.4 \times 10^5$. We approximate the per-sample forward cost by this parameter count, i.e., $M_{\mathrm{fwd}} \approx 5.4 \times 10^5$. Following our earlier assumption, we take the number of principal vectors to be a small fraction of the local data, $p = 0.02N \approx 10$.

Substituting these values into the denominator,

$$pF + 2M_{\mathrm{fwd}} \approx 10 \cdot 256 + 2 \cdot 5.4 \times 10^5 = 2{,}560 + 1.08 \times 10^6 \approx 1.08 \times 10^6,$$

so the term $2M_{\mathrm{fwd}}$ dominates. Consequently,

$$R \approx \frac{KTM_{\mathrm{fwd}}}{2M_{\mathrm{fwd}}} = \frac{KT}{2}.$$

For a representative clustered FL setting with $K = 5$ clusters and $T = 200$ communication rounds, we obtain

$$R \approx \frac{5 \cdot 200}{2} = 500,$$

meaning that the cumulative iterative clustering overhead is roughly $500\times$ larger than FEDDAG's one-time truncated SVD plus warm-up cost per client.

These estimates indicate that, in the regimes we study (small $p$ relative to $N$, moderate $K$, tens to hundreds of rounds, and a realistic convolutional backbone such as SimpleCNN), the additional clustering overhead of iterative $K$-model evaluation in clustered FL baselines dominates FEDDAG's one-time truncated SVD plus warm-up cost.

### A.6.2 ALTERNATIVE PRINCIPAL-VECTOR COMPUTATION VIA CLASS-WISE SUBSAMPLING

Although the analysis above shows that the one-shot clustering overhead of FEDDAG is significantly smaller than the per-round clustering cost of iterative clustered FL methods, we can further reduce the overall cost and make FEDDAG's clustering overhead comparable to other non-clustering FL approaches by modifying how we compute principal vectors.

**Subsampled principal vectors.** In our original class-wise weighted data-similarity mechanism (Eq. 4–6), each client $i$ uses the full set of samples $D_{i,c}$ for class $c$ to compute $p$ principal vectors $U_c^i$ via truncated SVD. In the alternative variant, each client instead constructs a class-wise subsample $\tilde{D}_{i,c} \subseteq D_{i,c}$ of size $s_{i,c}$, chosen according to its local computation and memory budget (e.g., a fixed per-class cap or a fixed fraction of $|D_{i,c}|$), and applies truncated SVD (Klema & Laub, 1980) on $\tilde{D}_{i,c}^\top$ to obtain $p$ principal vectors $\tilde{U}_c^i = [\tilde{u}_1, \ldots, \tilde{u}_p]$, which replace $U_c^i$ in the subsequent pipeline. The server then proceeds exactly as in Eq. 4–6, computing principal angles and applying class-frequency weighting using $\tilde{U}_c^i$, so that the final similarity $\tilde{\mathcal{V}}_{i,j}$ is obtained from subsampled data.

**Extension to CC-graph construction.** The same sampling idea can be applied when constructing the cluster complementarity graph (CC-graph) in §4. Instead of using all available samples to compute the alignment scores $\Gamma_{p,q,c}$ between clusters $p$ and $q$ for class $c$, we compute $\Gamma_{p,q,c}$ from class-wise subsamples drawn from the participating clusters. This further reduces the overall computation and communication overhead of FEDDAG.

### A.7 PRIVACY CONSIDERATIONS

Privacy is a foundational aspect of federated learning, which aims to enable collaborative model training while protecting the sensitive data of individual clients. In the context of FEDDAG, we examine the privacy implications of both the similarity estimation and representation-sharing phases. During client clustering, FEDDAG constructs a weighted, class-wise data similarity matrix using a small set of class-representative principal vectors and per-class sample counts provided by each client. Crucially, the shared principal vectors are reduced linear combinations of local data and do not expose any raw samples or labels. Moreover, each client contributes fewer than 1% of such vectors per class, ensuring minimal data exposure. This approach aligns with prior privacy-aware clustering methods Vahidian et al. (2023), which also transmit low-dimensional representative vectors to the server. In more privacy-sensitive deployments, additional protection mechanisms can be integrated into FEDDAG. For instance, secure aggregation protocols (Bonawitz et al., 2017), encryption techniques, or differential privacy can be used to protect the shared principal vectors.

Privacy mechanisms can also be used to prevent leakage of the class-frequency information that we use when weighting similarity values. For example, FLIPS (Bhope et al., 2023) employs a trusted execution environment (TEE) to protect label distributions, which are used to select a diverse set of clients during FL training. Similarly, FEDDAG could employ differential privacy (DP) (Dwork, 2006) by perturbing class counts with carefully calibrated noise so that the contribution of any single example is statistically hidden. As another privacy-based option, FEDDAG could employ homomorphic encryption (HE) (Gentry, 2009), allowing the server to compute weights directly on ciphertexts without learning the raw values. However, designing and implementing such privacy mechanisms is beyond the scope of this work, so we do not pursue them further.

---

**Algorithm 2:** Proximity Matrix Computation

---

**Input:** Principal vectors $U^*$, sparsified gradients $\tilde{\Delta}^*$ for all clients
**Output:** $\mathcal{A}$, proximity matrix between all client pairs
1 **Function:** `ProximityMatrix`$(U^*, \tilde{\Delta}^*)$
2 **for** *client* $i = 1, \ldots, N$ and **for** *client* $j = 1, \ldots, N$ **do**
3     **for** *class* $c = 1, \ldots, C$ **do**
4        Compute $\mathcal{V'}_{i,j,c}$ using Eq. 4
5        Compute $\mathcal{W'}_{i,j,c}$ using Eq. 5
6     Compute $\mathcal{V}_{i,j}$ using Eq. 6 and $\hat{\mathcal{V}} \leftarrow \text{normalize}(\mathcal{V})$
7     Compute $\mathcal{G}_{i,j}$ using Eq. 3 and $\hat{\mathcal{G}} \leftarrow \text{normalize}(\mathcal{G})$
8 Initialize weight vector $\mathbf{w} = (w_1, \ldots, w_N)^\top \in [0,1]^N$ randomly
9 **while** *not converged* **do**
10     Compute entropy loss $\mathcal{L}_{\text{en}}$ using Eq. 8
11     $\mathbf{w} \leftarrow \mathbf{w} - \eta \nabla_{\mathbf{w}} \mathcal{L}_{\text{en}}$; and $\mathbf{w} \leftarrow \text{clip}(\mathbf{w}, 0, 1)$ // `MLP-based update`
12 Compute $A_{i,j}$ as in Eq. 7 and **return** $\mathcal{A}_{i,j}$

---

**Algorithm 3:** Clustering Threshold Search in FL

---

**Input:** Proximity matrix $\mathcal{A}_{i,j}$, threshold set $S_\alpha$
**Output:** Optimal clustering $\{\mathbb{C}_1, \ldots, \mathbb{C}_Z\}$
1 **Function** `OptimalClustering`$(\mathcal{A}, S_\alpha)$:
2     Initialize empty list `records`
3     **for** $\alpha \in S_\alpha$ **do**
4        Generate candidate clustering $\mathbb{C}^\alpha$ using hierarchical clustering (HC) on $\mathcal{A}$ with threshold $\alpha$
5        Compute $\mathcal{L}_1$ and $\mathcal{L}_2$ (Eq. 9) for $\mathbb{C}^\alpha$
6        Total clustering score $\mathcal{L}_{\{\mathbb{C}_1, \ldots, \mathbb{C}_Z\}} = \mathcal{L}_1 + \lambda \mathcal{L}_2$
7        Save tuple $(\alpha, \mathcal{L}_{\{\mathbb{C}_1, \ldots, \mathbb{C}_Z\}})$ to `records`
8     Select $\alpha^*$ with low score and relatively small $Z$ from `records`
9     **return** Optimal clustering $\{\mathbb{C}_1, \ldots, \mathbb{C}_Z\} \leftarrow \mathbb{C}^{\alpha^*}$

---

To further mitigate information leakage during gradient-based similarity estimation, FEDDAG can adopt encryption strategies similar to those proposed in Sattler et al. (2020). During cross-cluster feature sharing (see §4), when a cluster requests representations from a source cluster, only the aggregated gradients computed from the source cluster's clients are shared. No individual client's gradient information is exposed at any point.

### A.8 GENERALIZATION TO NEWCOMERS — ALGORITHMS 4

**High-level idea.** In real-world FL systems, new clients may join after the initial clustering and model training have already begun. Moreover, clients may not always remain continuously available. To handle such cases, we extend FEDDAG with a lightweight mechanism that allows new clients to seamlessly join existing clusters without disrupting ongoing training. Specifically, each new client computes its data and gradient information, which are used to extend the proximity matrix to include similarity values for the new client. This updated matrix is then used by the clustering algorithm to determine the appropriate cluster assignment. Once assigned, the client is integrated into the designated cluster without re-evaluating the optimal clustering or retraining any previously learned weights.

**Details of the method.** The process for integrating a new client $i_{\text{new}}$ is similar to that used for initial clients (as in §3). FEDDAG first performs local training on $i_{\text{new}}$'s data for $t_g$ rounds to reach partial convergence. Afterwards, client $i_{\text{new}}$ computes its sparsified gradient update $\tilde{\Delta}^{i_{\text{new}}}$ and class-wise principal vectors $U_c^{i_{\text{new}}}$ and sends them to the server. The server updates the existing data similarity matrix $\hat{\mathcal{V}}_{i,j}$ and gradient similarity matrix $\hat{\mathcal{G}}_{i,j}$ to their extended forms $\hat{\mathcal{V}}_{i,j}^{\text{new}}$ and $\hat{\mathcal{G}}_{i,j}^{\text{new}}$, incorporating information from the new client. To combine the data and gradient, FEDDAG initially learns a weight vector $\mathbf{w}$ (see §3.3). To integrate the new client, FEDDAG extends this process by assigning a weight $w_{i_{\text{new}}} \in [0,1]$ and learning it using the same entropy loss (Eq. 8) used during initial training, but optimizing only for $w_{i_{\text{new}}}$ without modifying existing weights. Using the extended weights $\mathbf{w}_{\text{new}}$, the proximity matrix $\mathcal{A}_{i,j}^{\text{new}}$ is computed based on $\hat{\mathcal{V}}_{i,j}^{\text{new}}$, $\hat{\mathcal{G}}_{i,j}^{\text{new}}$, and $w_{i_{\text{new}}}$, as defined in Eq. 7 in §3.3.

Finally, the server reuses the previously selected clustering threshold $\alpha^*$ (from Optimal Clustering in §3.4) and performs a single hierarchical clustering (HC) pass on $\mathcal{A}_{i,j}^{\text{new}}$ to assign $i_{\text{new}}$ to a cluster $\mathbb{C}_{z(i_{\text{new}})}$. After assignment, client $i_{\text{new}}$ initializes its model from the corresponding cluster's global parameters and directly joins the existing FEDDAG training flow (i.e., the **else** branch at line 15 in

---

**Algorithm 4:** Generalization to Newcomers

---

**Input:** New client $i_{\text{new}}$, clustering threshold $\alpha^*$, current clusters $\{\mathbb{C}_1, \ldots, \mathbb{C}_Z\}$, current proximity matrix $\mathcal{A}_{i,j}$, data matrix $\mathcal{V}_{i,j}$, gradient matrix $\mathcal{G}_{i,j}$

**Output:** Updated client $i_{\text{new}}$ models

1 **Function** NewcomerIntegration($i_{new}$, $\alpha^*$):
2      Initialize client $i_{\text{new}}$ with random $\theta^0_{i_{\text{new}}}$
3      Set local counter $t_{i_{\text{new}}} = 0$
     // Tracks local warm-up rounds
4      **for** *each global round t* **do**
5          **if** $t_{i_{new}} < t_g$ **then**
6              Local training of $\theta^0_{i_{\text{new}}}$ using local data (no federation)
7              $t_{i_{\text{new}}} \leftarrow t_{i_{\text{new}}} + 1$
8              **if** $t_{i_{new}} = t_g$ **then**
9                  Client $i_{\text{new}}$ sends $\tilde{\Delta}^{i_{\text{new}}}$ and $U_c^{i_{\text{new}}}$ to server
                 // --- Extend proximity matrix ---
10                  Server extends $\hat{\mathcal{V}}^{\text{new}}_{i_{\text{new}},j}$ and $\hat{\mathcal{G}}^{\text{new}}_{i_{\text{new}},j}$ to include the new client
11                  Server initializes $w_{i_{\text{new}}} \in [0, 1]$ and learns it using Eq. 8 (§3), keeping existing weights fixed
12                  Server extends proximity matrix $\mathcal{A}^{\text{new}}_{i,j}$ using Eq. 7 (§3) with $\hat{\mathcal{V}}^{\text{new}}_{i,j}$ and $\hat{\mathcal{G}}^{\text{new}}_{i,j}$
                 // --- Cluster assignment ---
13                  Server executes hierarchical clustering with $\alpha^*$ on $\mathcal{A}^{\text{new}}_{i,j}$ to assign $i_{\text{new}}$ to cluster $\mathbb{C}_{z(i_{\text{new}})}$
14                  Client $i_{\text{new}}$ sets $\theta^{1f}_{i_{\text{new}}}, \theta^c_{i_{\text{new}}}$ from $(\Theta^{1f}_{z(i_{\text{new}})}, \Theta^c_{z(i_{\text{new}})})$
                 // Aggregate secondary encoders from related clusters (via $\mathcal{H}$)
15                  Client $i_{\text{new}}$ sets $\theta^{2f'}_{i_{\text{new}}} \leftarrow \sum_{j:H(j,z(i_{\text{new}}))=1} \Theta^{2f}_j$
16          **else**
             // --- Standard training phase (same as **else** branch in Algorithm 1) ---
17              Train $(\theta^{1f}_{i_{\text{new}}}, \theta^c_{i_{\text{new}}})$ via SGD using Eq. 17
             // Primary training
18              Train $\theta^{2f'}_{i_{\text{new}}}$ via SGD using Eq. 20
             // Secondary training
19              Broadcast updated $(\theta^{1f}_{i_{\text{new}}}, \theta^c_{i_{\text{new}}}, \theta^{2f'}_{i_{\text{new}}})$ to server

---

Algorithm 1). This extension enables efficient onboarding of new clients by reusing the established clustering threshold and global models, avoiding disruption to ongoing training. The complete process is summarized in Algorithm 4. Also, we evaluate the generalization capability of FEDDAG to unseen clients through experiments reported in Appendix §B.6.

## A.9    HANDLING DATA-DISTRIBUTION SHIFT

**High-level idea.** After FEDDAG has converged, the data of already-clustered clients may still evolve over time (e.g., new sensor drifts, changes in user behavior). If the local distribution of a client drifts too far from what its current cluster represents, the global model quality may degrade. We, therefore, add a mechanism that decides whether a client needs to be re-evaluated for cluster assignment. In addition, to accommodate a growing client population, FEDDAG periodically re-assesses the clustering to ensure the configuration remains consistent with the evolving client landscape. Specifically, the Wasserstein distance (Duan et al., 2021) is employed to track shifts in the class distribution of each client's local data over time; when a significant shift is detected, the system recomputes that client's data and gradient representations and re-evaluates its proximity to other clients using the same similarity fusion mechanism described in §3. This enables re-clustering of the client without disrupting other participants or restarting global training.

**Client re-evaluation.** Let $\mathcal{P}_i^{(t)}$ denote the empirical class histogram of client $i$ at round $t$. Every $\delta'$ rounds, we compute the 1-Wasserstein distance[3] between the current and previous histograms as $W_1\big(\mathcal{P}_i^{(t)}, \mathcal{P}_i^{(t-\delta')}\big)$. Client $i$ is marked as shifted if:

$$W_1\big(\mathcal{P}_i^{(t)}, \mathcal{P}_i^{(t-\delta')}\big) \; > \; \tau_i \; := \; \frac{0.2}{LabelSize} \cdot n_i, \tag{41}$$

where $n_i$ is the number of new samples processed by client $i$ since round $t-\delta'$. Eq. 41 flags a shift when roughly 20% of local data has changed.

---

[3]For image classification, we treat classes as discrete points on the line $0, \ldots, C-1$; the 1-Wasserstein distance then has a closed form based on cumulative histograms.

A shifted client does not immediately trigger global re-clustering. Instead, its cluster assignment is re-evaluated through a procedure that re-computes data and gradient information for the server (similar to generalizing to newcomer clients in §A.8). For the next $t_g$ rounds, the shifted client $i$ trains its primary encoder and classifier on local data *without federation* so that the resulting gradients reflect its own distribution rather than the global model. After local training, the client computes its gradient update $\Delta^i$, applies $k$-sparsification to obtain $\tilde{\Delta}^i$, re-computes class-wise principal vectors $U_c^i$ from local data, and sends $U_c^i$ and $\tilde{\Delta}^i$ to the server. These components update the data similarity matrix $\hat{\mathcal{V}}_{i,j}$, the gradient similarity matrix $\hat{\mathcal{G}}_{i,j}$, and the proximity matrix $\mathcal{A}_{i,j}$. The server then re-evaluates clients' cluster assignments by performing a single hierarchical-clustering pass on the updated $\mathcal{A}_{i,j}$ using the fixed optimal threshold $\alpha^*$ (derived in §3.4). If a reassignment occurs, the client initializes its model from the corresponding global model and continues training.

**Accommodating growing population.** To support an expanding set of participants, FEDDAG periodically re-evaluates the clustering after a specified number of new clients have joined. This reassessment determines whether the updated client distribution warrants a change in the cluster structure. Concretely, FEDDAG re-runs the optimal clustering selection procedure by sweeping over candidate threshold values $\alpha$ (as in §3.4). If a new clustering configuration is chosen, the algorithm updates the necessary components (e.g., the cluster complementarity graph, re-initializes the global model from client models) and resumes training, ensuring consistency with the evolving client landscape.

### A.10    REASSESSING THE NOVELTY OF FEDDAG AND THE GRS MECHANISM

A core novelty of FEDDAG is its dual-encoder design for knowledge sharing among clusters via the proposed Global Representation Sharing (GRS) mechanism. At first glance, this architecture may appear similar to (i) personalized/shared head designs in personalized FL and (ii) personalized/shared feature-extractor designs in feature-skewed FL. Moreover, one may view FEDDAG's GRS mechanism as unnecessary overhead compared to the personalized/shared head baselines, thus requiring a clear justification for adopting this architecture. In this section, we reassess the novelty of FEDDAG relative to prior personalized FL approaches and explain why FEDDAG's GRS is fundamentally different from simply sharing a single global encoder.

**Comparison to traditional FL.** Classical personalized FL often assumes that a single feature extractor can be globally shared and only the classifier head needs to be personalized, which works well when client feature distributions are roughly aligned. In heavily non-IID settings, however, clients are grouped into clusters with severe label skew and feature shift; in those cases, a shared encoder/feature extractor cannot satisfy both global and local needs, as different clients can have different learning objectives based on their data distribution. FedBR (Guo et al., 2023) explicitly shows via visualization that feature extraction is not a purely global, universally shared job, as in, even for the same input, local-model and global-model features can differ significantly.

To tackle this generalization vs. personalization issue, a separate line of FL work maintains both local and global feature extractors or adds supervisors/auxiliary encoders to relate them. For example, FedBR (Guo et al., 2023) uses a local and a global feature extractor to reduce classifier bias while preserving client-specific features, and FedSimSup (Liu et al., 2025) lets each client hold two full models—a local supervisor and an inter-learning model—where the supervisor aligns the inter-learning model with heterogeneous local data.

**Comparison to other personalized/global FL methods.** In summary, FEDDAG can be included in the family of methods that add an extra encoder to balance generalization and personalization. However, it leverages the clustered FL framework to improve upon existing designs. While the idea of decoupling a model into two components is not new by itself, how these encoders are trained and what they represent in FEDDAG is fundamentally different. Most prior global/local feature schemes maintain: **(i) Global part:** a shared global feature extractor, **(ii) Personalized part:** a per-client local feature extractor. How FEDDAG differs compared to above setitng is dfined below.

**1. Global feature vs. selective complementary features.** In the above-mentioned pFL approach, sharing a single global feature extractor can be suboptimal. The global feature space is influenced by all clients, including:

- Clients with very different learning objectives or data distributions, and
- Clients with poor or unstable features (e.g., very few samples for certain classes).

This mixture can lead to negative and noisy feature transfer: aligning a representation with *all clients* is often harmful, especially when only a subset of clients are truly relevant or complementary.

In contrast, FEDDAG does not simply blend all clients into one global representation. Instead, it:

- Uses the CC-Graph to identify which clusters are actually beneficial, and
- Imports features only from those complementary clusters via the secondary encoder.

In this sense, FEDDAG factorizes what would otherwise be a monolithic global feature space into:

(cluster-specific personal features) + (selected complementary features from other clusters),

thereby avoiding contamination from unrelated or comparatively less relevant client sources.

**2. Secondary encoder trained directly on other clusters' data.** In FEDDAG, the secondary encoder of the learner cluster is explicitly trained on the datasets of clients from source clusters:

- The secondary encoder parameters are sent to a source cluster identified by the CC-Graph.
- At the source cluster, that encoder is locally trained on the clients' data, and the resulting gradients or updates are then sent back to the learner clusters.

Thus, the secondary encoder acts as a parameter carrier that travels across clusters, learns on other clusters' data, and then returns with an updated gradient.

This is fundamentally different from a single global feature extractor in feature-skew FL. Those approaches typically rely on knowledge-transfer signals such as pseudo-data, distilled logits, feature embedding, or prototype features to gather global representation; to our knowledge, they do **not** perform explicit remote training of a model component on other clients' local data. This limits their ability to fully exploit the complementary feature structure present in other domains.

In summary, by leveraging the structure and training workflow of clustered federated learning, FED-DAG's secondary encoder performs *targeted cross-cluster feature extraction* guided by the CC-Graph, rather than functioning as just another shared or personalized feature extractor.

## A.11 DUAL-ENCODER ALTERNATIVE INITIALIZATION AND TRAINING

**High-level idea.** In our dual-encoder architecture, the goal is for the primary and secondary encoders to capture *complementary* information. In the main FEDDAG pipeline, this is facilitated by initializing the primary encoder using partially converged gradients obtained during gradient-based similarity estimation, rather than initializing both encoders at random. Here, we investigate an alternative strategy to initialize encoders. regularizer. This variant modifies FEDDAG dual-encoder initialization and architecture as follows:

- Both encoders in each cluster are initialized randomly
- The primary-encoder objective is augmented with a regularization term that penalizes excessive alignment between the feature representations of the two encoders.

All other components of FEDDAG (cluster-wise FedAvg, CC-Graph–guided enrichment, secondary-encoder updates, etc.) remain unchanged. This variant is described in detail below:

**Details of the method.** At first, FEDDAG initializes both the primary-encoder parameters $\Theta^{1f}$ and the secondary-encoder parameters $\Theta^{2f}$ randomly, as described above. Then, during the primary-encoder training phase (see Eq. 16), to explicitly encourage $\phi^{(1)}(\cdot; \theta^{1f})$ and $\phi^{(2)}(\cdot; \Theta^{2f})$ to learn complementary representations rather than collapse to similar feature directions, we introduce a diversity regularizer $R_{\mathrm{div}}$ that penalizes excessive alignment between their outputs, as follows:

$$R_{\mathrm{div}}(\theta_i^{1f}, \Theta_{z(i)}^{2f}) = \mathbb{E}_{x \sim D_i} \left[ \left( \cos(\phi^{(1)}(x; \theta_i^{1f}), \ \phi^{(2)}(x; \Theta_{z(i)}^{2f})) \right)^2 \right], \tag{42}$$

where $\cos(\cdot, \cdot)$ denotes cosine similarity. Large cosine similarity indicates alignment between the encoders; minimizing $\cos^2$ therefore pushes the encoders toward complementary feature directions.

The local objective, as shown in Eq. 16, after modification becomes:

$$\ell_i^{\mathrm{div}}(\theta_i^{1f}, \theta_i^c) = \mathcal{L}\Big(Y_i, \psi(\phi^{(1)}(X_i; \theta_i^{1f}), \phi^{(2)}(X_i; \Theta_{z(i)}^{2f}); \theta_i^c)\Big) + \lambda_{\mathrm{div}}\, R_{\mathrm{div}}(\theta_i^{1f}, \Theta_{z(i)}^{2f}), \quad (43)$$

where $\lambda_{\mathrm{div}} > 0$ controls the strength of the diversity term. During local SGD, client $i$ updates $(\theta_i^{1f}, \theta_i^c)$ using $\ell_i^{\mathrm{div}}(\theta_i^{1f}, \theta_i^c)$, while $\Theta_{z(i)}^{2f}$ remains fixed and is updated later via the secondary-encoder enrichment step. The secondary-encoder training and all other components of FEDDAG remain unchanged. We also evaluate the effectiveness of this alternative encoder initialization empirically as shown in §B.8.

# B  ADDITIONAL EXPERIMENTS

In this section, we show implementation details, ablation studies, additional experiments regarding hyperparameter selection and sensitivity, and FEDDAG performance on different data distributions.

## B.1  IMPLEMENTATION DETAILS

We now describe the implementation details used in our experiments, including model architectures and training hyperparameters. For datasets such as CIFAR-10 and SVHN, we adopt a convolutional neural network (LeCun et al., 2002) composed of three convolutional layers followed by two fully connected layers. For FMNIST, we use a simpler architecture with two convolutional layers and a single dense layer. Local training on each client is performed using stochastic gradient descent (SGD) with a learning rate of 0.01, momentum of 0.5, weight decay of $1 \times 10^{-4}$, and a batch size of 64. Each client trains locally for 10 epochs per round. Unless stated otherwise, we run a total of 200 global communication rounds, with 20% of clients sampled per round. We report classification performance using balanced accuracy, averaged across clients to account for non-IID data distributions.

## B.2  HYPERPARAMETER TUNING

In the context FEDDAG, hyperparameters play a crucial role in determining the model's performance, stability, and robustness. To better understand the effectiveness of FEDDAG, we investigate how sensitive the algorithm is to variations in different hyperparameters.

**Local Steps $(t_g)$.** The parameter $t_g$ controls the number of local training epochs each client performs on its own data before sending gradient information to the server. This step is crucial for estimating each client's gradient direction, which is used to compute the gradient similarity matrix. Since this training is done without any federation, the resulting gradients reflect only the client's local data. The choice of $t_g$ affects the trade-off between computation efficiency and the quality of similarity estimation. Ideally, we want $t_g$ to be as small as possible, while still enabling the gradients to converge enough to produce meaningful similarity measurements. Table 6 shows how accuracy varies with different values of $t_g$ across datasets. In these experiments, the gradient similarity matrix alone (instead of combining data and gradient) was used as the proximity matrix to assess how effectively gradient information captures client similarity. We also switch off dual-encoder and only use single encoder FEDDAG* (see §5). The setup follows Data Distribution I (see §5) with $\alpha' = 1$, $\rho = 30\%$, and consistent hyperparameter settings. Each communication round included 10 local training steps. As shown, increasing $t_g$ improves accuracy initially, as longer local training leads to more stable and comparable gradients. However, accuracy plateaus around $t_g = 2$ for most datasets, indicating that the gradients have sufficiently converged for reliable similarity estimation. Beyond this point, additional local steps yield diminishing returns. Therefore, $t_g = 2$ provides a good trade-off between accuracy and efficiency.

**Weight Range for Data Similarity Matrix $(\delta)$.** The parameter $\delta$ controls the sensitivity of the data similarity matrix to dataset size imbalance when comparing clients. Specifically, this weighting mechanism penalizes similarity scores between clients with large differences in dataset sizes, thereby reflecting the quantity shift more accurately. The impact of these size-based penalties is governed by the value of $\delta$: smaller values result in minimal influence, while larger values increase the penalty's effect. Each computed similarity value is reweighted and normalized into the range $[1 - \delta, 1 + \delta]$, with $\delta \in [0, 1)$, allowing the final similarity score to scale by at most a factor of two.

Table 6: Test accuracy for different values of local training rounds $t_g$ (with 10 local steps per round) across datasets, evaluated under Data Distribution I with 30% class skew and Dirichlet $\alpha' = 1$.

| $t_g$ | CIFAR-10 | SVHN | FMNIST |
|---|---|---|---|
| 1 | 80.81±0.59 | 84.82±0.24 | 93.18±0.11 |
| 2 | 83.34±0.52 | 90.05±0.16 | 93.18±0.11 |
| 3 | 83.34±0.52 | 90.05±0.16 | 93.18±0.11 |

Table 7 report test accuracy for various values of $\delta$ across multiple datasets. Since the weighting mechanism primarily addresses size disparity and quantity shift, we examine its effect under the Dirichlet concentration factor: $\alpha' = 0.25$ (severe shift). To isolate the effect of different values of $\delta$ on accuracy, we use only the data similarity matrix (instead of combining data and gradient similarity) when computing the proximity matrix for clustering. These experiments follow the Data Distribution I described in §5, using identical hyperparameters. From Table 7, we observe that higher $\delta$ values (e.g., 0.6) can lead to improved clustering and accuracy. This suggests that the weighting scheme is particularly beneficial in highly heterogeneous environments, where accounting for dataset size differences enhances similarity estimation.

Table 7: Accuracy metrics for various values of weight range $\delta$ under Data Distribution I (Dirichlet $\alpha' = 0.25$, 20% class skew).

| $\delta$ | CIFAR-10 | SVHN | FMNIST |
|---|---|---|---|
| **0.2** | 87.51±0.18 | 91.74±0.08 | 91.79±0.08 |
| **0.4** | 87.51±0.18 | 91.74±0.08 | 92.21±0.09 |
| **0.6** | 87.95±0.13 | 91.91±0.13 | 92.21±0.09 |
| **0.8** | 87.95±0.13 | 91.91±0.13 | 92.21±0.09 |
| **1.0** | 87.95±0.13 | 91.91±0.13 | 92.21±0.09 |

**Top-$k$ values in *CC-Graph*.** The parameter $k$ determines how many top-ranked clusters are selected as knowledge sources for each target cluster in the complementarity graph $H$. This graph guides which clusters will supply feature representations to others during the secondary encoder training phase. For every row in the complementarity score matrix (Eq. 12), only the top-$k$ highest scoring entries are retained to form directed edges. A smaller $k$ limits each cluster to fewer sources, possibly reducing noise but also restricting diversity. In contrast, a larger $k$ increases the opportunities for learning from complementary clusters but may include low-quality connections that dilute representation quality.

Table 8 shows how varying the number of source clusters $k$ in the complementarity graph $H$ affects the performance of FEDDAG. Experiments are conducted under Data Distribution I with Dirichlet concentration parameter $\alpha' = 1$ and 30% label skew, while keeping all other hyperparameters fixed. We observe that performance generally improves when $k \geq 2$, benefiting from knowledge transfer across multiple relevant clusters. In some cases, increasing $k$ beyond 2 continues to help (e.g., FMNIST), while in others (e.g., SVHN), it leads to marginal drops in accuracy. This suggests that the optimal value of $k$ depends on the dataset characteristics and the number of clusters in the current formation. We adopt $k = 2$ as a balanced choice to ensure diversity while maintaining relevance.

Table 8: Accuracy for different values of top-$k$ retained in the complementarity graph $H$ under Data Distribution I (Dirichlet $\alpha' = 1$, 30% class skew).

| $k$ | CIFAR-10 | SVHN | FMNIST |
|---|---|---|---|
| **1** | 90.85±0.13 | 97.09±0.08 | 97.71±0.05 |
| **2** | **91.02±0.12** | **97.19±0.04** | 98.36±0.10 |
| **3** | 90.95±0.16 | 97.01±0.07 | **98.57±0.08** |
| **4** | 90.96±0.21 | 96.78±0.12 | 98.41±0.11 |

**Sparsified Gradient $\tilde{\Delta}^i$.** From Section §3.1 that, after a short local warm-up, each client $i$ computes a gradient update $\Delta^i$ on its local data and transmits a $k$-sparsified version $\tilde{\Delta}^i$ to the server. Here, $\tilde{\Delta}^i$ retains only a small random subset of coordinates of $\Delta^i$, and the *gradient sparsification ratio* refers to the fraction of coordinates that are kept in $\tilde{\Delta}^i$ when constructing the gradient similarity matrix in Eq. 3. In this subsection, we study how the final test accuracy varies as we change this sparsification

ratio. To isolate the effect of sparsification, we use *only* gradient-based similarity (no data-based similarity) when forming the client similarity matrix. We also switch off dual-encoder and only use single encoder FEDDAG* (see §5). The experimental setup matches our main configuration for Data Distribution I: we use a 30% label-skew with Dirichlet $\alpha' = 1$, fix the local warm-up to $t_g = 2$ rounds with 10 local steps per round, and vary the fraction of coordinates retained in each $\tilde{\Delta}^i$.

Table 9: Test accuracy (%) as a function of the gradient sparsification ratio with only gradient similarity and singel encoder (fraction of coordinates retained in $\tilde{\Delta}^i$) under Data Distribution I (30% label skew, $\alpha' = 1$, $t_g = 2$).

| Sparsity (entries kept) | CIFAR-10 | SVHN | FMNIST |
|---|---|---|---|
| 0.1% | $82.52 \pm 0.60$ | $89.23 \pm 0.25$ | $92.81 \pm 0.14$ |
| 0.5% | $83.34 \pm 0.52$ | $90.05 \pm 0.16$ | $93.18 \pm 0.11$ |
| 1% | $83.34 \pm 0.52$ | $90.05 \pm 0.16$ | $93.18 \pm 0.11$ |
| 2% | $83.34 \pm 0.52$ | $90.05 \pm 0.16$ | $93.18 \pm 0.11$ |
| 20% | $83.34 \pm 0.52$ | $90.05 \pm 0.16$ | $93.18 \pm 0.11$ |

Table 9 reports the test accuracy as a function of the sparsification ratio (percentage of coordinates kept in $\tilde{\Delta}^i$). We observe that keeping as little as $0.5\%$ of the gradient coordinates in $\tilde{\Delta}^i$ is already sufficient to obtain essentially the same accuracy as much denser settings. For sparsification ratios at or above $0.5\%$, the resulting gradient-based similarities are very similar, leading to almost identical clustering structure and final performance. In contrast, at $0.1\%$ sparsity, the similarities become noisier, slightly degrading clustering quality and accuracy. Based on these results, in our main experiments we choose sparsification ratios in the range of $1$–$2\%$ for $\tilde{\Delta}^i$, which provides a good trade-off between communication efficiency and clustering quality.

## B.3 EXPERIMENTS ON DATA DISTRIBUTION I

**Exp 1: Performance Evaluation.** This section presents the additional results referenced in the main paper for $\alpha' = 1$, under class skew $\rho = 20\%$ and $30\%$, following the setup described in §5. The results, shown in Table 10, further validate the effectiveness of FEDDAG on Data Distribution I under moderate quantity shift. The same set of baselines is used, and results are reported across all four datasets.

Table 10: Exp 1: Performance comparison for Data Distribution I with 20% and 30% non-IID label skew under low quantity shift (Dirichlet $\alpha' = 1$).

| Algorithm | 20% Label Skew | | | | 30% Label Skew | | | |
|---|---|---|---|---|---|---|---|---|
| | CIFAR-10 | FMNIST | SVHN | CIFAR-100 | CIFAR-10 | FMNIST | SVHN | CIFAR-100 |
| FedAvg | $46.20 \pm 0.97$ | $57.12 \pm 0.30$ | $74.61 \pm 0.36$ | $51.34 \pm 0.78$ | $57.48 \pm 0.17$ | $77.17 \pm 0.24$ | $68.34 \pm 0.45$ | $53.13 \pm 1.46$ |
| FedProx | $46.77 \pm 0.14$ | $56.81 \pm 0.16$ | $77.23 \pm 0.45$ | $53.38 \pm 0.86$ | $57.80 \pm 0.23$ | $73.87 \pm 0.25$ | $69.65 \pm 0.19$ | $53.97 \pm 0.85$ |
| PerFedAvg | $84.68 \pm 0.19$ | $91.18 \pm 0.21$ | $92.34 \pm 0.13$ | $69.43 \pm 0.22$ | $82.83 \pm 0.14$ | $94.74 \pm 0.17$ | $91.48 \pm 0.29$ | $60.70 \pm 0.30$ |
| FedSoft | $77.42 \pm 0.21$ | $87.64 \pm 0.35$ | $90.48 \pm 0.24$ | $65.98 \pm 0.37$ | $76.94 \pm 0.38$ | $89.56 \pm 0.37$ | $84.86 \pm 0.45$ | $56.61 \pm 0.31$ |
| PACFL | $90.45 \pm 0.30$ | $94.41 \pm 0.31$ | $94.96 \pm 0.12$ | $70.35 \pm 0.36$ | $87.01 \pm 0.38$ | $97.28 \pm 0.24$ | $94.36 \pm 0.19$ | $63.91 \pm 0.76$ |
| CFL | $72.80 \pm 0.66$ | $86.97 \pm 0.23$ | $82.06 \pm 0.34$ | $61.43 \pm 0.92$ | $71.85 \pm 0.79$ | $85.67 \pm 0.23$ | $80.23 \pm 0.25$ | $52.90 \pm 1.17$ |
| CFL-GP | $87.83 \pm 0.19$ | $91.45 \pm 0.27$ | $90.38 \pm 0.16$ | $69.73 \pm 0.20$ | $85.67 \pm 0.25$ | $96.82 \pm 0.24$ | $92.29 \pm 0.09$ | $61.24 \pm 0.73$ |
| FedGWC | $89.58 \pm 0.17$ | $93.56 \pm 0.09$ | $93.67 \pm 0.13$ | $72.75 \pm 0.29$ | $86.18 \pm 0.25$ | $96.97 \pm 0.14$ | $92.94 \pm 0.15$ | $61.35 \pm 0.43$ |
| FedRC | $76.12 \pm 0.28$ | $86.45 \pm 0.42$ | $89.22 \pm 0.31$ | $64.78 \pm 0.33$ | $75.12 \pm 0.31$ | $91.02 \pm 0.44$ | $83.67 \pm 0.38$ | $57.89 \pm 0.24$ |
| IFCA | $89.68 \pm 0.17$ | $94.02 \pm 0.09$ | $93.28 \pm 0.13$ | $72.86 \pm 0.29$ | $86.42 \pm 0.25$ | $96.61 \pm 0.14$ | $92.86 \pm 0.19$ | $61.34 \pm 0.43$ |
| **FEDDAG** | $\mathbf{94.53 \pm 0.12}$ | $\mathbf{96.82 \pm 0.18}$ | $\mathbf{97.04 \pm 0.23}$ | $\mathbf{75.32 \pm 0.33}$ | $\mathbf{91.02 \pm 0.12}$ | $\mathbf{98.36 \pm 0.10}$ | $\mathbf{97.19 \pm 0.04}$ | $\mathbf{67.17 \pm 0.61}$ |

**Exp 6: Convergence Under Limited Communication rounds.** We compare the performance of FEDDAG against SOTA baselines under a constrained communication budget of 80 rounds. Figure 3 reports the final local test accuracy versus the number of communication rounds for four datasets. The results demonstrate that FEDDAG consistently converges within 20 to 30 communication rounds, outperforming all other methods in both convergence speed and final accuracy.

## B.4 DATA DISTRIBUTION IV

This distribution evaluates FEDDAG under a combination of feature skew and label skew. To simulate feature skew, we follow an approach similar to FedRC (Guo et al., 2024), which leverages

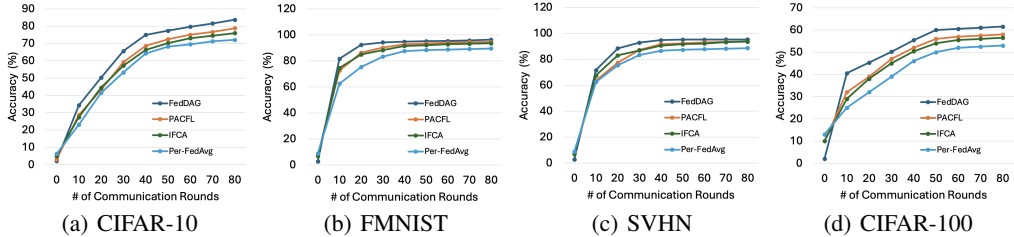

Figure 3: Exp 6: Accuracy vs. number of comm. rounds, Data Distribution I, non-IID (30%), $\alpha'=1$.

datasets (e.g., CIFAR-10-C) that apply diverse image corruptions or use domains, thereby introducing different feature styles. To simulate label skew, we adopt the LDA method (Hsu et al., 2019). Specifically, each client is assigned one of the available corruption types (e.g., fog, contrast, etc. for CIFAR-10-C) or domains (e.g., cartoon, photo, etc. for PACS) to create feature skew, and the samples are distributed using the Dirichlet factor $\alpha' = 1$.

Table 11: Exp 8: Performance comparison of various SOTA algorithms and FEDDAG under combined feature skew and label skew (Data Distribution IV). Feature skew arises from corruption types on CIFAR-10-C and TINY IMAGENET-C, and from domains on PACS and Office-Caltech-10.

| Algorithm | CIFAR-10-C | TINY IMAGENET-C | PACS | Office-Caltech-10 |
|---|---|---|---|---|
| FedAvg | $30.73 \pm 0.36$ | $18.43 \pm 0.43$ | $38.47 \pm 0.28$ | $46.12 \pm 0.31$ |
| PerFedAvg | $60.39 \pm 0.13$ | $25.54 \pm 0.31$ | $70.82 \pm 0.22$ | $67.34 \pm 0.23$ |
| FBLG | $60.25 \pm 0.30$ | $29.91 \pm 0.23$ | $72.38 \pm 0.37$ | $68.44 \pm 0.14$ |
| FedBR | $61.09 \pm 0.32$ | $30.83 \pm 0.25$ | $73.83 \pm 0.32$ | $71.37 \pm 0.18$ |
| FedMix | $60.11 \pm 0.14$ | $30.71 \pm 0.24$ | $73.25 \pm 0.41$ | $70.56 \pm 0.23$ |
| PACFL | $63.62 \pm 0.22$ | $33.53 \pm 0.38$ | $76.63 \pm 0.19$ | $73.31 \pm 0.14$ |
| CFL | $59.48 \pm 0.15$ | $28.97 \pm 0.26$ | $72.14 \pm 0.25$ | $68.02 \pm 0.19$ |
| FedRC | $61.82 \pm 0.21$ | $32.14 \pm 0.19$ | $76.28 \pm 0.17$ | $73.54 \pm 0.22$ |
| IFCA | $62.52 \pm 0.39$ | $32.33 \pm 0.19$ | $75.12 \pm 0.21$ | $72.48 \pm 0.29$ |
| **FEDDAG** | $\mathbf{65.62 \pm 0.31}$ | $\mathbf{36.27 \pm 0.32}$ | $\mathbf{80.34 \pm 0.27}$ | $\mathbf{76.28 \pm 0.20}$ |

**Dataset.** We use four datasets for this task in the FL setting: CIFAR-10-C, TINY IMAGENET-C (Hendrycks & Dietterich, 2019), PACS Li et al. (2017), and Office-Caltech-10 Gong et al. (2014), thereby covering both corruption-based and domain-level feature shift.

**Exp 8: Performance under Feature Skew.** We evaluate the performance of SOTA algorithms and FEDDAG on different datasets under a combination of feature skew and label skew. Each client is randomly assigned one of the 20 available corruption types (For CIFAR-10-C and TINY IMAGENET-C), or one of the four available domain types (for PACS and Office-Caltech-10). In both cases, the samples are distributed using a Dirichlet concentration factor $\alpha' = 1$. We randomly select 80% of clients for training and keep the remaining 20% as unseen clients, reporting test accuracy on these held-out clients using the final trained models. To more directly compare against approaches specialized for quantity shift—which naturally arises in this Dirichlet-based sample allocation—we also include the dedicated baseline FBLG Xu et al. (2024b), which employs a client-selection strategy that prioritizes clients with larger local datasets while grouping clients with similar sizes. The results in Table 11 show that FEDDAG consistently achieves higher accuracy than the baseline methods. This improvement is attributed to FEDDAG's data-based similarity metric, which provides more accurate feature similarity estimation compared to existing approaches. Across all four feature-skew benchmarks, FEDDAG achieves the best performance, providing direct empirical evidence of its robustness to feature distribution shift combined with label skew.

## B.5 EXPERIMENT ON LARGE-SCALE REAL-WORLD DATASET

To evaluate FEDDAG in a large-scale, real-world setting, we additionally evaluate it on the Google Landmarks dataset Weyand et al. (2020), following the setup of Licciardi et al. (2025). Specifically, we consider the Landmarks-Users-160K partition, where the dataset is partitioned into 1,000 clients based on the landmark dataset's authorship information. All other aspects of the experimental setup are kept the same as in our **Data Distribution I** experiments in Section 5. We compare FEDDAG against a group of established FL baselines, and the results are reported in Table 12.

Table 12: Performance on the large-scale real-world Google Landmarks dataset.

| Dataset | FedAvg | PACFL | CFL | FedGWC | IFCA | FedDAG |
|---|---|---|---|---|---|---|
| Google Landmarks | $36.53 \pm 0.24$ | $54.74 \pm 0.21$ | $45.29 \pm 0.28$ | $51.51 \pm 0.31$ | $51.97 \pm 0.16$ | $\mathbf{58.23 \pm 0.15}$ |

## B.6 EXPERIMENT ON GENERALIZATION TO NEWCOMERS

To assess the ability of FEDDAG to generalize to unseen clients (see Appendix A.8), we simulate a dynamic FL environment using Data Distribution I with 30% label skew and Dirichlet concentration factor $\alpha' = 1$. Initially, training is performed on 80 out of 100 clients for 80 communication rounds, following the standard FEDDAG procedure. At the end of this phase, the remaining 20 clients join the system as newcomers. Each newcomer executes steps (1–15) of Algorithm 4 and is assigned to a cluster. Once assigned, the client receives the current global model from its designated cluster and personalizes it for 1 round (10 local epochs). To evaluate model quality, we report the average final test accuracy of the 20 newcomers across different datasets. As shown in Table 13, FEDDAG achieves better generalization to newcomers than competing methods. This improvement is attributed to its robust cluster assignment and generalization strategy for new clients.

Table 13: Test accuracy of newcomer clients, Data Distribution I with 30% label skew and $\alpha' = 1$.

| Algorithm | CIFAR-10 | FMNIST | SVHN |
|---|---|---|---|
| FedAvg | $55.38 \pm 0.15$ | $74.93 \pm 0.22$ | $66.86 \pm 0.28$ |
| PerFedAvg | $80.92 \pm 0.10$ | $92.62 \pm 0.17$ | $90.00 \pm 0.15$ |
| FedSoft | $74.98 \pm 0.27$ | $87.45 \pm 0.22$ | $83.59 \pm 0.12$ |
| PACFL | $85.33 \pm 0.15$ | $95.17 \pm 0.24$ | $92.76 \pm 0.08$ |
| CFL | $69.97 \pm 0.11$ | $83.64 \pm 0.13$ | $78.94 \pm 0.17$ |
| FedGWC | $84.30 \pm 0.13$ | $94.53 \pm 0.10$ | $91.56 \pm 0.06$ |
| FedRC | $73.36 \pm 0.26$ | $88.91 \pm 0.21$ | $82.36 \pm 0.12$ |
| IFCA | $84.55 \pm 0.22$ | $94.61 \pm 0.30$ | $91.58 \pm 0.22$ |
| **FEDDAG** | $\mathbf{88.23 \pm 0.18}$ | $\mathbf{96.84 \pm 0.23}$ | $\mathbf{95.74 \pm 0.13}$ |

## B.7 ADDITIONAL ABLATION STUDIES

To observe the contribution of different components and their behavior under different non-IID settings, we perform a set of targeted ablations that isolate each module (combining data and gradient, adaptive optimal clustering, and dual-encoder representation sharing).

### B.7.1 COMBINING DATA- AND GRADIENT-BASED SIMILARITY

To isolate the contribution of combining data- and gradient-based information for similarity computation, we perform an ablation study where, instead of using the combined similarity, we construct the client similarity matrix using either gradient-only or data-only similarity. We run this ablation on the FEDDAG* single-encoder variant (no dual-encoder sharing), so that performance differences can be attributed directly to the similarity design without interference from other techniques. Concretely, we compare: FEDDAG*-Grad (gradient-only similarity), FEDDAG*-Data (data-only similarity), and FEDDAG*-Data+Grad (our combined similarity). We evaluate on CIFAR-10 and SVHN under two different non-IID data distributions: (i) Data Distribution I with 30% label skew and high quantity shift (Dirichlet $\alpha' = 0.25$), and (ii) Data Distribution II with concept shift (see Section 5). The results for these two settings are reported in Table 14 and Table 15, respectively. These results

consistently show that weighted data-based similarity (FEDDAG$^*$-Data) outperforms gradient-only similarity, and that combining data- and gradient-based similarity (FEDDAG$^*$-Data+Grad) yields the best performance in both label+quantity-skew and concept-shift settings.

Table 14: Ablation on combining data and gradient under **Data Distribution I** (label skew 30%, Dirichlet $\alpha' = 0.25$).

| Method | CIFAR-10 | FMNIST | SVHN |
|---|---|---|---|
| FEDDAG$^*$-Grad (gradient-only) | $83.79 \pm 0.50$ | $91.26 \pm 0.38$ | $89.05 \pm 0.19$ |
| FEDDAG$^*$-Data (data-only) | $85.03 \pm 0.33$ | $91.82 \pm 0.25$ | $89.52 \pm 0.21$ |
| **FEDDAG$^*$-Data+Grad (combined)** | $\mathbf{86.95 \pm 0.21}$ | $\mathbf{92.18 \pm 0.15}$ | $\mathbf{90.97 \pm 0.13}$ |

Table 15: Ablation on similarity design under **Data Distribution II** (concept shift).

| Method | CIFAR-10 | FMNIST | SVHN |
|---|---|---|---|
| FEDDAG$^*$-Grad (gradient-only) | $64.52 \pm 0.45$ | $83.88 \pm 0.32$ | $81.17 \pm 0.21$ |
| FEDDAG$^*$-Data (data-only) | $67.10 \pm 0.23$ | $85.74 \pm 0.28$ | $83.15 \pm 0.14$ |
| **FEDDAG$^*$-Data+Grad (combined)** | $\mathbf{67.79 \pm 0.27}$ | $\mathbf{86.03 \pm 0.21}$ | $\mathbf{83.73 \pm 0.19}$ |

### B.7.2 DUAL-ENCODER REPRESENTATION SHARING

Here, we examine whether the accuracy gains from inter-cluster global representation sharing (GRS) via the dual-encoder architecture (see §4) arise from genuine feature enrichment or simply from increased model capacity. We already performed this ablation on Data Distribution I (20% label skew, Dirichlet $\alpha' = 0.25$; see Table 3 in §5). To further observe the behavior of dual-encoder sharing under a different non-IID regime, we repeat this ablation in the *concept-shift* setting (see Data Distribution II in §5). Using the same three variants of FEDDAG as in Table 3, we obtain the following results as shown in Table 16. Similar to the experiment under Data Distribution I (label-skew setting), full FEDDAG achieves clear accuracy gains over both the single-encoder (FEDDAG$^*$) and the no-sharing dual-encoder (FEDDAG$^\dagger$) variants across all three datasets under concept shift. This indicates that cross-cluster representation sharing remains beneficial even when the primary challenge is a mismatch in local decision boundaries rather than pure label skew.

Table 16: Ablation of cross-cluster representation sharing under concept shift (Data Distribution II in §5), comparing the single-encoder baseline (FEDDAG$^*$), a dual-encoder variant without cross-cluster sharing (FEDDAG$^\dagger$), and full FEDDAG.

| Algorithm | CIFAR-10 | FMNIST | SVHN |
|---|---|---|---|
| FEDDAG$^*$ (single encoder) | $67.79 \pm 0.27$ | $86.03 \pm 0.21$ | $83.73 \pm 0.19$ |
| FEDDAG$^\dagger$ (dual encoder, no sharing) | $67.51 \pm 0.22$ | $85.91 \pm 0.15$ | $83.65 \pm 0.24$ |
| **FEDDAG** (dual encoder + sharing) | $\mathbf{69.13 \pm 0.23}$ | $\mathbf{88.79 \pm 0.19}$ | $\mathbf{85.06 \pm 0.26}$ |

### B.8 EVALUATION OF THE ALTERNATIVE DUAL-ENCODER INITIALIZATION

We empirically evaluate the alternative dual-encoder initialization and diversity-regularization strategy described in §A.11. In this variant, both encoder parameter sets are initialized randomly, and the local primary-encoder objective is augmented with the diversity regularizer to encourage complementary feature extraction. To assess its effectiveness, we compare this variant against the warm-start initialization used in the main FEDDAG pipeline.

**Experiment setup and results.** We evaluate on CIFAR-10 and SVHN under the same 20% label-skew configuration with Dirichlet parameter $\alpha' = 1.0$ used in Data Distribution I (Section 5). Each experiment is repeated across three random seeds, and we report the mean and standard deviation of the final test accuracy. The diversity-regularized variant achieves accuracy comparable to the main (warm-start) version of FEDDAG on both datasets, with slight improvements in certain cases. These results suggest that while warm-start initialization is effective, the random-init + regularization approach also provides a competitive—and in some settings slightly stronger alternative.

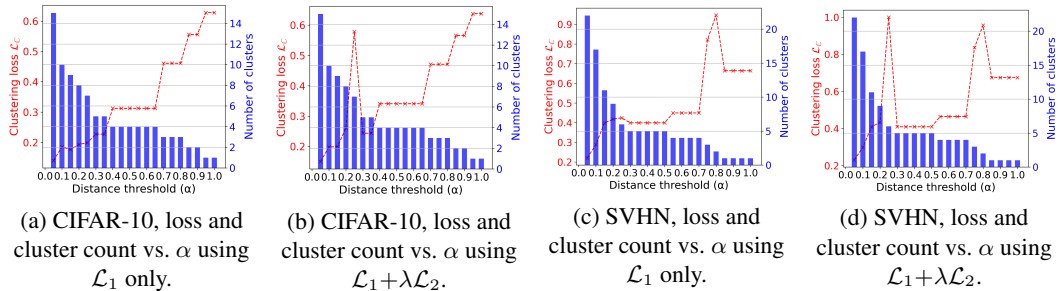

| (a) CIFAR-10, loss and cluster count vs. $\alpha$ using $\mathcal{L}_1$ only. | (b) CIFAR-10, loss and cluster count vs. $\alpha$ using $\mathcal{L}_1 + \lambda\mathcal{L}_2$. | (c) SVHN, loss and cluster count vs. $\alpha$ using $\mathcal{L}_1$ only. | (d) SVHN, loss and cluster count vs. $\alpha$ using $\mathcal{L}_1 + \lambda\mathcal{L}_2$. |

Figure 4: Effect of the federated-aware clustering loss on the adaptive clustering mechanism under Data Distribution I (30% label skew, Dirichlet $\alpha' = 0.25$) for CIFAR-10 and SVHN.

Table 17: Comparison of dual-encoder initialization strategies in FEDDAG (Data Distribution I).

| Encoder Initialization Method | CIFAR-10 | FMNIST | SVHN |
|---|---|---|---|
| Main (warm-start) (see §4) | $90.76 \pm 0.12$ | $93.82 \pm 0.20$ | $93.91 \pm 0.23$ |
| Random init+regularizer (§A.11) | $90.93 \pm 0.16$ | $93.76 \pm 0.17$ | $93.86 \pm 0.10$ |

## B.9 SECONDARY-ENCODER SCHEDULING UNDER RESOURCE CONSTRAINTS

The dual-encoder design in FEDDAG doubles the number of encoder parameters compared to a single-encoder model and introduces extra communication for secondary-encoder updates. While our ablation on FEDDAG* (single encoder) (see 5) shows that the dual-encoder architecture is beneficial for robustness, it is important to understand the trade-off between accuracy and the additional computation/communication overhead, especially for resource-constrained edge devices.

To this end, we evaluate lighter-weight training schedules that *throttle* secondary-encoder updates to reduce computation overhead.. Let a $K{:}1$ *schedule* denote a pattern where we perform $K$ rounds of standard FEDDAG primary training (updating the primary encoder and classifier) followed by one secondary-encoder enrichment round (updating only the secondary encoder). The original FEDDAG corresponds to updating both encoders every round, i.e., no throttling of secondary updates.

We evaluate four primary:secondary scheduling patterns—1:1, 5:1, 10:1, and 15:1—under a fixed compute/communication budget of 80 communication rounds. All experiments are conducted on CIFAR-10 and SVHN under the same heterogeneous setting as Data Distribution I (30% label skew, Dirichlet $\alpha'$=0.25). The results are summarized in Table 18.

Table 18: Accuracy of FEDDAG under different primary:secondary training scheduling patterns (Data Distribution I, 80 comm. rounds). "Full FEDDAG" updates both encoders every round.

| Schedule (Primary : Secondary) | CIFAR-10 | SVHN |
|---|---|---|
| Full FEDDAG (**both**) | $\mathbf{89.87 \pm 0.19}$ | $\mathbf{92.65 \pm 0.11}$ |
| 1 : 1 | $80.34 \pm 0.44$ | $86.26 \pm 0.32$ |
| 5 : 1 | $87.65 \pm 0.28$ | $91.31 \pm 0.22$ |
| 10 : 1 | $87.25 \pm 0.31$ | $91.46 \pm 0.20$ |
| 15 : 1 | $86.54 \pm 0.35$ | $90.87 \pm 0.27$ |

The 1:1 schedule significantly underperforms because the primary encoder is updated only every other round and thus remains under-trained. In contrast, the 5:1 schedule provides the best trade-off: it reduces the number of secondary-encoder updates by 80% while maintaining accuracy close to the full FEDDAG model on both datasets. The 10:1 and 15:1 schedules further reduce the number of secondary updates but incur a slightly larger accuracy drop. Overall, these results show that a modest throttling of secondary-encoder updates (e.g., 5:1) can substantially reduce the effective compute and communication devoted to the secondary encoder while preserving most of the dual-encoder performance gains.

## B.10 EVALUATING THE OPTIMAL CLUSTERING MECHANISM

To understand the efficacy of the optimal clustering mechanism (see §3.4) in FEDDAG, we examine its behavior when the number of *true* underlying data distributions is large. To do this, we design an experiment where the inherent number of clusters is intentionally high. The experiment is set up and performed as follows:

**Dataset setup.** We use the SVHN and CIFAR-10 datasets to construct a federated setting with an inherently large number of ground-truth clusters. Each dataset has 10 classes; any pair of classes defines a possible two-class distribution, yielding a total of $\binom{10}{2} = 45$ distinct two-class combinations. From these 45 possibilities, we select *11 distinct two-class combinations* to create a controlled setting with **11 ground-truth clusters** (e.g., class pairs $(1, 3)$, $(2, 8)$, etc.).

Each client is assigned exactly one of these 11 two-class combinations. For each chosen pair of classes, we collect all corresponding samples and distribute them across the assigned clients using a Dirichlet sampler, which introduces within-cluster heterogeneity while preserving the underlying two-class structure. As a result, each client contains data from *exactly two* classes, while the overall population spans **11 distinct underlying distributions**. This setting allows us to test whether FEDDAG can automatically discover a relatively large number of true clusters.

**Experiment on optimal clustering procedure.** After FEDDAG computes adjacency matrix, we run hierarchical clustering over a grid of distance $\alpha$ and evaluate our clustering loss $\mathcal{L}_{\mathbb{C}}$. We start from $\alpha = 1.0$ and decrease $\alpha$ in steps of 0.05. For each value of $\alpha$, we record (i) the resulting number of clusters and (ii) the corresponding clustering loss $\mathcal{L}_{\mathbb{C}}$. The results for optimal clustering on CIFAR-10 and SVHN are summarized jointly in Table 19 and visualized in Figure 5.

Table 19: Number of clusters and clustering loss $\mathcal{L}_{\mathbb{C}}$ as a function of the distance threshold $\alpha$ for CIFAR-10 and SVHN in the inherent high-cluster (i.e., 11) distribution setting.

| | CIFAR-10 | | SVHN | | | CIFAR-10 | | SVHN | |
|---|---|---|---|---|---|---|---|---|---|
| $\alpha$ | #clusters | $\mathcal{L}_{\mathbb{C}}$ | #clusters | $\mathcal{L}_{\mathbb{C}}$ | $\alpha$ | #clusters | $\mathcal{L}_{\mathbb{C}}$ | #clusters | $\mathcal{L}_{\mathbb{C}}$ |
| 1.000 | 1 | 0.770 | 1 | 0.675 | **0.500** | **11** | **0.149** | 5 | 1.000 |
| 0.950 | 1 | 0.770 | 1 | 0.675 | **0.450** | **11** | **0.149** | 7 | 0.896 |
| 0.900 | 2 | 1.000 | 1 | 0.675 | **0.400** | **11** | **0.149** | 9 | 0.688 |
| 0.850 | 3 | 1.000 | 1 | 0.675 | **0.350** | **11** | **0.149** | **11** | **0.343** |
| 0.800 | 3 | 1.000 | 2 | 0.866 | **0.300** | **11** | **0.149** | **11** | **0.343** |
| 0.750 | 3 | 1.000 | 2 | 0.866 | **0.250** | **11** | **0.149** | **11** | **0.343** |
| 0.700 | 4 | 1.000 | 3 | 1.000 | 0.200 | 12 | 0.251 | **11** | **0.343** |
| 0.650 | 4 | 1.000 | 4 | 1.000 | 0.150 | 13 | 0.161 | 14 | 0.315 |
| 0.600 | 5 | 1.000 | 4 | 1.000 | 0.100 | 14 | 0.117 | 18 | 0.205 |
| 0.550 | 6 | 1.000 | 5 | 1.000 | 0.050 | 14 | 0.117 | 24 | 0.141 |

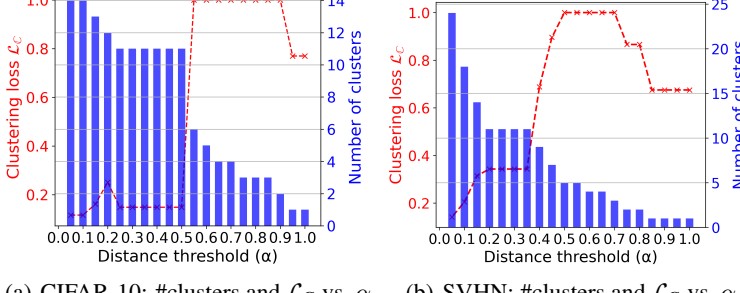

(a) CIFAR-10: #clusters and $\mathcal{L}_{\mathbb{C}}$ vs. $\alpha$    (b) SVHN: #clusters and $\mathcal{L}_{\mathbb{C}}$ vs. $\alpha$

Figure 5: Behavior of FEDDAG's adaptive clustering mechanism in a setting with inherent high number of clusters (e.g., $> 10$) ground-truth distributions.

For CIFAR-10, as $\alpha$ decreases from 1.0, the clustering initially remains extremely coarse (between 1 and 6 clusters), and the loss stays high and saturated at 1.0, reflecting severe *under-clustering*, where

many heterogeneous clients are incorrectly merged. A clear transition occurs around $\alpha \approx 0.50$: the number of clusters jumps to 11, and the loss drops sharply from 1.0 to approximately 0.15. Importantly, this *11-cluster solution forms a stable plateau* across a wide threshold range $\alpha \in [0.25, 0.50]$, with both the cluster count and the loss remaining effectively unchanged. Lowering $\alpha$ below 0.25 further fragments clusters into 12–14 smaller groups, but the loss only improves marginally (from $\approx 0.15$ to $\approx 0.12$). We interpret this as *over-segmentation* rather than revealing additional meaningful structure. In contrast, the 11-cluster configuration dominates across a broad $\alpha$ interval and matches the true number of underlying distributions. For SVHN, we observe a similar trend. At large $\alpha$ (between 0.85 and 1.0) the solution is clearly under-clustered (1–2 clusters) with high loss. As $\alpha$ decreases, an 11-cluster configuration emerges and remains stable for $\alpha \in [0.20, 0.35]$ with loss around 0.34. Pushing $\alpha$ below 0.20 further splits clusters (14–24 clusters) and only slightly reduces the loss (down to $\approx 0.14$), again indicating over-segmentation rather than meaningful clusters.

Overall, these experiments show that FEDDAG's adaptive clustering mechanism *scales effectively to scenarios with more than 10 true distributions* on both CIFAR-10 and SVHN, and can recover the correct number of underlying clusters (here, $K^\star = 11$) without ever hard-coding $K$ into FEDDAG.

