# OpenReview forum: "FedDAG: Clustered Federated Learning via Global Data and Gradient Integration for Heterogeneous Environments"
_ICLR.cc/2026/Conference — ICLR 2026 Poster_

### Official Review · Reviewer_8htn · 2025-10-25

**Soundness:** 3
**Presentation:** 3
**Contribution:** 4
**Rating:** 4
**Confidence:** 4

**Summary:**

This work proposes a novel clustering method for federated learning, FedDAG. FedDAG combines data and gradient information to cluster clients more effectively. FEDDAG leverages cross-cluster representation sharing and incorporates an efficient mechanism to automatically determine the optimal number of clusters. Experiments are conducted to evaluate the performance of the proposed mechanism.

**Strengths:**

1. The structure of this paper is clear and reasonable.

2. The paper is easy to follow.

3. Experiments show that the performance of the proposed method is significantly better than existing methods in different scenarios.

**Weaknesses:**

1. The literature review section should be expanded to include more complete related work and discuss several related subtopics.

2. Directly reporting the class sample count will leak sensitive sample distribution information.

3. The novelty of the dual encoder structure is limited, similar to the personalized/shared head of pFL works and the personalized/shared feature extractor of feature-skewed FL works.

4. Although the author claims that FEDDAG is the only model that can solve all four data heterogeneity problems, the paper lacks experiments on feature distribution shifts. Commonly used datasets like Office-Caltech-10, PACS, and DomainNet should be studied.

5. The baseline is mainly compared with clustered FL methods. It should also be compared with FL methods that address different heterogeneity issues.

6. The layout of the paper can be improved. For example, the experimental results in Table 2 and Figure 1 are too far away from the experimental section; the method diagram in Figure 2 could be placed in the main text.

**Questions:**

See weaknesses. I would like to change my score if the authors could address the concerns.

---

> ### Author Response · Authors · 2025-11-24
> **Comment by authors addressing the questions - Part 1**
>
> **W1.**  We thank the reviewer for this helpful suggestion. In the revised manuscript, we have expanded the literature review to include additional subtopics relevant to our work, including clustered FL, knowledge distillation, and meta-learning. The updated text (now in Section 2) will read as follows:
>
>
>
>
> ---
>
>
> ### Literature review
> There exists an extensive body of work on improving the performance of FL in data-heterogeneous environments through clustered FL, knowledge distillation, meta-learning, client selection, data augmentation, and other techniques. Below, we summarize the approaches that are most relevant to our work.
>
>
>
>
> **Clustered FL** techniques address distribution shift by grouping clients based on their data distributions. PACFL (Vahidian et al., 2023) clusters clients by analyzing principal angles between client data subspaces, but it ignores label information, making it prone to incorrect clustering under concept shift. Ding and Wang (2022) construct \(K\) shared models based on each client’s dataset contribution. Another line of work (Ghosh et al., 2020; Licciardi et al., 2025) uses loss values on gradients to iteratively cluster clients each round. Other methods group clients based on gradient or parameter similarity (Sattler et al., 2020; Zhang et al., 2024), while soft-clustering approaches allow clients to join multiple clusters (Ruan and Joe-Wong, 2022; Guo et al., 2024). Additional methods, such as Long et al. (2023), Marfoq et al. (2021), and Wu et al. (2023) rely on maximizing log-likelihood functions or modeling joint distributions. Compared to these methods, FedDAG combines data and gradient information for improved clustering and further enables knowledge sharing across clusters.
>
>
>
>
> **Knowledge distillation (KD)** approaches, such as Lin et al. (2020) and Li and Wang (2019), use a global dataset to transfer knowledge from local teacher models to a global student model. FedFTG (Zhang et al., 2022b) trains a generator to approximate the input space of local models and uses it to generate pseudo-data. Another line of work, data-free KD, generates pseudo-data directly from a pretrained teacher model (Guo et al., 2023; Fang et al., 2019; Chen et al., 2019). DeepImpression (Nayak et al., 2019) reconstructs approximate real data by modeling the output space of the teacher model, while DeepInversion (Yin et al., 2020) refines pseudo-data by regularizing intermediate feature distributions. Unlike these methods, FedDAG updates its global parameters using data from complementary source clusters, enabling cross-cluster knowledge sharing without relying on public or synthetic datasets.
>
>
>
>
> **Meta-learning approaches** include personalized FL (Arivazhagan et al., 2019; Liang et al., 2020; Li et al., 2024b) and model-regularization methods (Li et al., 2021a; T. Dinh et al., 2020; Karimireddy et al., 2020). Per-FedAvg (Fallah et al., 2020) provides a personalized variant of FedAvg based on the MAML framework. FedPer (Arivazhagan et al., 2019) and FedRep (Collins et al., 2021) split the backbone into a feature extractor and a head to share feature information, while FedRoD (Chen and Chao, 2021) maintains a shared extractor and two classification heads. FedSimSup (Liu et al., 2025) employs a local supervisor and data-similarity-weighted inter-learning model to better align global knowledge with heterogeneous local data. Our method, FedDAG, preserves global–personal decoupling via a dual-encoder architecture in which global parameters are updated only by a selected subset of clients, rather than influenced by all clients.
>
> ---
>
>
> We have also included additional FL subtopics, such as data-augmentation FL, in Appendix A.1 Additional Related Works of the paper, which is written as follows:
>
>
> **Data Augmentation**–based FL techniques propose sharing a small global dataset among clients and combining it with their local datasets to mitigate heterogeneity (Tuor et al., 2021; Xin et al., 2020). Approaches such as Xin et al. (2020), Yoon et al. (2021), and Guo et al. (2023) employ generative adversarial networks (GANs) or averaged local data (Random Sample Mean) to create privacy-preserving pseudo-data that is used to reduce bias in client models. Astraea (Duan et al., 2019) constructs a globally balanced data distribution by performing local augmentation across participating clients. VHL (Tang et al., 2022) creates virtual samples from noise shared across clients to regularize training by aligning local feature representations with these virtual data. Unlike data-augmentation–based FL methods that share or synthesize auxiliary samples, FedDAG mitigates heterogeneity by training a secondary encoder on other clusters’ data within the clustered FL framework.

---

> > ### Author Response · Authors · 2025-11-24
> > **Comment by authors addressing the questions - Part 2**
> >
> > **W2.**  We thank the reviewer for raising the concern regarding whether directly reporting per-class sample counts may leak sensitive information. We address this by clarifying how class-frequency information can be protected—or approximated without revealing exact values. The following discussion has been added to **Appendix A.6 (Privacy Considerations)**.
> >
> > ---
> > ### Appendix A.6 Privacy Considerations
> > Privacy mechanisms can be used to protect the class-frequency information that we use when weighting similarity values. For example, FLIPS (Bhope et al., 2023) employs a trusted execution environment (TEE) to safeguard label distributions, which are used to select a diverse set of clients during FL training. Similarly, FedDAG could employ differential privacy (DP) (Dwork et al., 2006) by perturbing class counts with carefully calibrated noise such that the contribution of any single example is statistically hidden. Another option is homomorphic encryption (HE) (Gentry, 2009), which would allow the server to compute weights directly on ciphertexts without accessing raw values. Designing and integrating such mechanisms is beyond the scope of the present work, so we do not pursue them further.
> >
> >
> > [1] Bhope, Rahul Atul, et al. "FLIPS: federated learning using intelligent participant selection." Proceedings of the 24th International Middleware Conference. 2023.
> >
> > **W3.** We thank the reviewer for raising this concern. We agree that the idea of decoupling a model into two components is not new by itself; however, how these encoders are trained and what they represent in FedDAG is fundamentally different. Most pFL and feature-skew FL works use a decomposition of the form:
> >
> >
> >
> >
> > - **Global/shared part:** a shared feature extractor or shared head
> > - **Personalized part:** a per-client feature extractor or a per-client head
> >
> >
> >
> >
> > So their effective decomposition is:
> >
> >
> >
> >
> > > (shared feature across all clients) + (private feature per client)
> >
> >
> >
> >
> > ---
> >
> >
> >
> >
> > ## How FedDAG differs
> >
> >
> >
> >
> > ### 1. Global feature vs. selective complementary features
> >
> >
> >
> >
> > In these pFL approaches, sharing a single global feature extractor can be suboptimal. The global feature space is influenced by all clients, including:
> >
> >
> >
> >
> > - clients with very different learning objectives or data distributions, and
> > - clients with poor or unstable features (e.g., very few samples for certain classes)
> >
> >
> >
> >
> > This mixture can lead to negative and noisy feature transfer: aligning a representation with *all* clients is often harmful, especially when only a subset of clients are truly relevant or complementary.
> >
> >
> >
> >
> > In contrast,  FedDAG does not simply blend all clients into one global representation. Instead, it:
> >
> >
> >
> >
> > - uses the CC-Graph to identify which clusters are actually beneficial (i.e., have complementary class distributions and sufficient data), and
> > - imports features only from those complementary clusters via the secondary encoder.
> >
> >
> >
> >
> > In this sense,  FedDAG factorizes what would otherwise be a monolithic global feature space into:
> >
> >
> >
> >
> > > (cluster-specific personal features) + (selected complementary features from other clusters)
> >
> >
> >
> >
> > thereby avoiding contamination from unrelated or comparatively less relevant client sources.
> >
> >
> > ---
> > ### 2. Secondary encoder trained directly on other clusters’ data
> >
> >
> >
> >
> > In FedDAG, the secondary encoder of the learner cluster is explicitly trained on the datasets of clients from source clusters:
> >
> >
> >
> >
> > - The secondary encoder parameters are sent to a source cluster identified by the CC-Graph.
> > - At the source cluster, that encoder is locally trained on the clients’ data, and the resulting gradients or updates are then sent back to the learner clusters.
> >
> >
> >
> >
> > Thus, the secondary encoder acts as a parameter carrier that travels across clusters, learns on other clusters’ data, and then returns with an updated gradient.
> >
> >
> >
> >
> > This is fundamentally different from a “shared head” or a single global feature extractor in feature-skew FL. Those approaches typically rely on shared backbones plus local heads, or on knowledge-transfer signals such as pseudo-data, distilled logits, or prototype features; to our knowledge, they do **not** perform explicit remote training of a model component on other clients’ local data. This limits their ability to fully exploit the complementary feature structure present in other domains.
> >
> >
> >
> >
> > We also empirically verify in our ablation studies (Section 5, Ablation studies) that the gain is not merely due to the increased parameter size of adding a second encoder.
> >
> >
> >
> >
> > ---
> >
> >
> >
> >
> > By leveraging the structure and training workflow of clustered federated learning, FedDAG’s secondary encoder performs *targeted cross-cluster feature extraction* guided by the CC-Graph, rather than functioning as just another shared or personalized feature extractor. This cross-cluster training mechanism, combined with the CC-Graph, constitutes the main novelty of our dual-encoder design relative to existing pFL and feature-skew FL works.

---

> > > ### Author Response · Authors · 2025-11-24
> > > **Comment by authors addressing the questions - Part 3**
> > >
> > > **W4.** We thank the reviewer for highlighting the importance of evaluating feature distribution shift and for pointing us to standard benchmarks such as Office-Caltech-10, PACS, and DomainNet. We would like to clarify that our original submission **already evaluates FedDAG under feature-skew settings** in **Appendix B.5 (Data Distribution IV)**, and, following the reviewer’s suggestion, we have additionally incorporated two new feature-shift benchmarks (PACS and Office-Caltech-10). Below, we summarize what was included in the initial submission and what we added during the rebuttal period.
> > >
> > >
> > > **Initial submission (corruption-based feature skew).**
> > > As described in the initial submission, Appendix B.5 (Data Distribution IV), we used CIFAR-10-C and TinyImageNet-C, each containing 20 corruption types (e.g., fog, contrast, blur, etc.), following the feature-corruption protocol of FedRC [1]. During data partitioning, we (i) assign a single corruption type to each client, and (ii) within that corruption, distribute samples across classes to clients using a Dirichlet distribution. We randomly select 80% of the clients for training and keep the remaining 20% as unseen clients. Our reported metric is the test accuracy on these 20% held-out clients using the final trained FedDAG model. Some representative results are:
> > >
> > >
> > > | Algorithm  | CIFAR-10-C (Acc. %) | TinyImageNet-C (Acc. %) |
> > > |-----------|---------------------|-------------------------|
> > > | FedAvg    | 30.73 ± 0.36        | 18.43 ± 0.43            |
> > > | PACFL     | 63.62 ± 0.22        | 33.53 ± 0.38            |
> > > | IFCA      | 62.52 ± 0.39        | 32.33 ± 0.19            |
> > > | FedDAG    | 65.62 ± 0.31        | 36.27 ± 0.32            |
> > >
> > >
> > > **New addition during rebuttal (domain-level feature shift).**
> > > In response to the reviewer’s suggestion, we additionally evaluate FedDAG on PACS and Office-Caltech-10. Similar to the previous setup, we treat each domain as a distinct feature distribution: we assign one domain (out of four) to each client and, within that domain, distribute samples to clients using a Dirichlet distribution. We again train on 80% of the clients and test on the remaining 20% unseen clients, reporting their test accuracy as the final metric. The results are summarized below. The results show that FedDAG achieves the best accuracy on PACS and Office-Caltech-10 and, together with the corruption-based results, indicate consistent gains across all four datasets.
> > >
> > >
> > >
> > > | Algorithm  | PACS (Acc. %)      | Office-Caltech-10 (Acc. %) |
> > > |-----------|---------------------|-----------------------------|
> > > | FedAvg    | 38.47 ± 0.28        | 46.12 ± 0.31                |
> > > | PerFedAvg | 70.82 ± 0.22        | 67.34 ± 0.23                |
> > > | PACFL     | 76.63 ± 0.19        | 73.31 ± 0.14                |
> > > | CFL       | 72.14 ± 0.25        | 68.02 ± 0.19                |
> > > | FedRC     | 76.28 ± 0.17        | 73.54 ± 0.22                |
> > > | IFCA      | 75.12 ± 0.21        | 72.48 ± 0.29                |
> > > | FedDAG    | 80.34 ± 0.27        | 76.28 ± 0.20                |
> > >
> > >
> > >
> > >
> > > **Regarding DomainNet.**
> > > Based on your suggestion, we have added experiments on PACS and Office-Caltech-10. In our existing feature-skew setup, we already evaluate two datasets (CIFAR-10-C and Tiny ImageNet-C). For DomainNet, we have not completed the experiments yet; we are actively working on this evaluation and will include the results in a future version of the paper.
> > >
> > >
> > >
> > >
> > > We have expanded the description in Appendix B.5 to incorporate these details and add the new PACS and Office-Caltech-10 results, thereby strengthening our empirical evidence that FedDAG remains robust under diverse forms of feature distribution shift.
> > >
> > >
> > > [1] Guo, Yongxin, Xiaoying Tang, and Tao Lin. “FedRC: Tackling diverse distribution shifts challenge in federated learning by robust clustering.” arXiv preprint arXiv:2301.12379 (2023).

---

> > > > ### Author Response · Authors · 2025-11-24
> > > > **Comment by authors addressing the questions - Part 4**
> > > >
> > > > **W5.** We thank the reviewer for pointing out that our baselines mainly focused on clustered FL methods. In response, we additionally compare FedDAG with two methods designed for heterogeneous federated settings: **FedMix** [1] and **FedBR** [2].
> > > >
> > > > FedMix [1] generates privacy-preserving augmented data by averaging local batches into mean representations and then applying Mixup-style interpolation during local updates, which improves robustness under heterogeneous client distributions without sharing raw data. FedBR [2] reduces local learning bias by balancing classifier outputs on label-agnostic pseudo-data and using a min–max contrastive loss between **local** and **global** feature extractors on each client, encouraging local features to align with global representations instead of overfitting to client-specific patterns.
> > > >
> > > > We evaluate FedMix and FedBR on **Data Distribution I** with **20%** and **30% non-IID label skew** under **low quantity shift** (Dirichlet α′ = 0.25), using the same experimental protocol as in our main experiments. The results are summarized below (we integrated the final numbers into the corresponding tables in the revised paper, to be uploaded within a short period):
> > > >
> > > > ### 20% Label Skew
> > > >
> > > > | Algorithm | CIFAR-10 (Acc. %)    | FMNIST (Acc. %)      | SVHN (Acc. %)        | CIFAR-100 (Acc. %)    |
> > > > |-----------|----------------------|----------------------|----------------------|-----------------------|
> > > > | FedMix    | 77.94 ± 0.26         | 83.55 ± 0.31         | 83.12 ± 0.29         | 60.33 ± 0.24          |
> > > > | FedBR     | 81.62 ± 0.28         | 85.32 ± 0.23         | 84.05 ± 0.30         | 61.61 ± 0.37          |
> > > > | **FedDAG**| **90.76 ± 0.12**     | **93.82 ± 0.20**     | **93.91 ± 0.23**     | **72.84 ± 0.30**      |
> > > >
> > > >
> > > > ### 30% Label Skew
> > > >
> > > > | Algorithm | CIFAR-10 (Acc. %)    | FMNIST (Acc. %)      | SVHN (Acc. %)        | CIFAR-100 (Acc. %)    |
> > > > |-----------|----------------------|----------------------|----------------------|-----------------------|
> > > > | FedMix    | 76.90 ± 0.33         | 81.96 ± 0.27         | 82.21 ± 0.34         | 53.55 ± 0.30          |
> > > > | FedBR     | 81.48 ± 0.37         | 84.12 ± 0.25         | 84.78 ± 0.38         | 56.32 ± 0.33          |
> > > > | **FedDAG**| **89.87 ± 0.19**     | **92.72 ± 0.13**     | **92.65 ± 0.11**     | **63.21 ± 0.60**      |
> > > >
> > > >
> > > >
> > > >
> > > >
> > > > These additional results have been added to the experimental section in the final version, demonstrating that FedDAG remains competitive even against non-clustered methods specifically designed to address heterogeneous data.
> > > >
> > > > [1] Yoon, T., Shin, S., Hwang, S. J., and Yang, E. “FedMix: Approximation of Mixup under mean augmented federated learning.” 2021b.
> > > > [2] Guo, Y., Tang, X., and Lin, T. “FedBR: Improving federated learning on heterogeneous data via local learning bias reduction.” ICML, 2023.
> > > >
> > > >
> > > > **W6.** We thank the reviewer for this helpful suggestion regarding the paper’s layout. We have accommodated these changes in the updated version of the paper, which will be uploaded within a short period. Specifically, we relocated **Table 2** and **Figure 1** to appear directly within the experimental section, and we moved **Figure 2** into the main body of the paper to improve readability and flow.

---

> > > > > ### Comment · Reviewer_8htn · 2025-11-25
> > > > >
> > > > > I appreciate the authors' responses. Most of my concerns have been addressed. I have the following concerns that I hope the authors can address:
> > > > >
> > > > > 1. I couldn't find the revision of the paper, but only the original PDF. Could you please update it?
> > > > >
> > > > > 2. Regarding W5, my issue is that the authors used to compare FedDAG with cluster FL methods. However, they claim that FedDAG is effective for four types of shifts: label skew, feature skew, concept shift, and quantity shift. I believe that at least one tailored FL approach is needed as a baseline for comparison for each shift (or combined). Thank you for providing FedMix and FedBR, but I'm not sure which ones they correspond to. If time permits, I hope the authors can further address this concern.

---

> > > > > > ### Author Response · Authors · 2025-12-03
> > > > > > **Comment by authors addressing the questions - Part 5**
> > > > > >
> > > > > > **1.** We have submitted the updated file. We apologize for the delay as we were working to incorporate updates from all the reviewers.
> > > > > >
> > > > > > ---
> > > > > >
> > > > > > **2.** We thank the reviewer for emphasizing the importance of comparing FedDAG against tailored FL approaches for each type of distribution shift. As noted in our previous response, our newly added baselines (FedBR [1] and FedMix [2]) were originally designed and applied primarily under label-skew settings. To more directly address the reviewer’s concern, we have now conducted additional experiments and clarified how our baselines correspond to the four major shift types—label skew, feature skew, concept shift, and quantity shift. We also summarize the associated empirical comparisons.
> > > > > >
> > > > > > Most of our clustered FL baselines — PACFL, FedSoft, FedGWC, CFL, CFL-GP, and IFCA — adopt Dirichlet-based **label skew** in their original works and are specifically proposed to handle heterogeneous label distributions across clients. In our experiments in Section 5 (Data Distributions I/II), we follow this standard setting and compare FedDAG against these methods under label-skewed clients.
> > > > > >
> > > > > > For our **feature-skew** experiments in Appendix B.4 (Data Distribution IV), we incorporate two additional tailored baselines in direct response to the reviewer’s suggestion. **FedBR** [1] uses pseudo-data to reduce bias in local features and classifiers and to better align local and global feature representations. FedBR is explicitly evaluated on **PACS**, a standard domain-adaptation / feature-shift benchmark. **FedMix** [2] employs a federated mixup-style strategy to mitigate non-IID effects by sharing mean-augmented information, and is designed to address covariate/feature skew as well as label skew. Among our existing baselines, **FedRC** also includes explicit experiments under a feature-skew setting using **CIFAR-10-C** and **Tiny ImageNet-C**. Results from these two new baselines, along with the existing one, are presented below.
> > > > > >
> > > > > > In our feature-skew experiment (Appendix B.4, Data Distribution IV), we partition and distribute samples across clients using a Dirichlet distribution, which naturally induces **quantity shift** (i.e., varying client sample sizes) in addition to feature skew. To more directly compare against approaches specialized for quantity shift, we also include a new dedicated baseline, **FBLG** [3], which employs a client-selection policy that prioritizes clients with larger local datasets while grouping clients with similar sizes. The results for FBLG on these distributions are also presented below.
> > > > > >
> > > > > > For **concept shift**, we include experiments under a concept-shift setting in Section 5 against several baselines. One of our baselines, **FedRC**, states that it tackles combinations of different distribution shifts, and it includes experiments on concept shift in combination with other types of skew. In addition, we now add a comparison to the newly included baseline FedBR, which uses pseudo-data to reduce bias and address multiple forms of heterogeneity.
> > > > > >
> > > > > > [1] Yoon, T., Shin, S., Hwang, S. J., and Yang, E. “FedMix: Approximation of Mixup under mean augmented federated learning.” 2021b.
> > > > > >
> > > > > > [2] Guo, Y., Tang, X., and Lin, T. “FedBR: Improving federated learning on heterogeneous data via local learning bias reduction.” ICML, 2023.
> > > > > >
> > > > > > [3] Xu, Yi, et al. "Fblg: A local graph based approach for handling dual skewed non-iid data in federated learning." Proceedings of the Thirty-Third International Joint Conference on Artificial Intelligence, IJCAI-24, K. Larson, Ed. International Joint Conferences on Artificial Intelligence Organization. Vol. 8. 2024.
> > > > > >
> > > > > >
> > > > > > ...**continued**...

---

> > > > > > > ### Author Response · Authors · 2025-12-03
> > > > > > > **Comment by authors addressing the questions - Part 6**
> > > > > > >
> > > > > > > ...**continued**...
> > > > > > >
> > > > > > > ### Experiment results
> > > > > > > Experiment results for Distribution IV (feature-skew setting with Dirichlet-induced quantity shift), including feature-skew–focused baselines (the newly added FedBR and FedMix, and the existing FedRC) as well as FBLG, a baseline specialized for quantity-shift scenarios, are summarized below.
> > > > > > >
> > > > > > > | Algorithm | CIFAR-10-C        | Tiny ImageNet-C    | PACS              | Office-Caltech-10  |
> > > > > > > |-----------|-------------------|--------------------|-------------------|--------------------|
> > > > > > > | FBLG     | 60.25 ± 0.30      | 29.91 ± 0.23       | 72.38 ± 0.37      | 68.44 ± 0.14       |
> > > > > > > | FedBR     | 61.09 ± 0.32      | 30.83 ± 0.25       | 73.83 ± 0.32      | 71.37 ± 0.18       |
> > > > > > > | FedMix    | 60.11 ± 0.14      | 30.71 ± 0.24       | 73.25 ± 0.41      | 70.56 ± 0.23       |
> > > > > > > | FedRC     | 61.82 ± 0.21      | 32.14 ± 0.19       | 76.28 ± 0.17      | 73.54 ± 0.22       |
> > > > > > > | **FedDAG**| **65.62 ± 0.31**  | **36.27 ± 0.32**   | **80.34 ± 0.27**  | **76.28 ± 0.20**   |
> > > > > > >
> > > > > > > Experiment results for concept-shift performance comparison between FedBR and FedDAG.
> > > > > > > FedBR is the newly added baseline evaluated under the concept-shift setting.
> > > > > > > | Algorithm | CIFAR-10 | FMNIST | SVHN |
> > > > > > > |----------|----------|--------|------|
> > > > > > > | **FedBR** | 62.41 ± 0.28 | 82.81 ± 0.17 | 80.12 ± 0.22 |
> > > > > > > | **FedDAG** | **69.90 ± 0.22** | **88.93 ± 0.18** | **85.34 ± 0.25** |
> > > > > > >
> > > > > > > Across all feature-skew and concept-shift evaluations, FedDAG consistently outperforms all competing methods, often by a substantial margin. These experiments combine feature skew + label skew + quantity shift (Distribution IV) or concept shift + label skew, creating challenging heterogeneous environments in which client distributions differ simultaneously along multiple axes. Under these mixed-shift conditions, cluster-based FL approaches such as FedRC, PACFL, and IFCA generally perform better than the newly added baselines (FedBR and FedMix), as cluster-FL methods explicitly group clients with similar label distributions, which helps mitigate label-skew–induced instability.

---

### Official Review · Reviewer_VAQF · 2025-10-30

**Soundness:** 2
**Presentation:** 3
**Contribution:** 3
**Rating:** 6
**Confidence:** 4

**Summary:**

This paper proposes FedDAG, a novel clustered Federated Learning (FL) algorithm designed to handle diverse types of data heterogeneity, including label skew, feature skew, concept shift, and quantity shift. The core contributions are threefold: (1) a client similarity metric that fuses both data-based (via a weighted, class-wise principal angle comparison) and gradient-based information, with client-specific weights learned via an entropy loss; (2) a dual-encoder architecture that enables cross-cluster representation sharing to enrich feature learning without sacrificing cluster specialization; and (3) an adaptive clustering mechanism that automatically determines the optimal number of clusters using a novel federated-aware metric that penalizes over-splitting. Extensive experiments on multiple datasets and non-IID settings demonstrate that FedDAG consistently outperforms a wide range of state-of-the-art clustered and personalized FL baselines.

**Strengths:**

- Comprehensive Problem Formulation: The paper does an excellent job of identifying and articulating the key limitations of existing clustered FL methods, such as reliance on a single similarity modality (data or gradients), restricted intra-cluster knowledge sharing, and inability to handle all forms of data skew. The motivation is clear and well-justified.

- Novelty and Technical Sophistication: The proposed method is technically sound and introduces several novel ideas. The fusion of data and gradient similarities is not merely a weighted average but involves a learned, client-specific weighting scheme. The class-wise, weighted data similarity is a clear and meaningful improvement over prior work like PACFL. The dual-encoder architecture for cross-cluster sharing is a creative and well-motivated solution to the problem of isolated clusters.

- Thorough Empirical Evaluation: The experimental section is a major strength. The authors evaluate FedDAG across four different non-IID data distributions, encompassing all the heterogeneity types they claim to address. The comparison against a large set of strong baselines is comprehensive. The inclusion of ablation studies (FedDAG*, FedDAG+) effectively isolates the contribution of different components (clustering vs. sharing, and sharing vs. parameter increase).

- Practicality and Completeness: The paper thoughtfully addresses practical concerns such as communication/computation complexity, privacy implications, convergence (with a provided theorem), and dynamic scenarios like newcomer integration and data distribution shift. This makes the work feel mature and applicable to real-world FL systems.

**Weaknesses:**

- Computational and Architectural Overhead: The dual-encoder architecture inherently doubles the parameter count for the feature extractor compared to a single-model approach. While the ablation study (FedDAG†) convincingly shows that the gains are due to sharing and not just more parameters, this overhead is non-trivial for resource-constrained edge devices. The paper mentions the possibility of alternating training phases to mitigate this, but a more detailed discussion on the trade-offs (e.g., how much it slows convergence) would be beneficial.

- Clustering Stability and Cost: The clustering mechanism, while adaptive, is performed initially and then assumed to be stable (Assumption A.4 in the convergence analysis). The method for handling distribution shift (Appendix A.8) is reactive and requires clients to locally re-train, which incurs additional communication and computation cost. The overall one-time cost of the initial clustering phase (local warm-up, SVD, gradient sparsification) is non-negligible, though the paper argues it's a small fraction of total training. A more explicit comparison of this "clustering overhead" against other methods would be helpful.


- Clarity of the CC-Graph and Sharing Mechanism: The process for building the Cluster Complementarity Graph (CC-Graph) and the subsequent secondary encoder training is complex. The description in Section 4, while detailed, is challenging to follow on a first read. A more intuitive explanation of the "demand" and "supply" scores, and a clearer step-by-step walkthrough of the training phases, would improve accessibility.

**Questions:**

- How does the performance of FedDAG scale with an increasing number of clusters? The current experiments show a comparison against methods with a pre-set or automatically determined K. It would be insightful to see how FedDAG's adaptive mechanism performs when the inherent number of true data distributions in the client population is large (e.g., >10).

- In the dual-encoder training, the primary encoder is initialized from the locally pre-trained models to avoid redundancy. Was any other initialization strategy explored? Could randomly initializing both and using a regularization term to encourage diversity be a viable alternative?

- The authors need introduce some more related works based on clustering or data similarity:

   - Collaborative learning by detecting collaboration partners
   - Personalized Federated Learning under Local Supervision
   - Benchmarking data heterogeneity evaluation approaches for personalized federated learning,

---

> ### Author Response · Authors · 2025-11-24
> **Comment by authors addressing the questions - Part 1**
>
> **W1.** We thank the reviewer for raising this important point about computational and architectural overhead. To explore the alternating secondary encoder updates, we are now explicitly evaluating lighter-weight training schedules that trade off dual-encoder benefits against overhead:
> K:1 scheduling: perform (K) consecutive rounds of standard FedDAG primary training and then one secondary-encoder enrichment round with (K) tuned to the edge-device budget.
>
>
> These variants directly address the reviewer’s concern: they retain the architectural dual-encoder capacity (which our ablations show is necessary for robustness), while reducing the effective compute and communication devoted to the secondary encoder. We evaluate four primary:secondary scheduling patterns—1:1, 5:1, 10:1, and 15:1—under a fixed compute/communication budget (80 communication rounds). Here, for example, a 10:1 schedule means we perform 10 rounds updating the primary encoder, followed by 1 round updating the secondary encoder. All experiments are conducted on CIFAR-10 and SVHN under the same heterogeneous setting as Data Distribution~I (30\% label skew, Dirichlet $\alpha'$=0.25). The results are shown below:
>
>
>
>
> ### Secondary-Encoder Scheduling Results
>
>
> | Schedule (Primary : Secondary) | CIFAR-10 (Acc. %)    | SVHN (Acc. %)      |
> |--------------------------------|----------------------|--------------------|
> | **Full FedDAG (both)**  | **89.87 ± 0.19**     | **92.65 ± 0.11**   |
> | **1 : 1**                      | 80.34 ± 0.44         | 86.26 ± 0.32       |
> | **5 : 1**                      | 87.65 ± 0.28         | 91.31 ± 0.22       |
> | **10 : 1**                     | 87.25 ± 0.31         | 91.46 ± 0.20       |
> | **15 : 1**                     | 86.54 ± 0.35         | 90.87 ± 0.27       |
>
>
>
>
> The 1:1 schedule leads to a substantial drop in accuracy because the primary encoder is undertrained. In contrast, the 5:1 schedule provides the best trade-off: it alleviates the computation bottleneck while retaining accuracy that is very close to the full FedDAG model (where both primary and secondary encoders are updated every round). The 10:1 and 15:1 schedules also perform reasonably well, but exhibit a slight degradation in some cases.

---

> > ### Author Response · Authors · 2025-11-24
> > **Comment by authors addressing the questions - Part 2**
> >
> > **W2.**  We appreciate the reviewers’ interest in a clearer discussion of clustering cost. During FedDAG, there are two situations that require clustering: (i) an initial clustering at the start of the algorithm, and (ii) a re-clustering that can be triggered by significant distribution changes. We discuss both situations and their clustering overhead below.
> >
> > In the initial phase, each client runs $t_g$ local warm-up epochs and uploads a $k$-sparse gradient $\tilde{\Delta}_i$ together with a small number of per-class principal vectors $U^i_c$. As shown in Appendix B.2 (Hyperparameter Tuning), $t_g = 2$ rounds of local warm-up training are enough to obtain a gradient signature that can be used to compute client similarity. Importantly, the model is **not reset** after warm-up; clients simply do not participate in federation during this short period. Once clustering is assigned, training proceeds directly from the warmed-up model. The same warmed-up model parameters are also used to initialize the primary encoder during secondary-encoder initialization, so this warm-up is **not an additional or wasted training cost**.  After model warm-up, gradients are transmitted in *sparsified* form (only $1\text{-}2\\%$ of coordinates), which keeps the communication overhead small. For the principal vectors, the number sent is also very small (approximately $1\text{-}2\\%$ of the class data size), so communicating principal vectors $U^i_c$ is negligible compared to transmitting the full model parameters $|\Theta|$. The truncated SVD costs $\\mathcal{O}(FN^2)$ (where $N$ and $F$ are the size and feature of the client’s data) per client, and it is incurred only once during initialization and is amortized over all rounds. **Thus, for a client with $N$ samples and feature dimension $F$, the one-time truncated SVD to compute principal vectors costs $\\mathcal{O}(F N^2)$, and together with $t_g = 2$ local warm-up rounds constitutes the main clustering overhead in FedDAG.**
> >
> > After clustering is fixed, FedDAG does not require iterative clustering at every round. In contrast, approaches such as IFCA, CFL, CFL-GP, and FedSoft re-evaluate clustering at **every** communication round. At each round, the server sends all $K$ cluster models (each with parameters $\theta$) to the participating clients, and each participating client must evaluate all $K$ models locally. If FL training runs for $T$ rounds, this results in roughly $K T$ model evaluations per client and communication of $K T$ parameter vectors. In contrast, FedDAG performs clustering only once unless a drastic distribution shift is detected. Apart from the SVD cost $\mathcal{O}(N^2)$ and $t_g = 2$ rounds of local warm-up, FedDAG does not incur the recurring cost associated with iterative clustering in other clustered FL approaches.
> >
> > Finally, the distribution-shift mechanism in Appendix A.8 is a *deployment-time, infrequent* re-clustering step. Re-clustering is triggered only after a major distribution shift (e.g., a $20\\%$ change in Wasserstein distance). It is not used in our main experiments, and when triggered, it simply re-runs $t_g$ rounds of the same warm-up plus the SVD computation. The computation cost is the same as for the initial clustering, and the warmed-up model is *not* reset; training resumes directly from the updated client state after re-clustering.
> >
> > In the revised paper, we have introduced a separate subsection titled “Clustering overhead” in Appendix Section A, which describes the above aspects as shown above.
> >
> > ---
> >
> > **W3.** We thank the reviewer for this valuable feedback on the clarity of the CC-Graph construction and the secondary encoder training. In the updated version of the paper, which will be uploaded within a short period, we provide (i) a more intuitive explanation of the **demand** and **supply** scores, including a high-level example, and (ii) a clearer step-by-step walkthrough of the CC-Graph construction and secondary-encoder training phases. These revisions substantially improve the accessibility and readability of this part of the method.

---

> > > ### Author Response · Authors · 2025-11-24
> > > **Comment by authors addressing the questions - Part 3**
> > >
> > > **Q1.** To evaluate how FedDAG behaves when the number of *true* underlying data distributions is large, we designed an experiment where the inherent cluster count is high. We used **SVHN** and **CIFAR-10** to perform this experiment by splitting the data to simulate a setting with an inherently high number of clusters (e.g., 11), as described below:
> > >
> > > For these datasets (each with 10 classes), any pair of classes defines a possible two-class distribution, giving $\binom{10}{2} = 45$ total two-class combinations. From these 45 possibilities, we selected **11 distinct two-class combinations** to create a controlled setting with **11 ground-truth clusters** (e.g., classes $\{(1,3)\}$, $\{(2,8)\}$, etc.). Each client was assigned exactly **one of these 11** two-class combinations. For every chosen pair of classes, we collected the corresponding samples and distributed them across the assigned clients using a **Dirichlet sampler**, ensuring within-cluster heterogeneity. As a result, each client contains data from *exactly two* classes, while the full population spans **11 distinct underlying distributions**.
> > >
> > >
> > >
> > > FedDAG was then applied to this client population to assess how well its adaptive clustering mechanism can detect and organize clients according to these 11 latent distributions. Since we cannot include figures in the rebuttal system, we provide the numerical tables below; the updated version will include both the plots and the tables.
> > >
> > > After FedDAG fuses data-based and gradient-based similarities into the client adjacency matrix, we run agglomerative hierarchical clustering over a grid of distance thresholds α and evaluate our clustering loss $\\mathcal{L}\_{\mathbb{C}}$. We begin from $\\alpha = 1.0$ and decrease α in steps of 0.05. For each α, we record the number of clusters and the corresponding loss. The results for **CIFAR-10** are:
> > >
> > > | α     | #clusters | loss  |
> > > |-------|-----------|-------|
> > > | 1.000 | 1         | 0.770 |
> > > | 0.950 | 1         | 0.770 |
> > > | 0.900 | 2         | 1.000 |
> > > | 0.850 | 3         | 1.000 |
> > > | 0.800 | 3         | 1.000 |
> > > | 0.750 | 3         | 1.000 |
> > > | 0.700 | 4         | 1.000 |
> > > | 0.650 | 4         | 1.000 |
> > > | 0.600 | 5         | 1.000 |
> > > | 0.550 | 6         | 1.000 |
> > > | 0.500 | 11        | 0.149 |
> > > | 0.450 | 11        | 0.149 |
> > > | 0.400 | 11        | 0.149 |
> > > | 0.350 | 11        | 0.149 |
> > > | 0.300 | 11        | 0.149 |
> > > | 0.250 | 11        | 0.149 |
> > > | 0.200 | 12        | 0.251 |
> > > | 0.150 | 13        | 0.161 |
> > > | 0.100 | 14        | 0.117 |
> > > | 0.050 | 14        | 0.117 |
> > >
> > >
> > > The corresponding results for **SVHN** are:
> > > | α     | #clusters | loss  |
> > > |-------|-----------|-------|
> > > | 1.000 | 1         | 0.675 |
> > > | 0.950 | 1         | 0.675 |
> > > | 0.900 | 1         | 0.675 |
> > > | 0.850 | 1         | 0.675 |
> > > | 0.800 | 2         | 0.866 |
> > > | 0.750 | 2         | 0.866 |
> > > | 0.700 | 3         | 1.000 |
> > > | 0.650 | 4         | 1.000 |
> > > | 0.600 | 4         | 1.000 |
> > > | 0.550 | 5         | 1.000 |
> > > | 0.500 | 5         | 1.000 |
> > > | 0.450 | 7         | 0.896 |
> > > | 0.400 | 9         | 0.688 |
> > > | 0.350 | 11        | 0.343 |
> > > | 0.300 | 11        | 0.343 |
> > > | 0.250 | 11        | 0.343 |
> > > | 0.200 | 11        | 0.343 |
> > > | 0.150 | 14        | 0.315 |
> > > | 0.100 | 18        | 0.205 |
> > > | 0.050 | 24        | 0.141 |
> > >
> > > For **CIFAR-10**, as α decreases from 1.0, the clustering initially remains extremely coarse (1–6 clusters) and the loss stays high and saturated at 1.0, reflecting severe **under-clustering**, where many heterogeneous clients are incorrectly merged. Around α ≈ 0.9, the loss spikes again, indicating **degenerate over-splitting** (erratic clusters containing very few clients). A clear transition occurs around **α ≈ 0.50**: the number of clusters goes to **11**, and the loss drops sharply from 1.0 to ≈0.15. Importantly, this **11-cluster solution forms a stable plateau** across a wide threshold range **α ∈ [0.25, 0.50]**, with both the cluster count and loss remaining effectively unchanged. Lowering α below 0.25 continues to fragment clusters into 12–14 smaller groups, but the loss only improves marginally (≈0.15 → ≈0.12). We interpret this as **over-segmentation**, not as revealing additional meaningful structure. In contrast, the 11-cluster configuration dominates across a broad α interval and matches the true number of underlying distributions.
> > >
> > > For **SVHN**, we see a similar trend. At large α (0.85–1.0) the solution is under-clustered (1–2 clusters) with high loss; as α decreases, an **11-cluster configuration emerges and remains stable** for **α ∈ [0.20, 0.35]** with loss ≈0.34. Pushing α below 0.20 further splits clusters (14–24 clusters) and only slightly reduces the loss (down to ≈0.14), indicating over-segmentation rather than additional meaningful structure.
> > >
> > >
> > > Overall, these experiments show that FedDAG’s adaptive clustering mechanism **scales effectively to scenarios with more than 10 true distributions** on both CIFAR-10 and SVHN, without ever hard-coding the number of $K$.

---

> > > > ### Author Response · Authors · 2025-11-24
> > > > **Comment by authors addressing the questions - Part 4**
> > > >
> > > > **Q2.** We thank the reviewer for this great insight. To accommodate the reviewer’s suggestion, we define an alternative version of FedDAG that
> > > > (i) initializes *both* encoders at random in each cluster and
> > > > (ii) adds a regularization term that explicitly encourages diversity between the primary and secondary encoders.
> > > > At a high level, we modify only the primary-encoder training phase of FedDAG: the primary encoder (which is updated locally) is trained with an additional diversity penalty that depends on the fixed secondary encoder. The details are given below.
> > > >
> > > > **Primary-encoder training before.**
> > > >
> > > > For a client $i$ in cluster $z$, the dual-encoder prediction is
> > > >
> > > > $$
> > > > F\_z(X\_i) = \\psi(\\phi^{(1)}(X\_i;\\theta^{1f}\_i), \\phi^{(2)}(X\_i;\\Theta^{2f}\_{z(i)}); \\theta^c\_i)
> > > > $$
> > > >
> > > > where $\\phi^{(1)}$ and $\\phi^{(2)}$ are the primary and secondary encoders and $\\psi$ is the classifier.   In the original FedDAG, the primary encoder and classifier are trained locally by minimizing the standard cross-entropy loss
> > > >
> > > > $$
> > > > \\ell\_i(\\theta^{1f}\_i,\\theta^c\_i) = \\mathcal{L}(Y\_i, \\psi(\\phi^{(1)}(X\_i;\\theta^{1f}\_i), \\phi^{(2)}(X\_i;\\Theta^{2f}\_{z(i)}); \\theta^c\_i))
> > > > $$
> > > >
> > > > with gradients taken with respect to $(\\theta^{1f}\_i,\\theta^c\_i)$ only, while $\\Theta^{2f}\_{z(i)}$ is kept fixed during this step.
> > > >
> > > > **Primary-encoder training after adding the diversity regularizer.**
> > > >
> > > > In the “random initialization + diversity regularization” variant, we modify this local objective by augmenting it with a term that penalizes the two encoders for producing overly aligned features. We introduce the diversity regularizer:
> > > >
> > > > $$
> > > > R\_{\\mathrm{div}}(\\theta^{1f}\_i,\\Theta^{2f}\_{z(i)}) = \\mathbb{E}\_{x\\sim D\_i}[(\\cos(\\phi^{(1)}(x;\\theta^{1f}\_i), \\phi^{(2)}(x;\\Theta^{2f}\_{z(i)})))^2]
> > > > $$
> > > >
> > > > where $\\cos(\\cdot,\\cdot)$ denotes cosine similarity between the two encoder feature vectors. High cosine similarity means the two encoders are aligned; minimizing $\\cos^2$ therefore pushes them toward complementary feature directions.
> > > >
> > > > The modified local primary loss after incorporating the regularizer $R\_{\\mathrm{div}}(\\theta^{1f}\_i,\\Theta^{2f}\_{z(i)})$ is then:
> > > >
> > > > $$
> > > > \\ell\_i^{\\mathrm{div}}(\\theta^{1f}\_i,\\theta^c\_i) = \\mathcal{L}(Y\_i, \\psi(\\phi^{(1)}(X\_i;\\theta^{1f}\_i), \\phi^{(2)}(X\_i;\\Theta^{2f}\_{z(i)}); \\theta^c\_i)) + \\lambda\_{\\mathrm{div}}\, R\_{\\mathrm{div}}(\\theta^{1f}\_i,\\Theta^{2f}\_{z(i)})
> > > > $$
> > > >
> > > > with $\\lambda\_{\\mathrm{div}}>0$ controlling the strength of the diversity constraint.
> > > >
> > > > Algorithmically, this change affects only the primary-encoder local update: during local SGD, each sampled client $i$ in cluster $z$ replaces $\\ell\_i$ by $\\ell\_i^{\\mathrm{div}}$ when updating $(\\theta^{1f}\_i,\\theta^c\_i)$, while the secondary encoder parameters $\\Theta^{2f}\_z$ remain fixed in this phase and are still updated later by the usual secondary-encoder enrichment step of FedDAG.   All other parts of the pipeline (cluster-level FedAvg, CC-Graph–driven enrichment, etc.) remain unchanged.
> > > >
> > > > **Empirical observation.**
> > > > We implemented this “random initialization + diversity regularization” variant and evaluated it on CIFAR-10 and SVHN under the same 30% label-skew, Dirichlet $\alpha' = 1.0$ configuration as in Data Distribution I (see Section 5, main paper), with  $\\lambda\_{\\mathrm{div}}=0.01$.  The resulting accuracies are shown below:
> > > >
> > > > | FedDAG encoder initialization      | CIFAR-10            | SVHN               |
> > > > |-----------------------------------|---------------------|--------------------|
> > > > | Warm-start (ours)             | $90.76 \pm 0.12$    | $93.91 \pm 0.23$   |
> > > > | Random init + regularizer    | $90.93 \pm 0.16$    | $93.86 \pm 0.10$   |
> > > >
> > > > As the table indicates, the “random initialization + diversity regularization” variant achieves performance comparable to the warm-start version of FedDAG on both CIFAR-10 and SVHN, with slightly improved accuracy in some cases.
> > > >
> > > > We have included this experiment in the updated version of the paper (Appendix), which will be uploaded shortly.

---

> > > > > ### Author Response · Authors · 2025-11-24
> > > > > **Comment by authors addressing the questions - Part 5**
> > > > >
> > > > > **Q3.** We thank the reviewer for pointing out these additional relevant works. We have now incorporated all three into our literature review in the updated paper, which will be submitted shortly.
> > > > >
> > > > > **Paper 1**
> > > > > **Citation:** Ding, Shu and Wang, Wei. *Collaborative Learning by Detecting Collaboration Partners.* In *Advances in Neural Information Processing Systems (NeurIPS)*, volume 35, pages 15629–15641, 2022.
> > > > > **Added text in paper:**
> > > > > “Ding & Wang (2022) constructs $K$ shared models based on each client’s dataset contribution.”
> > > > >
> > > > > **Paper 2**
> > > > > **Citation:** Liu, Qiqi, Li, Jiaqiang, Liu, Yuchen, Jin, Yaochu, Lyu, Lingjuan, Wu, Xiaohu, and Yu, Han. *Personalized Federated Learning under Local Supervision.* In *Proceedings of the IEEE/CVF International Conference on Computer Vision (ICCV)*, pages 4069–4079, 2025.
> > > > > **Added text in paper:**
> > > > > “FedSimSup (Liu et al., 2025) uses a local supervisor and data-similarity–weighted inter-learning model to better align global knowledge with heterogeneous local data.”
> > > > >
> > > > > **Paper 3**
> > > > > **Citation:** Li, Zhilong, Wu, Xiaohu, Tang, Xiaoli, He, Tiantian, Ong, Yew-Soon, Chen, Mengmeng, Liu, Qiqi, Lao, Qicheng, and Yu, Han. *Benchmarking Data Heterogeneity Evaluation Approaches for Personalized Federated Learning.* In *International Workshop on Trustworthy Federated Learning*, pages 77–92, 2024.
> > > > > **Added text in paper:**
> > > > > “Meta Learning approaches include personalized FL (Li et al., 2024b) ...”

---

### Official Review · Reviewer_uk8Q · 2025-10-30

**Soundness:** 3
**Presentation:** 4
**Contribution:** 2
**Rating:** 4
**Confidence:** 4

**Summary:**

This paper proposes a clustered federated learning framework, termed FedDAG, which is designed to address performance degradation under heterogeneous (non-IID) client data. Focusing on different non-IID scenarios, FedDAG applied a learnable weight combination for data and gradient similarities integration.  It also includes an adaptive clustering metric to automatically determine the optimal number of clusters. In addition, FedDAG employs a dual-encoder architecture for cross-cluster representation sharing. Experiments across four types of non-IID scenarios (label skew, feature skew, concept shift, and quantity shift) show consistent improvements over several baselines, including PACFL, IFCA, and FedRC

**Strengths:**

1.  The paper clearly defines different types of non-IID heterogeneity and systematically designs corresponding modules to treat each.

2. The framework integrates clustering, similarity measurement, and inter-cluster knowledge transfer into a cohesive system.

3. FedDAG performs well across datasets and heterogeneity settings, demonstrating robustness in practice.

4. Presentation is clear and easy to follow.

**Weaknesses:**

1. The novelty of this work is limited. Most components extend existing approaches with minor tweaks. The framework lacks a central new idea or theoretical insight.

2. There is no convergence, optimality, or complexity analysis; the method’s robustness under different heterogeneity settings is justified only empirically.

3. The ablation studies are incomplete.

- The sensitivity of the gradient similarity module to the number of local optimization steps and the sparsification ratio is not examined, leaving uncertainty about the balance between communication efficiency and clustering accuracy.

- The data similarity module includes both class-frequency weighting and entropy-based weighting for fusion, yet their individual effects are not isolated or discussed.

- The paper does not verify whether each designed module actually mitigates the corresponding non-IID type (e.g., whether the class-wise weighting improves label skew, or whether the entropy weighting helps with concept shift). This makes it difficult to assess whether the proposed design behaves as intended.

4. All core experiments are conducted on relatively small and clean benchmark datasets (CIFAR-10/100, FMNIST, and SVHN). Although the appendix includes an additional comparison on Tiny-ImageNet, this test remains limited in scope and scale and is insufficient to demonstrate robustness or scalability to more realistic, large-scale federated settings.

**Questions:**

1. Can the authors provide theoretical convergence or complexity analysis for FedDAG in different non-IID cases?

2. How do the different modules (data weighting, gradient sparsification, adaptive clustering) behave under various types of non-IID settings?

3. What is the computational and communication overhead compared to PACFL or IFCA?

4. Can the authors test FedDAG on other modalities or large datasets?

---

> ### Author Response · Authors · 2025-11-25
> **Comment by authors addressing the questions - Part 1**
>
> **W1.** Among the novel components of **FedDAG**, the most distinctive is the **CC-Graph–guided cross-cluster knowledge sharing via a dual-encoder architecture**. At a high level, this dual-encoder design may appear similar to personalized FL approaches that separate shared vs. local heads, or to FL frameworks where clients maintain local feature extractors alongside a global feature extractor. We agree that the idea of decoupling a model into two components is not new by itself; however, how these encoders are trained and what they represent in **FedDAG** is fundamentally different. Most pFL and feature-skew FL works use a decomposition of the form:
>
>
> - **Global/shared part:** a shared feature extractor or shared head
> - **Personalized part:** a per-client feature extractor or a per-client head
>
>
> So their effective decomposition is:
>
>
> > (shared feature across all clients) + (private feature per client)
>
>
> ---
>
>
> ### How FedDAG differs
>
>
> #### 1. Global feature vs. selective complementary features
>
>
> In these pFL approaches, sharing a single global feature extractor can be suboptimal. The global feature space is influenced by all clients, including:
>
>
> - clients with very different learning objectives or data distributions, and
> - clients with poor or unstable features (e.g., very few samples for certain classes)
>
>
> This mixture can lead to negative and noisy feature transfer: aligning a representation with *all* clients is often harmful, especially when only a subset of clients are truly relevant or complementary.
>
>
> In contrast, **FedDAG** does not simply blend all clients into one global representation. Instead, it:
>
>
> - uses the CC-Graph to identify which clusters are actually beneficial (i.e., have complementary class distributions and sufficient data), and
> - imports features only from those complementary clusters via the secondary encoder.
>
>
> In this sense, **FedDAG** factorizes what would otherwise be a monolithic global feature space into:
>
>
> > (cluster-specific personal features) + (selected complementary features from other clusters)
>
>
> thereby avoiding contamination from unrelated or comparatively less relevant client sources.
>
>
> ---
> #### 2. Secondary encoder trained directly on other clusters’ data
>
>
> In **FedDAG**, the secondary encoder of the learner cluster is explicitly trained on the datasets of clients from source clusters:
>
>
> - The secondary encoder parameters are sent to a source cluster identified by the CC-Graph.
> - At the source cluster, that encoder is locally trained on the clients’ data, and the resulting gradients or updates are then sent back to the learner clusters.
>
>
> Thus, the secondary encoder acts as a parameter carrier that travels across clusters, learns on other clusters’ data, and then returns with an updated gradient.
>
>
> This is fundamentally different from a “shared head” or a single global feature extractor in feature-skew FL. Those approaches typically rely on shared backbones plus local heads, or on knowledge-transfer signals such as pseudo-data, distilled logits, or prototype features; to our knowledge, they do **not** perform explicit remote training of a model component on other clients’ local data. This limits their ability to fully exploit the complementary feature structure present in other domains.
>
>
> We also empirically verify in our ablation studies (Table 2, Section 5) that the gain is not merely due to the increased parameter size of the second encoder.
>
>
> ---
>
>
> By leveraging the structure and training workflow of clustered federated learning, **FedDAG**’s secondary encoder performs *targeted cross-cluster feature extraction* guided by the CC-Graph, rather than functioning as just another shared or personalized feature extractor. This cross-cluster training mechanism, combined with the CC-Graph, constitutes the main novelty of our dual-encoder design relative to existing pFL and feature-skew FL works.
>
>
> ----
>
> **W2.** Thank you for the comment. We would like to clarify that the paper **does include** both convergence and complexity analyses.
>
> - **Convergence analysis** is provided in **Appendix A.4 (Convergence Analysis)**, where we formally characterize FedDAG as a shared–personalized optimization problem and derive a convergence rate under standard smoothness, variance, and heterogeneity assumptions.
> - **Communication and computation complexity** is analyzed in **Appendix A.5 (Communication and Computation Complexity)**, where we break down the per-round costs for both primary and secondary encoder updates, including the CC-Graph sharing phase.
>
> We will revise the main paper to ensure these analyses are more clearly signposted so that readers do not miss the theoretical sections.

---

> ### Author Response · Authors · 2025-11-25
> **Comment by authors addressing the questions - Part 2**
>
> **W3. i) Sensitivity of gradient similarity to local steps and sparsification.**
> We thank the reviewer for this comment. We clarify that the sensitivity to the number of local optimization steps (\(t_g\)) is already evaluated in the initial submission (Appendix B.2, “Hyperparameter Tuning”), where we explicitly vary \(t_g\) and measure FedDAG’s performance. We summarize those results here and, in addition, we now report complementary experiments on the sparsification ratio of the gradient signatures.
>
> ---
>
> ### (1) Sensitivity to the number of local optimization steps
>
> In Appendix B.2 of the original submission, we study $t_g$, which is precisely the number of local warm-up rounds each client performs before sending gradient information to the server. In this study, we use only the **gradient similarity matrix** as the proximity matrix (no data-based similarity), to isolate the effect of gradients on clustering. We experiment on **Data Distribution I** with 30% label skew and Dirichlet parameter $\alpha' = 1$ on the **FedDAG*** single-encoder variant. Each local round consists of 10 local steps.
>
> **Test accuracy vs. number of warm-up rounds $t_g$:**
>
> | t_g (warm-up rounds) | CIFAR-10       | SVHN           | FMNIST         |
> |--------------------------|----------------|----------------|----------------|
> | 1                        | 80.81 ± 0.59   | 84.82 ± 0.24   | 93.18 ± 0.11   |
> | 2                        | 83.34 ± 0.52   | 90.05 ± 0.16   | 93.18 ± 0.11   |
> | 3                        | 83.34 ± 0.52   | 90.05 ± 0.16   | 93.18 ± 0.11   |
>
> This indicates that \(t_g = 2\) is already sufficient for good clustering. Larger \(t_g\) produces the same clustering and thus only increases local computation without improving accuracy.
>
> ---
>
> ### (2) Sensitivity to gradient sparsification ratio
>
> To address the second part of the comment, we additionally evaluate how accuracy varies with different sparsification ratios when constructing the gradient similarity matrix.
>
> Similar to the local warm-up experiment, this sparsification study again uses **only gradient similarity** (no data-based similarity), so that sparsification is the only changed factor. The experimental setup is the same as above, and we fix the local warm-up to $t_g = 2$ rounds, with 10 local steps per round,  on the **FedDAG*** single-encoder variant.
>
> **Test accuracy vs. sparsification ratio:**
>
> | Sparsity (coords kept) | CIFAR-10       | SVHN           | FMNIST         |
> |------------------------|----------------|----------------|----------------|
> | 0.1%                   | 82.52 ± 0.60   | 89.23 ± 0.25   | 92.81 ± 0.14   |
> | 0.5%                   | 83.34 ± 0.52   | 90.05 ± 0.16   | 93.18 ± 0.11   |
> | 1%                     | 83.34 ± 0.52   | 90.05 ± 0.16   | 93.18 ± 0.11   |
> | 2%                     | 83.34 ± 0.52   | 90.05 ± 0.16   | 93.18 ± 0.11   |
> | 20%                    | 83.34 ± 0.52   | 90.05 ± 0.16   | 93.18 ± 0.11   |
>
> We observe that sparsifying the gradient and keeping just or above 0.5% of the coordinates is already sufficient to obtain good accuracy. In contrast, 0.1% sparsity yields slightly lower accuracy because the similarities are noisier, resulting in somewhat worse clustering. Based on these results, in our experiments, we keep the sparsification ratio between 1–2%.
>
> **W3. ii) Weighted class similarity vs entropy-based fusion**
> To isolate the contribution of combining data- and gradient-based information for similarity computation, we perform an ablation study where, instead of using the combined similarity, we construct the client similarity matrix using either gradient-only or data-only similarity. We run this ablation on the **FedDAG*** single-encoder variant, where we switch off the dual-encoder representation sharing and keep only a single encoder, so that we can attribute performance differences directly to the similarity design without interference from other techniques. Concretely, we compare **FedDAG\*-Grad** (gradient-only similarity), **FedDAG\*-Data** (data-only similarity), and **FedDAG\*-Data+Grad** (our proposed combined similarity). We evaluate on CIFAR-10 and SVHN under **Data Distribution I with 30% label skew and a high degree of quantity shift (Dirichlet α′ = 0.25)** and report the accuracy results in the table below.
>
>
> | Method | CIFAR-10 | SVHN |
> | ----------- | ---------------- | ---------------- |
> | FedDAG*-Grad (gradient-only) | 83.79 ± 0.50 | 89.05 ± 0.19 |
> | FedDAG*-Data (data-only) | 85.03 ± 0.33 | 89.52 ± 0.21 |
> | **FedDAG\*-Data+Grad (combined)** | **86.79 ± 0.21** | **90.97 ± 0.13** |
>
>
> As shown in the table, **weighted class-wise data-based similarity** (FedDAG*-Data) achieves better accuracy than gradient-only similarity, and **combining** data-based and gradient-based similarity via our fusion matrix (FedDAG*-Data+Grad) yields better similarity, which in turn produces the best performance, outperforming both single-source variants.

---

> ### Author Response · Authors · 2025-11-25
> **Comment by authors addressing the questions - Part 3**
>
> **W3.iii)** Thank you for this helpful comment. We partially address the contribution of different components of FedDAG through ablation studies, where we disable specific functionality to evaluate how FedDAG performs without that component. For example:
>
> - **(i) Dual-encoder ablation:** disable cross-cluster sharing or switch to a single encoder.   **(reported in Experiment 5)**.
>
> - **(ii) Data–gradient fusion ablation:** use **data-only** or **gradient-only** similarity matrices to construct the adjacency graph and evaluate FedDAG.  **(see W3(ii) in our rebuttal above)**
>
> However, we agree that we have **not yet systematically evaluated each module under different non-IID types**. We are currently running these **module-isolated ablations across multiple non-IID regimes**, and we will include the corresponding results and analysis in the revised manuscript as soon as they are ready.
>
> **W4. Lack of additional datasets** Thank you for this observation. In response to reviewer feedback, we have **expanded our experimental evaluation beyond these settings** in two ways.
>
> First, we added two additional datasets, **PACS** and **Office-Caltech-10**, to our **feature-skew experiments under Data Distribution IV**. Similar to CIFAR-10-C and Tiny-Imagenet-C, we treat each domain as a distinct feature distribution: each client is assigned one domain (out of four), and within that domain, samples are distributed to clients using a Dirichlet sampler. We train on 80% of clients and test on the remaining 20% unseen clients. The results are:
>
> | Algorithm  | PACS      | Office-Caltech-10 |
> |-----------|---------------------|-----------------------------|
> | FedAvg    | 38.47 ± 0.28        | 46.12 ± 0.31                |
> | PerFedAvg | 70.82 ± 0.22        | 67.34 ± 0.23                |
> | PACFL     | 76.63 ± 0.19        | 73.31 ± 0.14                |
> | CFL       | 72.14 ± 0.25        | 68.02 ± 0.19                |
> | FedRC     | 76.28 ± 0.17        | 73.54 ± 0.22                |
> | IFCA      | 75.12 ± 0.21        | 72.48 ± 0.29                |
> | **FedDAG**| **80.34 ± 0.27**    | **76.28 ± 0.20**            |
>
> **FedDAG achieves the best accuracy on both PACS and Office-Caltech-10**, and, together with the corruption-based results on CIFAR-10-C and Tiny-ImageNet-C, this indicates **consistent gains across all four feature-shift benchmarks**.
>
> **Secondly,** we are currently running experiments on **larger and more heterogeneous datasets**, and the corresponding results will be uploaded with the revised version.

---

> > ### Author Response · Authors · 2025-11-26
> > **Comment by authors addressing the questions - Part 4**
> >
> > **Q1.** As noted in W2, the paper already includes both convergence and complexity analyses. **Appendix A.4** provides the general **convergence analysis** for FedDAG, and **Appendix A.5** analyzes the per-round communication and computation cost. These analyses are not tied to one specific non-IID pattern—their constants naturally reflect different types and severities of non-IID data. We will point readers to these sections more clearly in the revised version.
> >
> > ---
> >
> > **Q2.** We have partially examined the role of each module through targeted ablations. In particular:
> >
> > - **Dual-encoder ablation:** we disable cross-cluster sharing or switch to a single encoder.
> > - **Data–gradient fusion ablation:** we construct the adjacency graph using data-only or gradient-only similarity.
> >
> > These experiments (see W3(ii) and W3(iii)) already illustrate the contribution of individual components in a fixed non-IID setting.
> >
> > However, we acknowledge that we have **not yet evaluated each module across different non-IID types** (label skew, quantity skew, feature shift, concept shift). We are currently running these **module-isolated ablations under multiple non-IID regimes**, and the corresponding results and analysis will be included in the revised manuscript.
> >
> > ---
> >
> > **Q3.** We separate the communication and computation overhead of FedDAG into two parts. First, we discuss the **one-time clustering overhead**, and second, we discuss the **steady-state FL training overhead**. The comparison to other approaches such as PACFL and IFCA are also included below:
> >
> > ## The clustering overhead
> >
> > **Computation overhead:** In the initial phase, each client runs $t_g = 2$ local warm-up epochs. These warm-up updates are not discarded: once clustering is assigned, training continues directly from the warmed-up model, and the same warmed-up parameters are used to initialize both the primary and secondary encoders. Computing per-class principal vectors via truncated SVD then costs $\mathcal{O}(F N^2)$ for a client with $N$ samples and feature dimension $F$. This one-time SVD, together with $t_g = 2$ warm-up rounds, constitutes the main clustering-side computation overhead in FedDAG. **Comparison to other cluster FL approaches:**
> >
> > - PACFL also performs an SVD/PCA-type step per client to extract principal components, so the clustering-stage complexity is of the same order; FedDAG mainly adds the reuse of the $t_g$ warm-up for the model during the clustering phase.
> >
> > - In contrast, IFCA-style methods (IFCA, CFL, CFL-GP, FedSoft) re-evaluate clustering at every communication round: over $T$ rounds each client must evaluate $K$ cluster models, leading to roughly $K T$ model evaluations, whereas FedDAG pays the SVD + warm-up cost only once and then trains on a fixed cluster assignment.
> >
> >
> > **Communication overhead:** After warm-up, each client uploads a $k$-sparse gradient $\\tilde{\\Delta}\_i$ (only $1\\text{–}2\\%$ of coordinates) and a small number of per-class principal vectors $U^i\_c$ (about $1\\text{–}2\\%$ of the feature dimension). The resulting communication is negligible compared to sending the full model parameters $|\\Theta|$. **Comparison to other approaches:**
> >
> >    - PACFL has a similar one-shot communication pattern for principal components during clustering, as it also sends principal vectors. FedDAG incurs a slightly higher cost because it additionally sends the sparsified gradient.
> >
> >    - Iterative clustered FL methods such as IFCA/CFL/CFL-GP/FedSoft must send all $K$ cluster models (each of size $M$) to every participating client in every round, so each client receives on the order of $K M$ parameters per round.In contrast, FedDAG performs clustering only once, and the clustering-stage communication consists solely of the sparsified gradient + a few principal vectors—far smaller than transmitting $K$ full models per round when $K > 2$.
> >
> > In the next part, we discuss the FL training overhead of FedDAG and compare it to other approaches such as PACFL and IFCA:
> >
> > ....**continued in the next comment**....

---

> > > ### Author Response · Authors · 2025-11-26
> > > **Comment by authors addressing the questions - Part 5**
> > >
> > > ## Training overhead
> > >
> > > **Computation overhead.**
> > > After clustering is fixed, FedDAG performs local SGD on a dual encoder. If both encoders are updated every round, the local computation per round is at most about $2\times$ that of single-model FL approaches (one update for the primary encoder and classifier, one for the secondary encoder). In our paper, we also propose an alternating training schedule to tackle this, such as using a $K{:}1$ schedule (e.g., 5 rounds updating only the primary encoder, then 1 round updating the secondary encoder), which significantly reduces the effective compute devoted to the secondary encoder. As shown in our scheduling ablation (see experiment on alternate secondary encoder scheduling attached below), a $5{:}1$ schedule keeps accuracy very close to full FedDAG (e.g., $87.65 \pm 0.28$ vs. $89.87 \pm 0.19$ on CIFAR-10) while cutting secondary-encoder compute by roughly a factor of $5$.
> > >
> > > - Compared to **PACFL**, which trains a single encoder per client, FedDAG thus incurs extra local computation due to the secondary encoder, but this is mitigated by using the alternating $K{:}1$ scheduling.
> > > - Compared to **IFCA-style** methods, which require each client to evaluate all $K$ cluster models every round in order to choose its assignment, FedDAG is cheaper: our clients train only their assigned cluster model (plus the secondary encoder under the chosen schedule), rather than performing $K$ full forward/backward passes per round.
> > >
> > > **Communication overhead.**
> > > In steady-state training, FedDAG communicates at most **two encoder parameter sets per round**:
> > > - the **primary (cluster) encoder** and classifier head, and
> > > - the **secondary encoder** (when it is updated in that round).
> > >
> > > Let $M$ be the number of parameters in one encoder. Per round, FedDAG sends on the order of $2M$ encoder parameters to each participating client and receives updates of the same size back.
> > >
> > > - In **PACFL**, each client receives and returns a **single** encoder (plus head), i.e., about $M$ encoder parameters per round; FedDAG roughly doubles this to $2M$ due to the secondary encoder (which can be mitigated by using the alternating $K{:}1$ schedule).
> > > - In **IFCA-style clustered FL** (IFCA, CFL, CFL-GP, FedSoft), given $K$ clusters, the server must send **all $K$ cluster models** (each of size $M$) to every participating client in every round, and each client processes all $K$ models locally. This yields about $K M$ encoder parameters communicated per client per round. When $K > 2$, this $K M$ communication in IFCA-style methods is substantially larger than the $2M$ encoder communication in FedDAG, even though FedDAG maintains a dual-encoder architecture.
> > >
> > > ### Secondary-Encoder Scheduling Experiment
> > >
> > > As described in the training overhead above, to reduce the overhead of updating both encoders every round, we evaluate lighter-weight **\(K:1\)** schedules, where for \(K\) rounds update only the primary encoder followed by one round updating the secondary encoder. These schedules preserve the benefits of the dual-encoder design while significantly lowering compute and communication devoted to the secondary encoder.
> > >
> > > We test **1:1, 5:1, 10:1, and 15:1** schedules under a fixed 80-round budget on CIFAR-10 and SVHN using the same heterogeneous setting as Data Distribution I (30% label skew, Dirichlet $\alpha' = 0.25$). A **10:1** schedule, for example, updates the primary encoder for 10 rounds and the secondary encoder for 1 round, as shown below:
> > >
> > > | Schedule (Primary : Secondary) | CIFAR-10 (%) | SVHN (%) |
> > > |--------------------------------|--------------|----------|
> > > | **Full FedDAG (both)**         | **89.87 ± 0.19** | **92.65 ± 0.11** |
> > > | **1 : 1**                      | 80.34 ± 0.44 | 86.26 ± 0.32 |
> > > | **5 : 1**                      | 87.65 ± 0.28 | 91.31 ± 0.22 |
> > > | **10 : 1**                     | 87.25 ± 0.31 | 91.46 ± 0.20 |
> > > | **15 : 1**                     | 86.54 ± 0.35 | 90.87 ± 0.27 |
> > >
> > > The **1:1** schedule underperforms because the primary encoder is undertrained. The **5:1** schedule achieves the best cost–accuracy trade-off, retaining accuracy close to full FedDAG while reducing secondary-encoder updates by a factor of 5. The **10:1** and **15:1** schedules further reduce cost with some accuracy drop.
> > >
> > > ---
> > >
> > > **Q4.** We are currently running experiments on larger datasets, and we will upload the results as soon as they are ready.

---

> ### Author Response · Authors · 2025-12-03
> **Comment by authors addressing the questions - Part 6**
>
> **Q4.** To accommodate the reviewer’s comment, we additionally evaluate **FedDAG** on a large-scale real-world dataset: **Google Landmarks** [1], following the setup of [2]. Specifically, we consider the **Landmarks-Users-160K** partition, where the dataset is partitioned into **1,000 clients** based on the landmark dataset’s authorship information. All other aspects of the experimental setup are kept the same as in our **Data Distribution I** experiments in Section 5. We compare FedDAG against a group of established FL baselines, and the results are reported in the table below:
>
> | Dataset           | FedAvg             | PACFL        | CFL               | FedGWC            | IFCA              | FedDAG       |
> |-------------------|--------------------|--------------|-------------------|-------------------|-------------------|-------------|
> | Google Landmarks  | 36.53 ± 0.24       | 54.74 ± 0.21        | 45.29 ± 0.28      | 51.51 ± 0.31      | 51.97 ± 0.16     | 58.23 ± 0.15       |
>
> [1] Weyand, Tobias, et al. "Google landmarks dataset v2-a large-scale benchmark for instance-level recognition and retrieval." Proceedings of the IEEE/CVF conference on computer vision and pattern recognition. 2020.
>
> [2] Licciardi, Alessandro, et al. "Interaction-Aware Gaussian Weighting for Clustered Federated Learning." arXiv preprint arXiv:2502.03340 (2025).
>
> **Q2.**  We thank the reviewer for this insightful question. To address it, we have added a set of targeted ablations that isolate the effect of each module (combining data and gradient, adaptive optimal clustering, and dual-encoder representation sharing) under different non-IID regimes.
>
> 1. **Combining data and gradient.** To isolate the contribution of combining data- and gradient-based information for similarity computation, we perform an ablation study where, instead of using the combined similarity, we construct the client similarity matrix using either gradient-only or data-only similarity. We run this ablation on the **FedDAG\*** single-encoder variant, where we switch off the dual-encoder representation sharing and keep only a single encoder, so that performance differences can be attributed directly to the similarity design without interference from other techniques. Concretely, we compare **FedDAG\*-Grad** (gradient-only similarity), **FedDAG\*-Data** (data-only similarity), and **FedDAG\*-Data+Grad** (our proposed combined similarity). We evaluate on CIFAR-10 and SVHN under two different non-IID data distributions:
>    (i) **Data Distribution I** with 30% label skew and a high degree of quantity shift (Dirichlet α′ = 0.25), and
>    (ii) **Data Distribution II** with concept shift (as shown in Section 5)
>
>    The accuracies for these settings are given below.
>
>    **(i) Data Distribution I (label + quantity shift)**
>
>    | Method                         | CIFAR-10        | SVHN           |
>    |--------------------------------|-----------------|----------------|
>    | FedDAG\*-Grad (gradient-only)  | 83.79 ± 0.50    | 89.05 ± 0.19   |
>    | FedDAG\*-Data (data-only)      | 85.03 ± 0.33    | 89.52 ± 0.21   |
>    | **FedDAG\*-Data+Grad (combined)** | **86.95 ± 0.21** | **90.97 ± 0.13** |
>
>    **(ii) Data Distribution II (concept shift)**
>
>    | Method                          | CIFAR-10        | SVHN           |
>    |---------------------------------|-----------------|----------------|
>    | FedDAG\*-Grad (gradient-only)   | 64.52 ± 0.45    | 81.17 ± 0.21   |
>    | FedDAG\*-Data (data-only)       | 67.10 ± 0.23    | 83.15 ± 0.14   |
>    | **FedDAG\*-Data+Grad (combined)** | **67.79 ± 0.27** | **83.73 ± 0.19** |
>
>
>    These results consistently show that the weighted data-based similarity (FedDAG\*-Data) outperforms gradient-only similarity, and that combining data- and gradient-based similarity (FedDAG\*-Data+Grad) yields the best performance in both label+quantity-skew and concept-shift regimes.
>
> ...**continued**...

---

> > ### Author Response · Authors · 2025-12-04
> > **Comment by authors addressing the questions - Part 7**
> >
> > ..**continued**...
> >
> > **Q2. Continued.**
> >
> >
> > 2. **Adaptive Optimal Clustering** In our optimal clustering metric, the total loss
> > $\mathcal{L} = \mathcal{L}\_1 + \lambda \mathcal{L}\_2$
> > combines two terms:
> > (i) the compactness loss $\mathcal{L}_1$, which measures average intra-cluster proximity and encourages tight clusters, and
> > (ii) the degeneracy penalty $\mathcal{L}_2$, which is a \textsc{FedDAG}-specific term that explicitly penalizes *degenerate* very small clusters.
> > Intuitively, $\mathcal{L}_2$ is designed to discourage over-fragmented clusterings that would otherwise look acceptable under a compactness-only metric.
> >
> > To isolate the effect of this degeneracy penalty, we perform an ablation where we compare the full federated-aware loss
> > $\mathcal{L} = \mathcal{L}_1 + \lambda \mathcal{L}_2$
> > against a simplified variant that uses only the compactness term $\mathcal{L}_1$ (i.e., setting $\lambda = 0$ and dropping $\mathcal{L}_2$).
> > In both cases, the server runs the same hierarchical-clustering threshold search over $\alpha$ and selects the clustering that minimizes the corresponding loss.
> > We conduct this ablation on  CIFAR-10, SVHN under Data Distribution I with 30% label skew and strong quantity shift (Dirichlet parameter $\alpha' = 0.25$).  The results for SVHN are given here. And, the results for both CIFAR-10, SVHN, including figures, are also reported in the revised Appendix~B.7  Additional Ablation Studies.
> >
> > The clustering statistics (loss, number of clusters, and cluster formation) are summarized in the table below, where the **Cluster Formation** column compactly summarizes the number of clients per cluster (an entry of the form `Ck:n` means that cluster *k* contains *n* clients).
> >
> > Empirically, we observe that the $\mathcal{L}_1$-only variant does not reliably signal issues when the clustering contains highly imbalanced or degenerate structures: even configurations with very small clusters can still attain relatively low $\mathcal{L}_1$ values.
> > For example, around $\alpha \approx 0.25$, the $\mathcal{L}_1$ loss remains moderate despite the presence of a much smaller cluster (`C5:3`) alongside substantially larger ones (`C2:25`, `C3:21`, `C0:18`).
> > In contrast, the corresponding $\mathcal{L}_1 {+} \lambda \mathcal{L}_2$ value at $\alpha \approx 0.25$ jumps sharply (to 1.000000), because the degeneracy penalty $\mathcal{L}_2$ explicitly penalizes such small or unbalanced clusters.
> > More generally, with the combined loss $\mathcal{L}_1 + \lambda \mathcal{L}_2$, the clustering loss does not simply decrease monotonically as the number of clusters increases; when degenerate cluster formations appear, $\mathcal{L}_2$ raises the total loss and marks these configurations as undesirable.
> > This behavior is crucial for FedDAG to automatically select an optimal clustering, as it helps avoid degenerate solutions and favors clusterings with more balanced client groups.
> >
> > ...**continued**...

---

> ### Author Response · Authors · 2025-12-04
> **Comment by authors addressing the questions - Part 8**
>
> ...**continued**...
>
> ### SVHN clustering statistics (Data Distribution I, 30% label skew, $\alpha' = 0.25$)
>
> | $\alpha$ | Cluster Count | Cluster Formation (up to 10 smallest clusters)                                          | Loss ($\mathcal{L}_1$ only) | Loss ($\mathcal{L}_1 {+} \lambda \mathcal{L}_2$) |
> |---------:|--------------:|-----------------------------------------------------------------------------------------|-----------------------------:|-----------------------------------------------:|
> | 0.050    | 22            | [C1:1, C6:1, C7:1, C14:1, C16:1, C18:1, C19:1, C20:1, C21:1, C0:2]                      | 0.220788                     | 0.230788                                       |
> | 0.100    | 17            | [C1:1, C5:1, C6:1, C11:1, C13:1, C14:1, C15:1, C16:1, C10:2, C7:4]                      | 0.288712                     | 0.298712                                       |
> | 0.150    | 11            | [C1:1, C9:1, C10:1, C5:2, C8:3, C6:4, C2:12, C0:15, C7:15, C4:21]                       | 0.398571                     | 0.408571                                       |
> | 0.200    | 9             | [C1:1, C5:2, C8:3, C6:4, C2:12, C7:15, C0:17, C4:21, C3:25]                             | 0.418067                     | 0.433833                                       |
> | 0.250    | 6             | [C5:3, C1:16, C4:17, C0:18, C3:21, C2:25]                                               | 0.424598                     | 1.000000                                       |
> | 0.300    | 5             | [C1:16, C4:17, C0:18, C3:21, C2:28]                                                     | 0.399406                     | 0.409406                                       |
> | 0.350    | 5             | [C1:16, C4:17, C0:18, C3:21, C2:28]                                                     | 0.399406                     | 0.409406                                       |
> | 0.400    | 5             | [C1:16, C4:17, C0:18, C3:21, C2:28]                                                     | 0.399406                     | 0.409406                                       |
> | 0.450    | 5             | [C1:16, C4:17, C0:18, C3:21, C2:28]                                                     | 0.399406                     | 0.409406                                       |
> | 0.500    | 5             | [C1:16, C4:17, C0:18, C3:21, C2:28]                                                     | 0.399406                     | 0.409406                                       |
> | 0.550    | 4             | [C0:18, C3:21, C2:28, C1:33]                                                            | 0.449504                     | 0.464715                                       |
> | 0.600    | 4             | [C0:18, C3:21, C2:28, C1:33]                                                            | 0.449504                     | 0.464715                                       |
> | 0.650    | 4             | [C0:18, C3:21, C2:28, C1:33]                                                            | 0.449504                     | 0.464715                                       |
> | 0.700    | 4             | [C0:18, C3:21, C2:28, C1:33]                                                            | 0.449504                     | 0.464715                                       |
> | 0.750    | 3             | [C2:28, C1:33, C0:39]                                                                   | 0.821767                     | 0.836128                                       |
> | 0.800    | 2             | [C0:39, C1:61]                                                                          | 0.946721                     | 0.956721                                       |
> | 0.850    | 1             | [C0:100]                                                                                | 0.664269                     | 0.674269                                       |
> | 0.900    | 1             | [C0:100]                                                                                | 0.664269                     | 0.674269                                       |
> | 0.950    | 1             | [C0:100]                                                                                | 0.664269                     | 0.674269                                       |
> | 1.000    | 1             | [C0:100]                                                                                | 0.664269                     | 0.674269                                       |
>
>
>
> ...**Q2. continued**...

---

> ### Author Response · Authors · 2025-12-04
> **Comment by authors addressing the questions - Part 9**
>
> ...**continued**...
>
> **Q2. continued.**
>
> 3. **Dual-encoder representation sharing.** Here, we examine whether the accuracy gains from inter-cluster global representation sharing (GRS)
> via the dual-encoder architecture (see main paper §4) arise from genuine feature enrichment or simply from
> increased model capacity. We already performed this ablation on Data Distribution I (20% label
> skew, Dirichlet α′ = 0.25; see Table 3 in §5). To further observe the behavior of dual-encoder
> sharing under a different non-IID regime, we repeat this ablation in the concept-shift setting (see
> Data Distribution II in §5). Using the same three variants of FedDAG as in Table 3, we obtain the
> following results as shown in Table 16. Similar to the experiment under Data Distribution I (label-
> skew setting), full FedDAG achieves clear accuracy gains over both the single-encoder (FedDAG*)
> and the no-sharing dual-encoder (FedDAG†) variants across all three datasets under concept shift.
> This indicates that cross-cluster representation sharing remains beneficial even when the primary
> challenge is a mismatch in local decision boundaries rather than pure label skew.
>
> Table 16: Ablation of cross-cluster representation sharing under concept shift (Data Distribution II
> in §5), comparing the single-encoder baseline (FedDAG*), a dual-encoder variant without cross-
> cluster sharing (FedDAG†), and full FedDAG.
>
> ---------------------------------------------------------------
> | Algorithm                                | CIFAR-10 | FMNIST   | SVHN     |
> |------------------------------------------|----------|----------|----------|
> | FedDAG* (single encoder)                 | 67.79±0.27 | 86.03±0.21 | 83.73±0.19 |
> | FedDAG† (dual encoder, no sharing)       | 67.51±0.22 | 85.91±0.15 | 83.65±0.24 |
> | FedDAG (dual encoder + sharing)          | **69.13±0.23** | **88.79±0.19** | **85.06±0.26** |
> ---------------------------------------------------------------

---

### Official Review · Reviewer_Lm4p · 2025-10-30

**Soundness:** 3
**Presentation:** 3
**Contribution:** 3
**Rating:** 6
**Confidence:** 4

**Summary:**

FedDAG proposed a new clustered FL algorithm, designed to tackle data heterogeneity. Specifically, first, to improve clustering, FedDAG introduces a hybrid similarity evaluation by combining both data and gradient similarity, and then obtains a novel federated-aware metric to evaluate candidate clusters, enabling the framework to automatically determine the optimal number of clusters. Second, to enhance global representation sharing, FedDAG employs a dual-encoder architecture. This design allows the model to learn both intra-cluster and inter-cluster representation simultaneously during training, which further increases FedDAG’s performance.

**Strengths:**

1. FedDAG can address label skew, feature skew, concept shift, and quantity shift simultaneously. This comprehensive approach is rare in Federated Learning.
2. FedDAG can adaptively determine the optimal number of clusters. This mechanism effectively solves a long-standing practical issues where most methods require setting the number of clusters in advance. It is practical to handle dynamic scenarios, such as the arrival of new clients (Appendix A.7) and data distribution shifts over time (Appendix A.8) .
3. The proposed global representation sharing which is implemented via a dual-encoder architecture and a novel CC-Graph is a key innovation. By considering the "supply" and "demand" of class representations based on data rarity, this mechanism breaks the traditional information silos of clustered FL, allowing clusters to learn from complementary data sources across the entire network for the first time.

**Weaknesses:**

1. FedDAG has large computational overhead due to its complex design, involving an intricate multi-step process (e.g., SVD, clustering, CC-Graph, dual-encoder training, MLP optimization,). Specifically, for computational cost, server-side operations are at least O(N^2) for computing the similarity matrix and clustering, making scalability beyond the 100 clients tested questionable. For communication overhead, the dual-encoder training phase appears to double the communication cost compared to FedAvg, and the proposed mitigation (alternating training) is not clearly presented in the main algorithm.

2. The "category frequency weighting" in formulas (5) and (6) is specifically designed to address the issue of "quantity shift". Directly uploading category frequency is a serious privacy issue, as it completely exposes the label distribution of the client. Although the author acknowledges these issues in Appendix A.6 and proposes that “uniform weighting can be employed in place of class-frequency-based weighting of similarity values to prevent leakage of class distribution information.”, this solution logically completely denies one of the core contributions of the paper.

3. The concept of CC-Graph is novel, but its metric function is overly simplistic. It merely reflects differences in the ranking of numbers of samples belong to the same class among clients, but cannot capture disparities in quantity or quality of samples.

4. The design of the GRS mechanism is also open to debate. Traditional personalized federated learning treats the feature extraction layer as an aggregatable part of the whole parameters while preserving the personalization of the classifier layer. This design assumes feature extraction is globally similar across different datasets and different tasks. The dual-encoder architecture proposed in this paper introduces an independent global feature extractor. It concatenates representations from global extractor($\theta^{2f}$) with those from the cluster feature extractor($\theta^{1f}$) before feeding them to the downstream classification layer. This design lacks compelling justification and instead incurs significant communication overhead. More comparative experimental results and analysis should be included.

5. There is a significant disconnect between the paper's theoretical guarantees and the algorithm's practical operation. The convergence analysis in Appendix A.4 critically relies on Assumption A.4 (Stable clustering). This assumption is directly contradicted by the FedDAG algorithm's own design, which includes a dynamic phase to determine clusters (Algorithm 1, Lines 1-14) and further mechanisms to change cluster assignments in response to newcomers (Appendix A.7) or distribution shifts (Appendix A.8). Consequently, the provided theory only applies to a static scenario and fails to model the algorithm's core dynamic behaviors.

**Questions:**

see the above

---

> ### Author Response · Authors · 2025-11-24
> **Comment by authors addressing the questions - Part 1**
>
> **W1.** We thank the reviewer for raising this important point about computational and architectural overhead. To explore the alternating secondary encoder updates, we are now explicitly evaluating lighter-weight training schedules that trade off dual-encoder benefits against overhead:
> K:1 scheduling: perform (K) consecutive rounds of standard FedDAG primary training and then one secondary-encoder enrichment round with (K) tuned to the edge-device budget.
>
>
> These variants directly address the reviewer’s concern: they retain the architectural dual-encoder capacity (which our ablations show is necessary for robustness), while reducing the effective compute and communication devoted to the secondary encoder. We evaluate four primary:secondary scheduling patterns—1:1, 5:1, 10:1, and 15:1—under a fixed compute/communication budget (80 communication rounds). Here, for example, a 10:1 schedule means we perform 10 rounds updating the primary encoder, followed by 1 round updating the secondary encoder. All experiments are conducted on CIFAR-10 and SVHN under the same heterogeneous setting as Data Distribution~I (30\% label skew, Dirichlet $\alpha'$=0.25). The results are shown below:
>
>
>
>
> ### Secondary-Encoder Scheduling Results
>
>
> | Schedule (Primary : Secondary) | CIFAR-10 (Acc. %)    | SVHN (Acc. %)      |
> |--------------------------------|----------------------|--------------------|
> | **Full FedDAG (both)**  | **89.87 ± 0.19**     | **92.65 ± 0.11**   |
> | **1 : 1**                      | 80.34 ± 0.44         | 86.26 ± 0.32       |
> | **5 : 1**                      | 87.65 ± 0.28         | 91.31 ± 0.22       |
> | **10 : 1**                     | 87.25 ± 0.31         | 91.46 ± 0.20       |
> | **15 : 1**                     | 86.54 ± 0.35         | 90.87 ± 0.27       |
>
>
>
>
> The 1:1 schedule leads to a substantial drop in accuracy because the primary encoder is undertrained. In contrast, the 5:1 schedule provides the best trade-off: it alleviates the computation bottleneck while retaining accuracy that is very close to the full FedDAG model (where both primary and secondary encoders are updated every round). The 10:1 and 15:1 schedules also perform reasonably well, but exhibit a slight degradation in some cases.
>
> ---
>
> **W2.**  We thank the reviewer for raising the concern regarding whether directly reporting per-class sample counts may leak sensitive information. We address this by clarifying how class-frequency information can be protected—or approximated without revealing exact values. The following discussion has been added to **Appendix A.6 (Privacy Considerations)**.
>
> ---
> ### Appendix A.6 Privacy Considerations
> Privacy mechanisms can be used to protect the class-frequency information that we use when weighting similarity values. For example, FLIPS (Bhope et al., 2023) employs a trusted execution environment (TEE) to safeguard label distributions, which are used to select a diverse set of clients during FL training. Similarly, FedDAG could employ differential privacy (DP) (Dwork et al., 2006) by perturbing class counts with carefully calibrated noise such that the contribution of any single example is statistically hidden. Another option is homomorphic encryption (HE) (Gentry, 2009), which would allow the server to compute weights directly on ciphertexts without accessing raw values. Designing and integrating such mechanisms is beyond the scope of the present work, so we do not pursue them further.
>
>
> [1] Bhope, Rahul Atul, et al. "FLIPS: federated learning using intelligent participant selection." Proceedings of the 24th International Middleware Conference. 2023.

---

> ### Author Response · Authors · 2025-11-25
> **Comment by authors addressing the questions - Part 2**
>
> **W3.** Thank you for pointing out this issue. In the original version, the CC-Graph was defined purely in terms of **class counts** at the cluster level. For each class $c$, we computed a “demand’’ term $d\_{p,c}$ that is large when class $c$ is rare in cluster $\mathbb{C}\_p$, and a “supply’’ term $s\_{q,c}$ that is large when class $c$ is abundant in cluster $\mathbb{C}\_q$. The complementarity score between clusters $p$ and $q$ was then
> $$
> H\_{p,q}^{\\text{(old)}} = \sum\_{c \in C} d\_{p,c} \, s\_{q,c},
> \qquad H\_{p,p}^{\\text{(old)}} = -\\infty.
> $$
> This design captures **quantity-based** complementarity (who has more of which class), but we agree it mainly reflects *rankings* of class counts and does not explicitly capture the **quality** of the learned class-$c$ representations in each cluster.
>
> To make CC-Graph sensitive to representation quality, we now incorporate the per-class principal-angle information $\\mathcal{V'}\_{i,j,c}$ between client subspaces (Section 3.2) directly into the CC-Graph metric. Intuitively, $\\mathcal{V'}\_{i,j,c}$ measures how well the class-$c$ feature subspaces of clients $i$ and $j$ are aligned: small angles correspond to similar, high-quality shared representations, while large angles correspond to mismatched or incompatible representations. For each client pair $(i,j)$ and class $c$, we first clip the class-wise angle $\\mathcal{V'}\_{i,j,c}$ to $[0,90^\\circ]$ and map it linearly to $[0,1]$:
>
> $$
> \\tilde{\\mathcal{V'}}\_{i,j,c} = \\min (\\max (\\mathcal{V'}\_{i,j,c},0^\\circ\)\,90^\\circ), \quad  \quad \\Gamma\_{i,j,c}=1 - \\frac{\\tilde{\\mathcal{V'}}\_{i,j,c}}{90^\\circ}
> \\in [0,1].
> $$
>
>
> Here, $\\Gamma\_{i,j,c}$ is close to $1$ when the class-$c$ subspaces of clients $i$ and $j$ are well aligned and close to $0$ when they are almost orthogonal.
>
> To compute how aligned two clusters are for the CC-Graph, we next need a cluster-level measure of aggregate client alignment. To do this, for each class $c$ and each pair of clusters $(p,q)$, we consider all client pairs where one client belongs to $\\mathbb{C}\_p$, the other to $\\mathbb{C}\_q$. We then define the cluster-level alignment:
>
>
>
> $$\\bar{\\Gamma}\_{p,q,c} =
> \\frac{1}{|\mathbb{C}\_p|\,|\mathbb{C}\_q|}
> \sum\_{i \in \mathbb{C}\_p}
> \sum\_{j \in \mathbb{C}\_q}
> \Gamma\_{i,j,c}
> $$
>
>
> where $\\bar{\\Gamma}\_{p,q,c}$ is high when most clients in clusters $p$ and $q$ whose class-$c$ feature subspaces are strongly aligned, and low when all such pairs are poorly aligned.
>
> We then **modulate** the original demand–supply complementarity by this alignment factor, obtaining the revised CC-Graph score:
> $$
> H\_{p,q} = \sum\_{c \in C} d\_{p,c} \, s\_{q,c} \, \\bar{\\Gamma}\_{p,q,c},
> \qquad H\_{p,p} = -\\infty.
> $$
> Thus, the demand and supply terms $d\_{p,c}, s\_{q,c}$ still encode **how much** class $c$ is needed in $\\mathbb{C}\_p$ and how much of it is available in $\\mathbb{C}\_q$ (quantity side), while $\\bar{\\Gamma}\_{p,q,c}$ captures **how well** the corresponding class-$c$ feature subspaces are aligned between the two clusters (quality side). A high $H_{p,q}$ thus occurs when $\\mathbb{C}\_q$ both has more class-$c$ samples than $\\mathbb{C}\_p$ and has geometrically compatible class-$c$ representations. This refinement directly addresses the concern that CC-Graph does not capture **sample quality**. For the **quantity of samples**, we rely on cluster-level demand/supply terms $d_{p,c}$ and $s_{q,c}$, which are computed from aggregated class counts (not just rankings) and thus still encode quantity differences between clusters. To capture quantity, we deliberately avoid using raw per-client class frequencies. As noted in our **W2** response, we introduce a privacy mechanism to protect client-level class frequencies; directly incorporating those raw frequencies into the CC-Graph computation would complicate the metric.
>
> **Empirical observation.**
> Since the revised CC-Graph explicitly models cross-cluster client similarity to capture sample quality, it is particularly important to evaluate it under **concept shift**, where clients may share the same label sets but have substantially different feature distributions for those labels. To assess the practical impact of the revised CC-Graph, we re-ran our concept-shift experiments under the same heterogeneous setting on CIFAR-10, FMNIST, and SVHN, comparing the original CC-Graph against the modified version, as shown below. s shown below, the revised CC-Graph yields slightly increased accuracy across all three datasets, indicating that incorporating the principal–angle–based alignment improves the effectiveness of the complementarity graph.
> | Method              | cifar10         | fmnist         | svhn           |
> |---------------------|-----------------|----------------|----------------|
> | FedDAG                     | 69.13 ± 0.23    | 88.79 ± 0.19   | 85.06 ± 0.26   |
> | FedDAG (revised CC-graph)       | 69.90 ± 0.20    | 88.93 ± 0.13   | 85.34 ± 0.21   |

---

> > ### Author Response · Authors · 2025-11-25
> > **Comment by authors addressing the questions - Part 3**
> >
> > **W4.**  **Comparison to traditional FL.** We thank the reviewer for raising this question about the design of the global representation sharing (GRS) mechanism and the dual-encoder architecture. Classical personalized FL often assumes that a single feature extractor can be globally shared and only the classifier head needs to be personalized, which works well when client feature distributions are roughly aligned. In heavily non-IID settings, however, clients are grouped into clusters with severe label skew and feature shift; in those cases, a shared encoder/feature extractor cannot satisfy both global and local needs, as different clients can have different learning objectives based on their data distribution. FedBR [1] explicitly shows via visualization that feature extraction is not a purely global, universally shared job, as in, even for the same input, local-model and global-model features can differ significantly.
> >
> > To tackle this generalization vs personalization issue, a separate line of FL work maintains both local and global feature extractors or adds supervisors/auxiliary encoders to relate them. For example, FedBR[1] uses a local and a global feature extractor to reduce classifier bias while preserving client-specific features, and FedSimSup [2] lets each client hold two full models—a local supervisor and an inter-learning model—where the supervisor aligns the inter-learning model with heterogeneous local data.
> >
> >
> >
> > **Comparison to other personalized/global methods.** In summary, FedDAG can be included in the family of methods that add an extra encoder to balance generalization and personalization. However, it leverages the clustered FL framework to improve upon existing designs. While the idea of decoupling a model into two components is not new by itself, how these encoders are trained and what they represent in FedDAG is fundamentally different. Most prior global/local feature schemes maintain:
> >
> > - **Global part:** a shared global feature extractor
> > - **Personalized part:** a per-client local feature extractor
> >
> >
> > ## How FedDAG differs
> >
> >
> > ### 1. Global feature vs. selective complementary features
> >
> > In these pFL approaches, sharing a single global feature extractor can be suboptimal. The global feature space is influenced by all clients, including:
> >
> > - clients with very different learning objectives or data distributions, and
> > - clients with poor or unstable features (e.g., very few samples for certain classes)
> >
> >
> > This mixture can lead to negative and noisy feature transfer: aligning a representation with *all* clients is often harmful, especially when only a subset of clients are truly relevant or complementary.
> >
> >
> > In contrast,  FedDAG does not simply blend all clients into one global representation. Instead, it:
> >
> >
> > - uses the CC-Graph to identify which clusters are actually beneficial  and
> > - imports features only from those complementary clusters via the secondary encoder.
> >
> > In this sense,  FedDAG factorizes what would otherwise be a monolithic global feature space into:
> >
> >
> >
> >
> > > (cluster-specific personal features) + (selected complementary features from other clusters)
> >
> >
> > thereby avoiding contamination from unrelated or comparatively less relevant client sources.
> >
> >
> > ---
> > ### 2. Secondary encoder trained directly on other clusters’ data
> >
> >
> >
> >
> > In FedDAG, the secondary encoder of the learner cluster is explicitly trained on the datasets of clients from source clusters:
> >
> >
> >
> >
> > - The secondary encoder parameters are sent to a source cluster identified by the CC-Graph.
> > - At the source cluster, that encoder is locally trained on the clients’ data, and the resulting gradients or updates are then sent back to the learner clusters.
> >
> >
> >
> >
> > Thus, the secondary encoder acts as a parameter carrier that travels across clusters, learns on other clusters’ data, and then returns with an updated gradient.
> >
> > This is fundamentally different from a single global feature extractor in feature-skew FL. Those approaches typically rely on knowledge-transfer signals such as pseudo-data, distilled logits, feature embedding, or prototype features to gather global representation; to our knowledge, they do **not** perform explicit remote training of a model component on other clients’ local data. This limits their ability to fully exploit the complementary feature structure present in other domains.
> >
> > In summary, by leveraging the structure and training workflow of clustered federated learning, FedDAG’s secondary encoder performs *targeted cross-cluster feature extraction* guided by the CC-Graph, rather than functioning as just another shared or personalized feature extractor.
> >
> > [1] Guo, Y., Tang, X., and Lin, T. “FedBR: Improving federated learning on heterogeneous data via local learning bias reduction.” ICML, 2023.
> >
> > [2] Liu, Qiqi, et al. "Personalized Federated Learning under Local Supervision." Proceedings of the IEEE/CVF International Conference on Computer Vision. 2025.
> >
> > .....**continued in next comment** .....

---

> > > ### Author Response · Authors · 2025-11-25
> > > **Comment by authors addressing the questions - Part 4**
> > >
> > > **W4. Continued**
> > >
> > > ### Empirical observation
> > >
> > > To address your concern about the communication overhead of using a dual encoder, and to compare FedDAG with similar pFL global/personalized baselines, we performed the following experiments:
> > >
> > >
> > >
> > > **1. Additional baselines.**  To strengthen the experimental comparison, we additionally compare FedDAG against pFL methods that balance personalization and global sharing—FedBR[1] and FedMix[3].
> > >
> > > FedMix [3] mitigates heterogeneity by constructing privacy-preserving mixed samples from averaged local batches and Mixup-style interpolation, improving robustness without sharing raw data. FedBR [1] reduces local learning bias by balancing classifier outputs on label-agnostic pseudo-data and aligning local and global feature extractors via a min–max contrastive loss.
> > >
> > > The experiment is performed under the same setting as Data Distribution I (see Section 5 of our paper). The results are shown below:
> > >
> > > ### 20% Label Skew
> > >
> > > | Algorithm | CIFAR-10 (Acc. %)    | FMNIST (Acc. %)      | SVHN (Acc. %)        | CIFAR-100 (Acc. %)    |
> > > |-----------|----------------------|----------------------|----------------------|-----------------------|
> > > | FedMix    | 77.94 ± 0.26         | 83.55 ± 0.31         | 83.12 ± 0.29         | 60.33 ± 0.24          |
> > > | FedBR     | 81.62 ± 0.28         | 85.32 ± 0.23         | 84.05 ± 0.30         | 61.61 ± 0.37          |
> > > | **FedDAG**| **90.76 ± 0.12**     | **93.82 ± 0.20**     | **93.91 ± 0.23**     | **72.84 ± 0.30**      |
> > >
> > >
> > > ### 30% Label Skew
> > >
> > > | Algorithm | CIFAR-10 (Acc. %)    | FMNIST (Acc. %)      | SVHN (Acc. %)        | CIFAR-100 (Acc. %)    |
> > > |-----------|----------------------|----------------------|----------------------|-----------------------|
> > > | FedMix    | 76.90 ± 0.33         | 81.96 ± 0.27         | 82.21 ± 0.34         | 53.55 ± 0.30          |
> > > | FedBR     | 81.48 ± 0.37         | 84.12 ± 0.25         | 84.78 ± 0.38         | 56.32 ± 0.33          |
> > > | **FedDAG**| **89.87 ± 0.19**     | **92.72 ± 0.13**     | **92.65 ± 0.11**     | **63.21 ± 0.60**      |
> > >
> > > These additional results demonstrate that FedDAG remains competitive even against non-clustered methods specifically designed to handle heterogeneous data via personalized and global feature components. These results are included in the updated paper to be uploaded shortly.
> > >
> > > **2. Ablation studies** We also empirically verify in our ablation studies (Table 2, Section 5) that the gain is not merely due to increased parameter size due to adding a second encoder.
> > >
> > > **3. Communication overhead comparison to other Clustered FL approaches.** We agree that our communication overhead is higher than in single-model FL, but it is still significantly lower than in most clustered FL methods.
> > >
> > > Let $M$ be the number of parameters in one encoder.
> > >
> > > - **FedDAG:**
> > >   In each round, the server communicates roughly $2M$ parameters to a client (one primary encoder and one secondary encoder).
> > >
> > > - **Other Clustered FL**
> > > In other cluster FL approaches, such as  IFCA, CFL, CFL-GP, FedRC, etc., given $K$ clusters, the server must send all $K$ cluster models (each of size $M$) to every participating client in every round, and each client evaluates all $K$ models locally. This means that about $K M$ parameters are sent per client per round. This also incurs substantial computation overhead, since each client must evaluate all $K$ cluster models in every round.
> > >
> > > When $K > 2$, $KM$ is much larger than the $2M$ parameters used in FedDAG, so clustered FL baselines with iterative clustering typically incur substantially higher communication cost than our dual-encoder design.
> > >
> > >
> > > **4. Communication/computation overhead.** We agree that naively updating two encoders every round would be expensive. To control this cost, in our paper, we propose an alternating schedule where, in each round, either the primary (cluster) encoder or the secondary encoder is updated, but not both simultaneously. We perform experiments on the following type of scheduling:
> > >
> > > - **$K$:1 scheduling:** perform $K$ consecutive rounds of standard FedDAG primary training and then one secondary-encoder training round, with  $K$ tuned to the edge-device budget.
> > >
> > >
> > > In our experiments, we evaluated the trade-offs of using these lightweight mechanisms, for example, how much they slow convergence. Details of this experiment are provided in our response to **Reviewer 3, Weakness 1**. From that experiment, we observed that the 5:1 schedule (five rounds updating the primary encoder, followed by one round updating the secondary encoder) offers the best cost–accuracy trade-off, keeping performance close to full FedDAG (87.65 $\\pm$ 0.28 vs. 89.87 $\\pm$ 0.19).
> > >
> > >
> > >
> > >
> > > [1] Yoon, T., Shin, S., Hwang, S. J., and Yang, E. “FedMix: Approximation of Mixup under mean augmented federated learning.” 2021b.
> > > [3] Guo, Y., Tang, X., and Lin, T. “FedBR: Improving federated learning on heterogeneous data via local learning bias reduction.” ICML, 2023.

---

> ### Author Response · Authors · 2025-11-25
> **Comment by authors addressing the questions - Part 5**
>
> **W5.** Thank you for raising this issue. We clarify here, how our convergence analysis relates to the three dynamic components you mention: (i) the initial clustering phase (Algorithm 1, Lines 1–14), (ii) the handling of newcomers (App. A.7), and (iii) the mechanism for distribution shifts and re-clustering (App. A.8).
>
> **(1) Newcomers (App. A.7).**
> Our convergence theorem in App. A.4 assumes a fixed client set and partition $\\{\\mathbb{C}\_z\\}\_{z=1}^Z$ (Assumption A.4), with the rate controlled by variance and heterogeneity constants such as $\\bar{\\sigma}\_u^2$, $\\bar{\\sigma}\_V^2$, $\\delta\_{\\mathrm{in}}^2$, $\\delta\_{\\mathrm{out}}^2$, and $\\sigma\_{\\mathrm{share}}^2$. The newcomer mechanism in App. A.7 is meant for the practical case where new clients arrive whose distributions do not differ *significantly* from those of existing clients and therefore do not trigger re-clustering. Their effect can be viewed as a mild perturbation of these constants (e.g., slightly changing the empirical averages that define $\\bar{\\sigma}\_u^2$, $\\bar{\\sigma}\_V^2$, and $\\delta\_{\\mathrm{in}}^2$), so the *shape* of the bound remains the same, with updated constants reflecting the enlarged but still structurally similar population.
>
>
>
>
> **(2) Distribution shifts and re-clustering (App. A.8).**
> After the initial phase that combines data- and gradient-based similarity to determine clusters (Algorithm 1, Lines 1–14), our convergence analysis explicitly assumes a *fixed* clustering and a *stationary* data distribution (no *significant* drift that would trigger re-clustering). Theorem A.1 then shows that, under smoothness, bounded variance, and bounded heterogeneity, the stochastic updates (primary and secondary encoder updates, including CC-Graph sharing) drive the gradient norm of this specific objective $F$ down at a certain rate.
>
> In App. A.8, re-clustering is *intentionally* triggered only under *significant* distribution shift, quantified via a Wasserstein-distance threshold on the empirical label distributions. When this threshold is exceeded and local data distributions change substantially, the global objective itself changes from $F^{(0)}$ to a new objective $F^{(1)}$ with different weights, cluster means, and possibly different constants ($\\delta\_{\\mathrm{in}}, \\delta\_{\\mathrm{out}}, \\bar{\\sigma}\_u^2, \\bar{\\sigma}\_V^2, \\sigma\_{\\mathrm{share}}^2$). Consequently, the guarantee we prove for $F^{(0)}$ does *not* automatically transfer to $F^{(1)}$ after a re-clustering event: at that point, the algorithm is effectively optimizing a *new* objective in a new environment.
>
> This phenomenon is not specific to FedDAG: to the best of our knowledge, existing convergence analyses for clustered or personalized FL are all derived under a *stationary* data-generating distribution and a fixed underlying personalization / clustering structure. When the data distribution and grouping change significantly, the underlying optimization problem itself changes, since the objective is now to fit a new distribution. Any convergence bound proved under the old objective is therefore valid only *up to the change point*. PACFL [1], for example, effectively sidesteps drastic distribution shifts and re-clustering in its convergence analysis: mis-clustered clients and newcomers are treated as contributing to larger variance / heterogeneity terms, rather than being modeled via an explicit dynamic re-clustering process. Our analysis follows the same principle. Small mismatches or mild distribution changes (that do not trigger re-clustering) are absorbed into the variance and heterogeneity constants $\\bar{\\sigma}\_u^{2}$, $\\bar{\\sigma}\_V^{2}$, and $\\delta^{2}$, while the formal theorem assumes a fixed partition and hence does not attempt to capture the occasional re-clustering events. In contrast, algorithms such as IFCA re-evaluate cluster assignments every round at the algorithmic level, but their convergence guarantees are also derived with respect to an implicit fixed clustering / task structure, rather than a fully time-varying objective.
>
> Alternatively, we could in principle re-formulate the convergence analysis in a *phase-based* (piecewise-stationary) manner: each phase would correspond to a period between two re-clustering events, with its own fixed partition and re-defined variance / heterogeneity constants, and the overall behavior could then be obtained by stitching together these per-phase guarantees.
>
>
>
> [1] Vahidian, Saeed, et al. "Efficient Distribution Similarity Identification in Clustered Federated Learning via Principal Angles Between Client Data Subspaces." arXiv preprint arXiv:2209.10526 (2022).
>
> ...**continued in the next comment**...

---

> > ### Author Response · Authors · 2025-11-25
> > **Comment by authors addressing the questions - Part 6**
> >
> > **(3) Initial dynamic clustering phase (Algorithm 1, Lines 1–14).** We agree that the convergence analysis in App. A.4 does not explicitly model the initial clustering phase itself. Our intent is different: Algorithm 1, Lines 1–14, uses combined data+gradient similarity to construct a proximity matrix and run hierarchical clustering to obtain a partition of clients into clusters. Once this clustering is fixed, we *then* interpret FedDAG as a shared–personalized optimization problem at the cluster level and analyze the subsequent training dynamics. In other words, the convergence theorem assumes that a clustering $\\{\\mathbb{C}\_z\\}$ (Assumption A.4) has already been obtained from this initial phase, and it studies the behavior *after* clusters are formed, focusing on how the primary and secondary encoders jointly optimize the cluster-level objective. In the revision, we will make this scope explicit by clearly stating that App. A.4 analyzes the stationary training regime that starts once the initial clustering step has completed.
> >
> > We clarify the above points in the revised version by (i) explicitly describing the piecewise-stationary interpretation of FedDAG re-clustering events, and (ii) explaining that our convergence analysis begins *after* the initial clustering phase, once the client partition has been fixed.

---

> ### Comment · Reviewer_Lm4p · 2025-11-26
> **A comprehensive response.**
>
> I appreciated the authors for providing so comprehensive and insightful responses. Most of my concerns have been addressed with reasonable explanations or further solutions.
>
> A few additional suggestions:
>
> 1) While the authors proposed solutions—including privacy concerns and calculating CC-Graph including samples' quantity/quality—these inherently increase the framework's computational overhead also. In the response to W1, the authors only addressed the performance overhead from the dual encoder design, neglecting other costs. For instance, each client must perform SVD decomposition on its own dataset. For large datasets, SVD decomposition itself is a significant computational burden. While I understand that solving a complex problem inherently demands significant computation, performance optimization remains essential for a great and viable framework.
>
> 2) I hope the authors improve the writing to make the paper more straightforward and readable. While the core ideas are clear, the symbolic notation and formal descriptions are quite complex, which is mentioned by other reviewers also. If the new version include improvement of method, such as computing $\Gamma_{p,q,c}$ to address samples' quality and quantity issues, it would make more intricate. I hope the final version will provide more clear and simple formulations.

---

> ### Author Response · Authors · 2025-12-03
> **Comment by authors addressing the questions - Part 7**
>
> **1.** We thank the reviewer for raising this point and fully agree that computational efficiency is essential for a practical framework. To clarify the computational overhead of FedDAG, we discuss two aspects:
>
>
> 1. An explicit complexity comparison between FedDAG’s one-time truncated SVD step and the per-round iterative clustering overhead of IFCA-style methods.
>
>
> 2. A simple alternative scheme that **further reduces** the SVD overhead by subsampling a fixed number of examples per class while keeping the same similarity definition.
>
>
> -   -   -
> ### Comparison of FedDAG truncated overhead to iterative cluster FL approach
>
>
> While our truncated SVD step to compute per-class principal vectors has sizeable overhead, iterative clustered FL approaches like IFCA are substantially more expensive. Those iterative cluster FL approaches send all \(K\) cluster models to each client in every communication round, and each client evaluates **all \(K\) models**  to update cluster assignments. Our analysis below shows that, over all communication rounds, the cumulative overhead of IFCA-style methods becomes orders of magnitude larger than FedDAG’s one-time truncated SVD computation.
>
>
> ### (1) Cost of the SVD-based initial clustering vs. iterative K-model evaluation
>
>
>
>
> FedDAG’s initial clustering computes **per-class principal vectors via truncated SVD** on each client’s feature matrix. For a client with a local sample size $N$ and feature dimension $F$, we form a matrix $X$ of size $N \\times F$. A full SVD of $X$ would cost on the order of
>
>
>
>
> $${\\mathcal{O}\\bigl(\\min\\{N F^{2}, F N^{2}\\}\\bigr)},$$
>
>
>
>
> but FedDAG only needs the top $r \\ll \\min(N, F)$ singular vectors per class. Using standard truncated / randomized SVD methods as described here [1], this reduces the per-client cost to
>
>
>
>
> $${\\mathcal{O}(r N F)},$$
>
>
>
>
> which is linear in both $N$ and $F$. Importantly, this SVD step is executed **once** during a short warm-up / initial clustering phase and is **not repeated in every communication round** during federated training.
>
>
>
>
> In contrast, iterative clustering approaches such as IFCA, CFL, CFL-GP, FedGWC, FedRC, and FedSoft require **evaluating $K$ cluster models on each client in every round** to decide cluster assignments. In each communication round:
>
>
>
>
> - the client evaluates all $K$ models on its $N$ local samples
> - let $M\_{\\text{fwd}}$ denote the cost of a **single forward pass of the full model on one sample** (including all convolutional layers).
>
>
>
>
> Then
>
>
>
>
> - **Iterative clustering cost per round:**
>   $${\\mathcal{O}(K N M\_{\\text{fwd}})},$$
>
>
>
>
> - **Iterative clustering cost over $T$ rounds:**
>   $${\\mathcal{O}(K T N M\_{\\text{fwd}})}.$$
>
>
>
>
>
>
>
>
>
>
> ### Comparing the two costs via a ratio
>
>
>
>
> Consider the ratio between the *total* iterative clustering cost and FedDAG’s one-time SVD cost:
>
>
>
>
> - SVD cost: $\\mathcal{O}(r N F)$
> - Iterative cost: $\\mathcal{O}(K T N M\_{\\text{fwd}})$
>
>
>
>
> So
>
>
>
>
> $$
> R
> = \\frac{\\text{iterative cost}}{\\text{SVD cost}}
> = \\frac{K T N M\_{\\text{fwd}}}{r N F}
> = \\frac{K T}{r} \\cdot \\frac{M\_{\\text{fwd}}}{F}.
> $$
>
>
>
>
> Now plug in conservative assumptions that are actually **favorable to the iterative methods**:
>
>
>
>
> 1. The number of principal vectors is a small fraction of the local data:
>    $$r = 0.02 N \\quad$$  as in 2\% of the client data.
>
>
> 2. The forward cost per sample is tied to the model size.
>    Suppose we take an **extremely small** deep model whose number of parameters is only **10\%** of a dense $F \\times F$ layer. Let the parameter count be
>    $$
>    P = 0.1 F^{2},
>    $$
>    and assume the cost of one forward pass is proportional to the number of parameters:
>    $$
>    M\_{\\text{fwd}} \\approx 0.1 F^{2}.
>    $$
>
>
> Substitute into the ratio:
>
>
> $$
> R
> = \\frac{K T N M\_{\\text{fwd}}}{r N F}
> = \\frac{K T N (0.1 F^{2})}{r N F}
> = 0.1 \\cdot \\frac{K T F}{r}.
> $$
>
>
> Using the earlier assumption $r = 0.02 N$ (i.e., $r$ is 2\% of the local sample size), we obtain
>
>
> $$
> R
> = 0.1 \\cdot \\frac{K T F}{0.02 N}
> = 5 \\cdot \\frac{K T F}{N}.
> $$
>
>
> So, even under these very generous assumptions to the baselines, we get
>
>
> $$
> R \\approx 5 \\cdot \\frac{K T F}{N}.
> $$
>
> ...**continued**...

---

> ### Author Response · Authors · 2025-12-03
> **Comment by authors addressing the questions - Part 8**
>
> ### Concrete numbers for $K$ and $T$
>
>
> - For a moderate local dataset size $N = 500$ samples per client, feature dimension $F = 128$, $K = 5$ clusters, and $T = 500$ communication rounds:
>
>
>   $$
>   R = 5 \\cdot \\frac{5 \\cdot 500 \\cdot 128}{500}
>     = 5 \\cdot 5 \\cdot 128
>     = 3200.
>   $$
>
>
>   That is, the additional clustering overhead of IFCA / CFL-style methods is about **$3200\\times$ larger** than FedDAG’s one-shot truncated SVD clustering per client.
>
>
> - Even for a **single round** ($T = 1$) with $K = 5$, $F = 128$, and $N = 500$:
>
>
>   $$
>   R = 5 \\cdot \\frac{5 \\cdot 1 \\cdot 128}{500}
>     = 6.4,
>   $$
>
>
>   meaning the per-round iterative clustering overhead is already about **$6.4\\times$ larger** than the SVD cost, and then it grows **linearly** with both $K$ and $T$.
>
>
> So in the regimes we study (small $r$ relative to $N$, moderate $K$, tens–hundreds of rounds $T$, and realistic deep models where $M\_{\\text{fwd}}$ is at least of this order), the iterative $K$-model evaluation overhead clearly dominates FedDAG’s one-time SVD cost.
>
>
> [1] Halko, Nathan, Per-Gunnar Martinsson, and Joel A. Tropp. "Finding structure with randomness: Probabilistic algorithms for constructing approximate matrix decompositions." SIAM review 53.2 (2011): 217-288.
>
>
>
>
> ii)  To further reduce the overhead of the principal vector computation of FedDAG, we propose an alternative approach that uses a subset of the local dataset, instead of using the whole dataset.
> For large local datasets, the above procedure can be made more efficient by computing principal vectors on a \emph{subsampled} per-class dataset instead of using all of $D_{i,c}$. Concretely, for each client $i$ and class $c$, we draw a subset
> $$
> \widehat{D}\_{i,c} \subseteq D\_{i,c},
> \qquad
> |\widehat{D}\_{i,c}| = \min\({m, |D\_{i,c}|\}),
> $$
> where $m$ is a user-specified budget, and the subsample is drawn uniformly at random.
> We then apply the same truncated SVD procedure as above, but on the transpose of $\widehat{D}\_{i,c}$ instead of $D\_{i,c}$, to obtain per-class principal vectors $\widehat{U}^i\_c = [\hat{u}_1,\ldots,\hat{u}_p]$.  All subsequent steps—principal-angle computation, class-frequency weighting, and aggregation into the final similarity matrix—remain identical, as in Eq. 4 $\textendash$ 6.
>
>
>
> **2.** We have updated the descriptions of the CC-graph construction to make them more readable and straightforward. We also introduced the notation $\Gamma_{i,j,c}$ in an intuitive way so it is easier to follow. In addition, we have improved the writing of the secondary encoder training phase to make it more intuitive.

---

### Author Response · Authors · 2025-12-04
**Summary of rebuttals for AC - Part 1**

We would like to thank the reviewers, ACs, senior ACs and PC chairs for their time and effort. We believe we have addressed all reviewer comments and concerns by adding additional experimental analyses and providing further clarifications.

In summary, we added experiments on **3 additional datasets** and incorporated another **3  state-of-the-art methods** pointed by the reviewers to address the reviewers’ concerns. Across all of these 82 new experiments, our method consistently achieved the best performance. We also conducted **28 additional ablation studies** to further quantify the contribution of each component of our approach. Together, these additions have expanded the paper to **35 pages**, including the appendix. Below, we summarize how we addressed each of the reviewers’ comments and concerns.

**Reviewer 1 (reviewer Lm4p)** raised concerns regarding computational and communication overhead, leakage of class-frequency information, CC-Graph construction metric, novelty clarification of the dual encoder, and clarification regarding convergence analysis.

We addressed all these concerns raised by Reviewer 1 in great detail by providing additional experimental results and clarifications. In the revised manuscript, we:

(i) clarified the alternating $K : 1$ primary/secondary encoder schedule and its overhead–accuracy trade-offs (App. A.11);

(ii) expanded Appendix A.6 with a detailed privacy discussion for class-frequency sharing;

(iii) redesigned the CC-Graph with a subspace-alignment term and added concept-shift experiments (Section 4);

(iv) further clarified the dual-encoder novelty relative to pFL baselines (App. A.10);

(v) refined the convergence analysis via a piecewise-stationary view (App. A.4); and

(vi) reduced the principal-vector/SVD cost (App. A.6).

In the follow-up to our response, the reviewer stated that our replies were **“comprehensive and insightful responses”** and specifically noted that **”most of my concerns have been addressed with reasonable explanations or further solutions”**. We believe that the above remarks indicate that Reviewer 1 was satisfied with how we addressed his concerns.

**Reviewer 2 (reviewer uk8Q)** mainly raised concerns about the lack of convergence and complexity analysis of our algorithm, which are mentioned in multiple weaknesses and questions.
We clarified that these points are already addressed in our paper, where we provide both convergence and complexity analysis, and they are also referenced in our submitted reproducibility checklist.
Among other concerns, Reviewer 2 asked about the behavior of FedDAG’s modules under various types of non-IID data, the computational and communication overhead comparison, and the lack of evaluation on a large-scale dataset. We also addressed these points by providing additional analysis, incorporating a new large-scale dataset, and adding new experiments and ablation studies. In the revised manuscript, we:

(i) clarified the core novelty of FedDAG’s CC-Graph–guided dual encoder (App. A.10);

(ii) more clearly indicated the existing convergence (App. A.4) and complexity (App. A.5) analyses;

(iii) added sensitivity studies on warm-up steps and gradient sparsification (App. B.2);

(iv) added ablation studies on combining data and gradient, adaptive optimal clustering, and dual-encoder representation sharing under different non-IID settings (App. B.7);

(v) provided a detailed compute/communication overhead comparison, including $K : 1$ dual-encoder scheduling, against PACFL/IFCA (App. A.6);

(vi) extended our evaluation with new feature-shift benchmarks on PACS and Office-Caltech-10 (App. B.4); and

(vii) added a real-world experiment on Google Landmarks (Landmarks-Users-160K) (App. B.5).


**Reviewer 3 (reviewer VAQF)** mainly inquired straightforward questions about the effectiveness of our optimal clustering technique on data with inherently many clusters, alternative encoder initialization strategies, and additional related work. The reviewer also raised concerns about computational/architectural overhead, clustering stability and cost, and the clarity and readability of the Global Representation Sharing section.

We responded to these points directly, and addressed the Reviewer 3’s concerns with the following revisions:

(i) clarified the alternating $K : 1$ primary/secondary encoder schedule and its overhead–accuracy trade-offs (App. A.11);

(ii) added a fine-grained clustering-overhead analysis contrasting \textsc{FedDAG} with IFCA/CFL methods (App. A.6);

(iii) simplified and reorganized the CC-Graph and secondary-encoder sharing description for readability (Section 4);

(iv) added scalability experiments on $>10$ ground-truth clusters (App. B.10);

(v) evaluated a random-init + diversity-regularizer dual-encoder variant (App. B.8); and

(vi) incorporated the reviewer’s suggested related works into the literature review (Section 2).


...**continued**...

---

> ### Author Response · Authors · 2025-12-04
> **Summary of rebuttals for AC - Part 2**
>
> ...**continued**...
>
> **Reviewer 4 (reviewer 8htn)** stated some concerns related to our paper regarding the limited literature review section, leakage of class frequency information, clarification of dual-encoder novelty, additional baselines and experiments, and layout changes. The reviewer specifically stated that **”I would like to change my score if the authors could address the concerns”**.
>
> We responded to all of the reviewers’  concerns in detail. Later, the reviewer responded that **“most of my concerns have been addressed”**, and the reviewer appeared to be waiting for the updated PDF with the changed layout (regarding weakness W6) before changing his score. In the revised manuscript, we:
>
> (i) expanded the related work in Section 2 and Appendix A.1;
>
> (ii) added a new privacy discussion on class-frequency leakage in Appendix A.6;
>
> (iii) clarified the novelty of the dual-encoder and CC-Graph–guided secondary encoder training in Section 4;
>
> (iv) introduced additional baselines (FedBR, FedMix, FedRC, FBLG) and new feature-shift experiments on PACS and Office-Caltech-10 with explicit mapping to label, feature, concept, and quantity skew (App. B.4); and
>
> (v) relocated Table 2 and Figures 1–2 into the main body to better align the narrative with the key visuals.
>
>
>  Based on the reviewers’ response and our updated manuscript,  we believe we responded to all of the reviewer 4’s  concerns and, as clearly stated, the reviewer was satisfied with our responses.

---

### Meta-Review · Area_Chair_6CrU · 2026-01-08

**Summary:**

FedDAG introduces an innovative clustered Federated Learning framework, effectively breaking the traditional
information silos of clusters by integrating global data representations and gradient similarities.

From my point of view, the authors  have successfully addressed many concerns
raised leading to a reasonable level of acceptance.

More specifically, the rebuttal phase transformed the paper  into a robust contribution.
By addressing privacy concerns, reducing computational overhead,
and mathematically refining the clustering logic, the authors have delivered an interesting
solution for the most challenging heterogeneous FL environments. The significant empirical
gains and the responsiveness to technical critiques fully justify an  accept.

**Reviewer Concerns:**

addressed :
- privacy
- computational overhead

**Reviewer Scores:**

I am not able to answer this

---

### Decision · Program_Chairs · 2026-01-26

Accept (Poster)